# The origin of septin ring size control in budding yeast

Igor V Kukhtevich [ID] [1,4], Sebastian Persson [ID] [2,3,4], Francesco Padovani [ID] [1], Robert Schneider [ID] [1], Marija Cvijovic [ID] [2,3] ✉ & Kurt M Schmoller [ID] [1] ✉

## Abstract

The size of organelles and cellular structures needs to be tightly regulated and coordinated with overall cell size. A well-studied example is the Cdc42-driven polarization and subsequent septin ring formation in *Saccharomyces cerevisiae*, where the size of the resulting structures scales with cell size. However, the mechanisms underlying this scaling remain unclear. Here, we combine live-cell imaging, genetic perturbations, and three-dimensional mathematical modeling to investigate how septin ring size is controlled. Our integrative approach reveals that positive feedback in the polarization pathway, together with an increase of the amount of polarity proteins as cell size grows, can explain the scaling of the Cdc42 cluster and, consequently, septin ring diameter. Additionally, we show that in cells lacking the formin Bni1, where F-actin-cable assembly and directed polarization are disrupted, exocytosis becomes diffuse, leading to abnormally large septin rings. By integrating new experimental findings and mathematical modeling of yeast polarization, our study provides insights into the origin of septin ring size control.

**Keywords** Cdc42 Polarization; Septin Ring; Cell Size; Mechanistic Modeling; Budding Yeast
**Subject Category** Cell Adhesion, Polarity & Cytoskeleton

## Introduction

Self-assembly processes coordinate the reproducible and timely formation of subcellular structures with a defined spatiotemporal organization and are thereby critical for cell survival. A common model for self-assembly is the process of bud-site formation in the budding yeast *Saccharomyces cerevisiae*. Bud formation is a multistep process that employs regulatory strategies that are widespread across cellular processes and organisms [Bi and Park, 2012; Marquardt et al, 2019]. This includes the selection of a single bud site partially through symmetry breaking via cell polarization, followed by septin assembly and the creation of a bud neck of a specific size at the polarization site [Witte et al, 2017; Moran et al, 2019].

Cell polarization in budding yeast is well-studied experimentally and computationally [Park and Bi, 2007; Bement et al, 2024]. During polarization, Cdc42 cycles between three states (Fig. 1A): an active membrane-localized GTP-bound-state promoted by guanine nucleotide-exchange factors (GEFs), an inactive membrane-localized GDP-bound-state promoted by GTPase-activating proteins (GAPs), and a cytosolic GDI-bound state. Cdc42-driven recruitment of the GEF Cdc24 [Bose et al, 2001; Butty et al, 2002], combined with the fact that 2D diffusion of proteins in the membrane is slower than 3D diffusion in the cytosol [Woods and Lew, 2019], creates a positive feedback system that promotes the formation of a single active Cdc42-GTP cluster. In support of the central role of positive feedback, optogenetic recruitment of the GEF Cdc24 promotes polarization at the recruitment site [Witte et al, 2017]. Furthermore, computational models of positive feedback capture several observed phenomena, such as competition between polarity clusters [Goryachev and Leda, 2017].

In addition to positive feedback, negative feedback within the polarization pathway has been proposed [Kuo et al, 2014; Rapali et al, 2017]. In a phosphosite Cdc24 mutant, excess Cdc42 accumulates at the bud site [Kuo et al, 2014], suggesting a major feedback operates via inhibitory phosphorylation of the GEF Cdc24. Furthermore, in vitro experiments have shown that Bem1's ability to activate Cdc24 is reduced when Cdc24 is phosphorylated [Rapali et al, 2017]. Even though the precise mechanisms of negative feedback remain unknown, computational modeling has shown that it enhances the robustness of the polarization [Howell et al, 2012], suggesting it is one of the multiple mechanisms ensuring reliable polarization.

When the Cdc42-GTP cluster reaches its maximum size, septin recruitment starts, followed by septin ring assembly [Park and Bi, 2007]. Both the Cdc42-GTP cluster area as well as the diameter of the fully formed septin ring increase with cell volume, which is not explained by the increase in local curvature [Kukhtevich et al, 2020]. As we showed earlier, the septin ring diameter can be decoupled from the cluster area by deleting the formin *BNI1*, which leads to a larger septin ring, even though the Cdc42 cluster area is mostly unchanged. While this suggests that the size of the septin ring is set through a combination of septin recruitment and Cdc42 polarization, the mechanisms underlying the formation and size homeostasis of the structures at the budding yeast bud neck are still elusive.

Here, we use computational modeling in combination with yeast genetics and microfluidics-based time-lapse imaging to investigate

[1]Institute of Functional Epigenetics, Molecular Targets and Therapeutics Center, Helmholtz Zentrum München, Neuherberg, Germany. [2]Department of Mathematical Sciences, Chalmers University of Technology, Göteborg, Sweden. [3]Department of Mathematical Sciences, University of Gothenburg, Göteborg, Sweden. [4]These authors contributed equally: Igor V Kukhtevich, Sebastian Persson. ✉E-mail: marija.cvijovic@chalmers.se; kurt.schmoller@helmholtz-munich.de

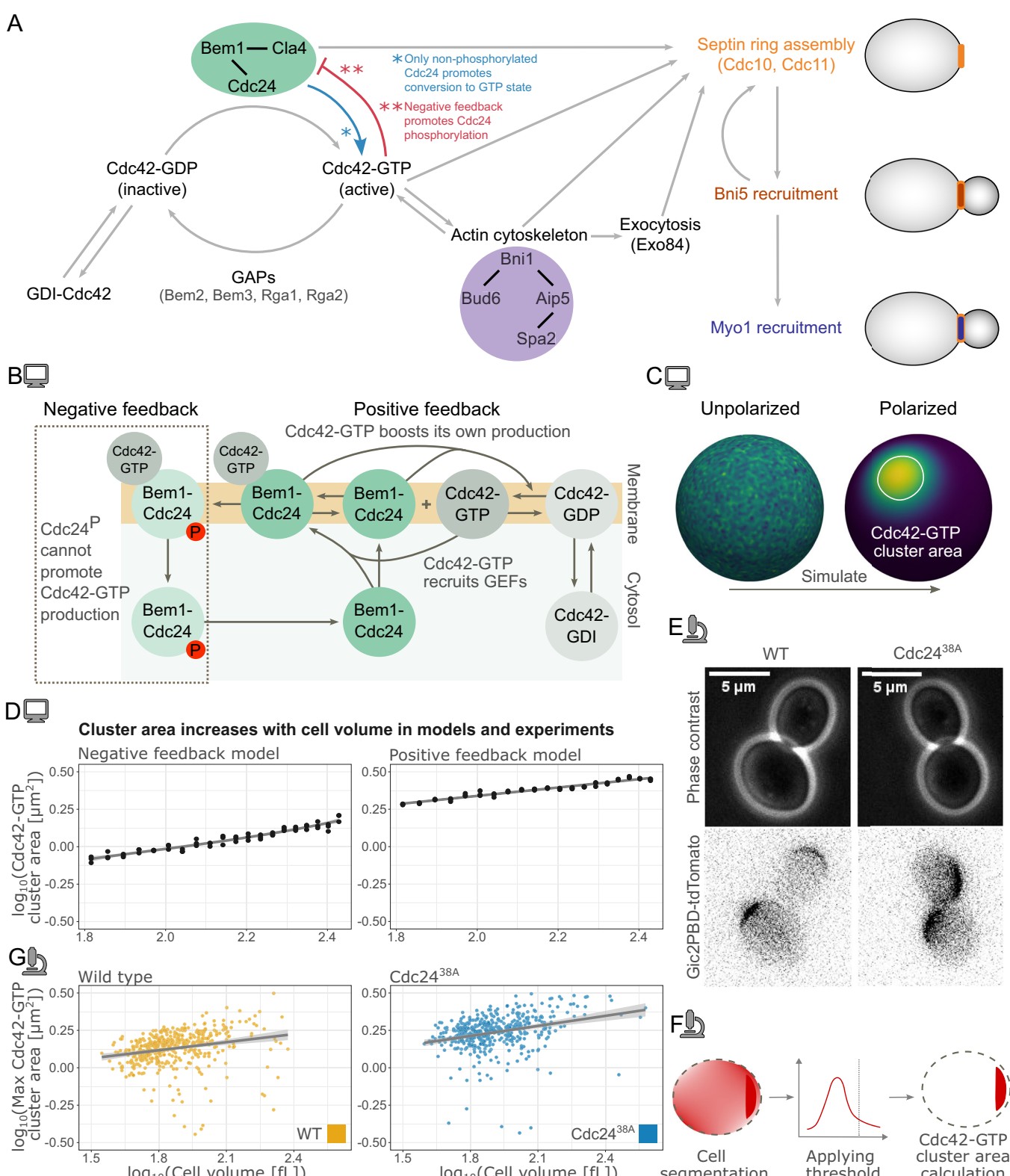

**A**

Bem1 — Cla4
Cdc24

*Only non-phosphorylated Cdc24 promotes conversion to GTP state

**Negative feedback promotes Cdc24 phosphorylation

Cdc42-GDP (inactive)

Cdc42-GTP (active)

GDI-Cdc42

GAPs (Bem2, Bem3, Rga1, Rga2)

Actin cytoskeleton

Bni1
Bud6    Aip5
Spa2

Exocytosis (Exo84)

Septin ring assembly (Cdc10, Cdc11)

Bni5 recruitment

Myo1 recruitment

**B**

Negative feedback

Positive feedback
Cdc42-GTP boosts its own production

Cdc42-GTP

Bem1-Cdc24 P

Cdc24P cannot promote Cdc42-GTP production

Bem1-Cdc24 P

Cdc42-GTP

Bem1-Cdc24

Bem1-Cdc24

+

Cdc42-GTP

Cdc42-GDP

Membrane

Cytosol

Cdc42-GTP recruits GEFs

Bem1-Cdc24

Cdc42-GDI

**C**

Unpolarized          Polarized

Cdc42-GTP cluster area

Simulate

**D**

**Cluster area increases with cell volume in models and experiments**

Negative feedback model

Positive feedback model

**E**

WT          Cdc24³⁸ᴬ

Phase contrast

5 µm          5 µm

Gic2PBD-tdTomato

**G**

Wild type          Cdc24³⁸ᴬ

WT          Cdc24³⁸ᴬ

**F**

Cell segmentation

Applying threshold

Cdc42-GTP cluster area calculation

**Figure 1. Regulatory pathways of Cdc42 polarization and analysis of its dependence on cell size.**

(A) Schematics of the regulatory pathways controlling cell polarization and downstream contractile ring assembly at the polarization site. (B) Schematic of the positive and negative feedback computational models for Cdc42-GTP polarization. In the positive feedback model, the components inside the dashed box are removed. P denotes phosphorylation. (C) Starting from a randomly perturbed unstable steady state, the model was simulated to a polarized steady state, and the Cdc42-GTP cluster area was then determined. (D) Cdc42-GTP cluster area measured at steady-state (after long simulation time) for the model plotted against cell volume in a double logarithmic scale. In each case, for $n = 60$ (three replicates per volume), cells with volumes in the range of 65 to 270 fL were simulated starting from random initial conditions. Left plot: Negative-feedback model, slope $= 0.35$ ($t$-test on slope parameter $p < 2 \times 10^{-16}$). Right plot: Positive feedback model, slope $= 0.33$ ($t$-test on slope parameter $p < 2 \times 10^{-16}$). (E) Representative microscopy images of budding yeast cells (phase contrast) and Cdc42-GTP clusters (Gic2PBD-tdTomato) for WT and Cdc24[38A] (negative feedback mutant) cells. (F) Cdc42-GTP cluster area is experimentally measured from microscopy images by applying a threshold and measuring the maximum cluster area reached before bud emergence (for more details on the analysis, see Methods section). (G) Maximum Cdc42-GTP cluster area experimentally measured from microscopy images plotted against the corresponding cell volume in a double logarithmic scale. Left: WT ($n = 463$ cells), slope $= 0.18$ ($t$-test on slope parameter $p < 2.6 \times 10^{-7}$). Right: Cdc24[38A] ($n = 478$), slope $= 0.39$ ($t$-test on slope parameter A $p < 2 \times 10^{-16}$). Two independent replicates were performed for the experiments. Computer icon—modeling results. Microscope icon—experimental results. Source data are available online for this figure.

the origin of the cell-volume dependence of the Cdc42 cluster area and septin ring diameter. We identify that positive feedback in the polarization pathway, along with an increase in the amount of key polarity proteins with cell volume, is sufficient to explain the scaling of the Cdc42-GTP cluster area with cell size. We show that in $bni1\Delta$ cells, F-actin cable assembly and polarization toward the bud site are disrupted, leading to diffuse exocytosis. With a novel mechanistic model of septin ring formation, we show that an increase in the Cdc42 cluster area drives septin ring volume scaling, and that if exocytosis supports septin recruitment, diffused exocytosis can explain the increased septin ring diameter observed in $bni1\Delta$ cells. Furthermore, consistent with experimental findings, modeling predicts that disruption of the Cdc24-dependent negative feedback in the polarization pathway leads to an increased Cdc42-GTP cluster area. Despite this increase in the Cdc42-GTP cluster area, the septin ring diameter is largely unaffected. This decoupling between septin ring diameter and Cdc42 cluster area can be partially recapitulated in our model by increasing the rate at which septin is recruited by polarity factors.

## Results

### Negative feedback in the polarization pathway is not required for scaling of the Cdc42-GTP cluster area with cell volume

Given the key role of Cdc42-GTP in bud site selection and initiation of septin ring assembly, we first investigated Cdc42 polarization and subsequently the consequences on septin ring formation.

We have previously shown that the Cdc42-GTP cluster area increases with cell volume [Kukhtevich et al 2020]. To gain insights into the mechanisms underlying this scaling, we turned to computational modeling. Since Cdc42 polarization is driven by a positive feedback loop [Kozubowski et al, 2008; Witte et al, 2017] and likely also involves a negative feedback loop [Kuo et al, 2014], we initially extended an existing two-dimensional model incorporating both feedback mechanisms [Kuo et al, 2014] into a three-dimensional geometry. This allowed for a more realistic, spherical approximation of the cell. We refer to this model as the 'negative feedback model'. In this model, Cdc42-GTP attracts its own effectors (Bem1-Cdc24-complex) to form a Bem1-Cdc24-Cdc42 complex, which in turn activates Cdc42 (Fig. 1B). Combined with membrane diffusion being slower than cytosolic diffusion, this

creates a positive feedback system where Cdc42 can polarize from stochastic fluctuations [Goryachev and Pokhilko, 2008; Wu et al, 2015]. Moreover, to account for reported phosphorylation-driven negative feedback [Kuo et al, 2014], it is assumed in the model that when the Bem1-Cdc24-Cdc42 complex reaches a high concentration, the Bem1-Cdc24 complex component is phosphorylated. This inhibits the ability of Bem1-Cdc24 to activate Cdc42 (Fig. 1B left).

Starting from a steady state with random perturbations to facilitate polarization, we simulated 60 cells undergoing the transition from an unpolarized to a polarized state with cell volumes ranging from 65 to 270 fL (Fig. 1C). We then determined the Cdc42-GTP cluster area for each cell at the final steady-state time point. In line with experimental observations [Kukhtevich et al, 2020], the simulation results show that the cluster area increases with cell volume ($R^2 = 0.95$) (Fig. 1D left), consistent with a power law relationship $A \sim V^{0.35}$.

Next, we aimed to understand the contributions of the negative and positive feedback in the model and asked whether the positive feedback is sufficient for the Cdc42-GTP cluster area scaling. To assess this, we removed the negative feedback in the model (Fig. 1B left). We refer to this model as the "positive feedback model". Again, cluster area followed a power law relationship with cell volume, $A \sim V^{0.33}$ ($R^2 = 0.98$) (Fig. 1D right). The cluster area is also larger compared to the negative feedback model (Fig. 1D). To validate our results beyond this single model structure, we applied a similar simulation approach to an alternate positive feedback polarization model [Borgqvist et al, 2021] and again found that the Cdc42-GTP cluster area increases with cell volume (Fig. EV1A-C). Additionally, since the mechanisms governing the negative feedback are not fully elucidated, we also explored an alternative way to simulate the effect of decreasing Cdc42-GTP activation. Specifically, we increased GAP activity in the positive feedback model. Consistent with the negative feedback model, also with increased GAP activity cluster area scaled with cell volume, and the cluster area was smaller (Fig. EV1D,E)

Our simulation results demonstrate that the positive feedback alone, i.e., without additional negative feedback, can explain the increase of the Cdc42-GTP cluster area with cell volume. To validate this experimentally, we investigated a strain in which the negative feedback loop is disrupted by a point mutation in Cdc24 [Kuo et al, 2014]. Briefly, Cdc42-GTP activates the p21-activated kinase Cla4, which then phosphorylates Cdc24. In the case of a Cdc24[38A] phosphosite mutant strain [Kuo et al, 2014], phosphorylation is prevented, weakening the negative feedback while keeping the positive feedback intact. To measure the Cdc42-GTP cluster

area, we used a microfluidic device and time-lapse imaging (see Methods section, Fig. 1E,F). As predicted by the model, the maximal Cdc42-GTP cluster area prior to bud emergence increased with cell volume for both Cdc24[38A] and WT strains (Fig. 1G). To confirm this result, we also employed a previously established system based on the tunable expression of the cell size regulator Whi5 to increase the range of experimentally accessible cell volumes [Kukhtevich et al, 2020; Claude et al, 2021]. This approach confirmed that cluster area increases with cell volume for both wild-type Cdc24 and Cdc24[38A] (Appendix Fig. S1).

## Scaling of Cdc42-GTP cluster area with cell volume requires that the amount of polarity proteins increases with cell volume

Results from computational modeling showed that positive feedback alone is sufficient to drive an increase in the Cdc42-GTP cluster area with increasing cell volume (Fig. 1). However, this is under the assumption that the concentration of all polarity proteins in the model is constant. While global protein amounts increase with cell volume, this increase is not proportional for every protein, leading to a cell-size-dependent decrease in the concentration of specific proteins [Swaffer et al, 2021; Lanz et al, 2024]. The dependency can be characterized using the slope of the double logarithmic dependence of protein concentration on cell volume [Claude et al, 2021; Lanz et al, 2024]. Specifically, a 'protein slope' equal to $-1$ is equivalent to a constant protein amount, a negative slope implies that amounts scale sublinearly with volume, and a slope equal to 0 means that concentration is constant and accordingly protein amount scales linearly with volume (Fig. 2A).

To computationally test how the Cdc42 cluster area would be affected by a dilution of polarity proteins with cell volume, we simulated both the 'negative feedback' and "positive feedback" models for different protein slopes. Specifically, we simulated $n = 60$ cells (65 to 270 fL) under three conditions: (i) constant protein amounts (protein slope $= -1$), and (ii-iii) two different scenarios of amounts that increase sublinearly with cell volume (protein slope $= -0.2$ or $-0.44$, respectively). The GAP concentration was kept constant in all simulations. In the negative feedback model (WT model), an increase in the total amounts of Cdc42 and the Bem1–Cdc24 complex leads to a corresponding increase in cluster area with cell volume (protein slope $= -0.2$ or $-0.44$; Fig. 2B). Similar trends hold for the positive feedback (Cdc24[38A]) model (Fig. 2B). However, when Cdc42 and Bem1-Cdc24 complex complex amounts were constant, the negative feedback model did not polarize when we increased volume. In the positive feedback model, the Cdc42-GTP cluster area did not scale with cell volume (protein slope $= -1$, Fig. 2B).

Our modeling results predict that if GAP concentration is constant, the amount of other polarity proteins must increase for the Cdc42-GTP cluster area to increase with cell volume. To investigate if this is indeed the case, we analyzed a recent proteomics dataset where a triple-SILAC workflow was used to measure protein concentrations of different-sized cells that were obtained through G1 arrest in either glucose (SCD) or ethanol/glycerol-containing media (SCGE) [Lanz et al, 2024]. To characterize the cell size dependence, a protein slope was then computed for each polarity protein (Fig. 2C). We found that both GAP and most polarity proteins are maintained at close to constant concentrations across cell volumes (protein slope $= 0$ in Fig. 2C), which is in line with an earlier report [Chiou et al, 2021]. Notably, the key polarity proteins Cdc42 (SCD: mean slope $= -0.14$, SCGE: mean slope $= -0.09$), Cdc24 (SCD: slope $= -0.16$, SCGE: slope $= -0.11$), Bem2 (SCD: slope $= 0.05$, SCGE: slope $= 0.01$), and Bem1 (SCD: slope $= -0.05$, SCGE: slope $= -0.01$) exhibit protein slopes close to zero, well within the range in which the model predicts the Cdc42-GTP cluster area to increase with cell volume (Fig. 2B). Similar trends hold for the polarity proteins in an experiment where different-sized cells were obtained via mutations (Fig. EV1F).

In summary, the amount of key polarity proteins such as Cdc24 and Bem2 increases with cell volume. Our modeling shows that if, instead, the amount of most polarity proteins remains constant, and only the amount of GAPs increases with cell volume, cluster formation becomes unstable and does not scale with volume.

## Potential mechanisms underlying the scaling of Cdc42-GTP cluster area with cell volume

Hitherto, our experimental and modeling results showed that cluster area increases with cell volume. We next turned to computational modeling to explore the mechanistic basis of this relationship.

In both the positive- and negative-feedback models, we found that maximum Cdc42-GTP concentration increases with cell volume (Fig. 2D,E left). Since diffusion strength depends on concentration gradients, this suggests larger cells produce larger clusters via a stronger diffusion-driven flux (Fig. 2H left). This raises the question of why local Cdc42-GTP concentration increases with volume. As shown earlier, polarity protein concentration remains nearly constant across cell volumes. Because membrane area scales quadratically with radius while cytosolic volume scales cubically, larger cells should be able to recruit more polarity proteins to the membrane before cytosolic concentrations become limiting, and the positive feedback should direct this recruitment to the cluster. Our reasoning suggests two possibilities to modulate cluster area: increase Cdc42-GTP concentration in the cluster by disrupting GTP hydrolysis, or limit the availability of key polarity proteins. We computationally tested the first prediction by removing the negative feedback or reducing GAP activity in our models, and found that this indeed leads to larger clusters (Figs. 1D and EV2A,B right), in line with previous modeling studies [Hubatsch et al, 2019]. Computationally, we have shown that a smaller protein slope, and thereby reducing both Cdc42 and Bem1–Cdc24 concentrations, results in smaller Cdc42-GTP clusters (Fig. 2B). However, we found that individually reducing Cdc42 concentration had little effect, whereas reducing Bem1–Cdc24 decreased cluster area (Fig. EV2A,B). This suggests that the amount of positive-feedback proteins regulates cluster size, but that reducing individual concentrations experimentally may only have minor effects.

Next, we sought to experimentally alter the Cdc42-GTP cluster area. Consistent with our modeling results, we found that Cdc24[38A] cells exhibited higher maximum intensity of the Gic2PBD-tdTomato reporter for Cdc42-GTP and increased cluster area compared to WT cells (Fig. 2F), with similar results in Whi5-induced cells (Appendix Fig. S1B,C). However, we note that in addition to the concentration of Cdc42-GTP, the absolute expression level of the reporter may affect the cluster intensities,

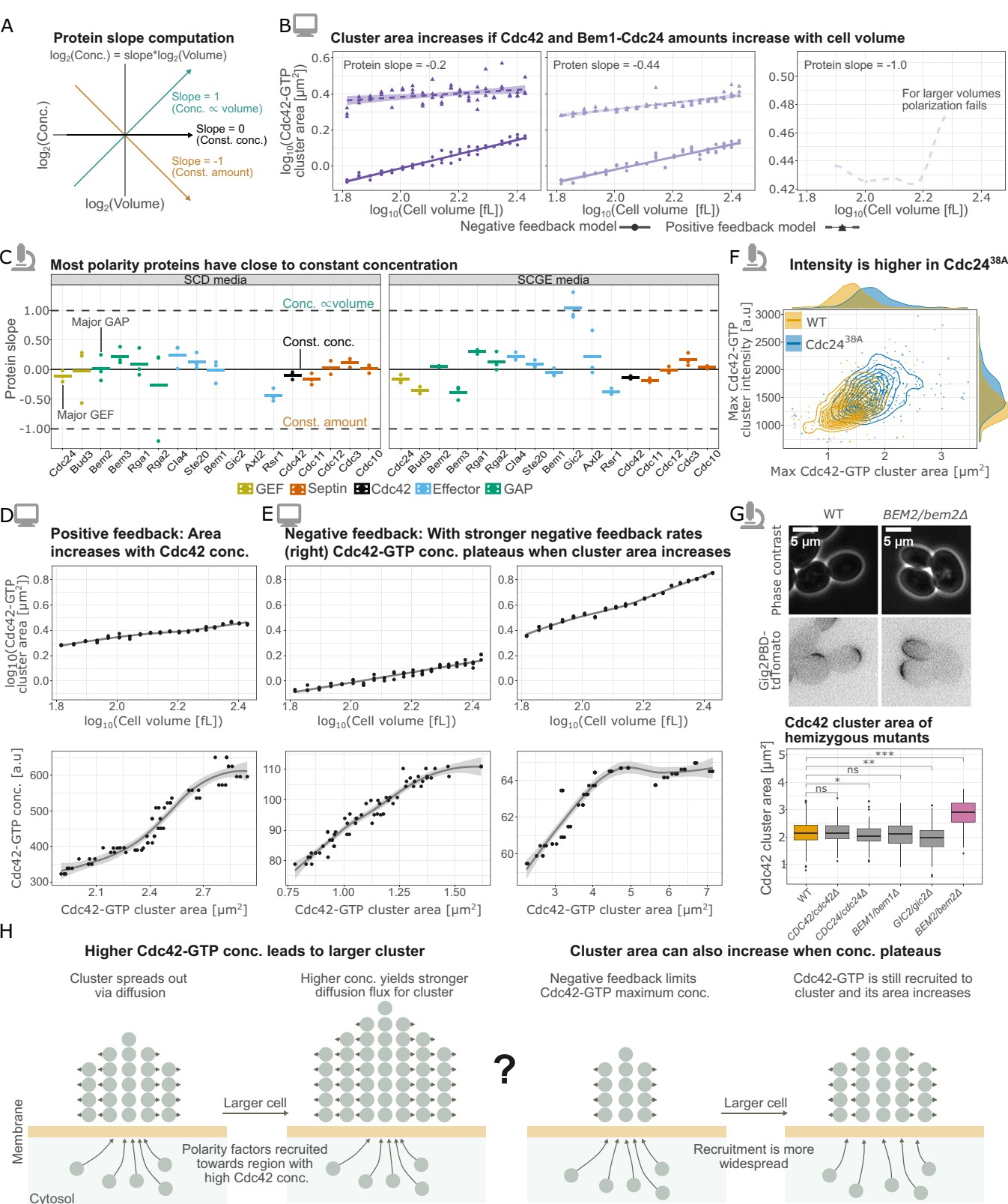

**A** Protein slope computation

**B** Cluster area increases if Cdc42 and Bem1-Cdc24 amounts increase with cell volume

**C** Most polarity proteins have close to constant concentration

**F** Intensity is higher in Cdc24³⁸ᴬ

**D** Positive feedback: Area increases with Cdc42 conc.

**E** Negative feedback: With stronger negative feedback rates (right) Cdc42-GTP conc. plateaus when cluster area increases

**G** 

Cdc42 cluster area of hemizygous mutants

**H** Higher Cdc42-GTP conc. leads to larger cluster — Cluster area can also increase when conc. plateaus

**Figure 2. The amount of key polarity proteins increases with cell volume to ensure robust polarization, while the interplay between positive and negative feedback determines the Cdc42-GTP concentration and cluster area.**

(A) Illustration of how concentration changes with cell volume for different protein slopes. (B) Modeling results for Cdc42-GTP cluster area measured at steady-state (after long simulation time) plotted against cell volume in a double logarithmic scale for the following conditions: Cdc42 and Bem1 complex amounts increase with cell volume (protein slope = −0.2 or −0.44), Cdc42 and Bem1 complex amounts constant (protein slope = −1). The two left panels show results from the negative and positive feedback models; only the positive feedback model is shown in the right panel, as it was the only model to polarize for this condition. In each case, $n = 60$ cells (three replicates per volume) with volumes ranging from 65 to 270 fL were simulated starting from random initial conditions. Lines in the left two panels show linear regression fits; the line in the right panel shows a loess-smoothing. (C) Analysis of experimental data from [Lanz et al, 2024] reveals that protein slopes for the majority of polarity proteins are close to 0, implying constant concentrations in SCD or SCGE media. Bars show the mean value of $n = 3$ biological replicates. Not all proteins we sought to analyze were included in the dataset (e.g., Gic2 for SCD media), a complete list of genes we aimed to analyze can be found in the Methods section. (D) Simulation results obtained with the positive feedback model: (top) Cdc42-GTP cluster area measured at steady-state (after long simulation time) plotted against cell volume in double logarithmic scale, (bottom) Cdc42-GTP concentration measured at the same time point as cluster area, plotted against the Cdc42-GTP cluster area. $n = 60$ (three replicates per volume) cells with volumes ranging from 65 to 270 fL were simulated starting from random initial conditions. Lines show loess smoothings. (E) Results for the negative feedback model are shown as in (D). The left plot corresponds to normal phosphorylation rates in the model; in the right plot, both the phosphorylation and dephosphorylation rates of the Bem1-Cdc24 are increased. As in (D), $n = 60$ cells, and the lines show loess smoothings. Note that the first panels in (E, D) are reproduced from Fig. 1D for comparison. (F) Experimental results for WT and Cdc24[38A] cells show maximum Cdc42-GTP intensity in the cluster at the time at which the cluster reaches its maximum area, plotted against the corresponding Cdc42-GTP cluster area. WT: $n = 463$ cells; Cdc24[38A]: $n = 478$ cells. (G) Experimental results for WT and hemizygous deletion mutants of key polarity proteins. (top) Representative images for WT and BEM2/bem2Δ cells. (bottom) Quantification of Cdc42-GTP cluster area: WT ($n = 198$), CDC42/cdc42Δ ($n = 52$), CDC24/cdc24Δ ($n = 131$), BEM1/bem1Δ ($n = 147$), GIC2/gic2Δ ($n = 100$), BEM2/bem2Δ ($n = 54$). The center line indicates the median; box limits show the 25th–75th percentiles (IQR); whiskers extend to the most extreme data points within 1.5× IQR; points represent outliers. Stars denote $p$ values *>0.05, **>0.0005, ***>5e-6 from Wilcoxon rank-sum test. (H) Illustration of how higher Cdc42-GTP concentration in the cluster can lead to an increase in its area (left), and how the cluster area can increase even when the Cdc42-GTP concentration plateaus in the presence of negative feedback (right). At least two independent replicates were performed for each experiment. Computer icon—modeling results. Microscope icon—experimental results. Source data are available online for this figure.

which makes it difficult to interpret intensity differences between strains. To further probe the regulation of cluster size, we reduced the availability of positive-feedback components by deleting one allele of individual polarity proteins in diploid strains (Fig. 2G; Appendix Fig. S2). In *BEM1*, *CDC24*, and *CDC42* hemizygous deletion mutants, the cluster area remained largely unchanged, consistent with our modeling predictions of redundancy. Interestingly, *GIC2* hemizygotes had slightly smaller clusters, indicating that protein abundance indeed regulates cluster area. We also experimentally tested reducing GAP concentration by deleting one copy of *BEM2* in diploid cells. In agreement with our modeling (Fig. EV2), this resulted in a noticeably larger Cdc42-GTP cluster (Fig. 2G).

Lastly, we used computational modeling to explore additional mechanisms that could drive cluster scaling. Interestingly, in the presence of negative feedback, the model predicts that Cdc42-GTP cluster area can increase with cell volume even without a corresponding increase in Cdc42-GTP concentration. Specifically, if both the phosphorylation and dephosphorylation rates of Bem1-Cdc24, which are associated with the negative feedback, are increased sufficiently, the maximum Cdc42-GTP concentration plateaus at larger cell volumes while the cluster area still increases (Fig. 2E right). In this scenario, the negative feedback prevents Cdc42-GTP from exceeding a threshold concentration, yet positive feedback continues to recruit polarity proteins to the cluster, enabling its growth (Fig. 2H, right).

## Deletion of polarisome complex components leads to increased septin ring diameter

After successfully modeling the scaling of the Cdc42-GTP cluster area as described above, we then set out to better understand the subsequent assembly of a septin ring. Similar to the Cdc42-GTP cluster area, the septin ring diameter increases with cell volume, which may indicate a causal link. Interestingly, despite an unchanged Cdc42 cluster area, *bni1Δ* cells show an increased

septin ring diameter compared to wild-type cells after accounting for cell volume (~28% increase) [Kukhtevich et al, 2020]. This highlights that the septin ring diameter is not solely determined by the Cdc42 cluster size. Prior to developing a computational model to account for this observation, we set out experimentally to better understand how formins and components of the polarisome complex contribute to septin ring formation.

First, we asked whether, similar to *bni1Δ*, also deletion of the second yeast formin Bnr1 [Park and Bi, 2007; Yu et al, 2011] or the additional polarisome components Spa2 and Bud6 [Liu et al, 2010; Xie et al, 2019] would lead to an increased septin ring diameter. We introduced the corresponding deletions into a strain carrying an mCitrine-tagged allele of the septin Cdc10 and performed microfluidics-based time-lapse experiments. We found that *spa2Δ* and *bud6Δ* cells partially phenocopy the effect of *bni1Δ* cells on cell geometry, cell volume and Cdc10 ring diameter, which is increased by ~13% compared to wild-type cells. By contrast, the septin ring diameter is largely unaffected in *bnr1Δ* cells (Fig. 3A–E). Based on this, we can conclude that polarisome components are more important for setting the septin ring diameter than the second formin Bnr1 and that Bni1 has a larger effect on the ring diameter than Spa2 and Bud6.

## Spatial confinement of exocytosis is critical for septin ring diameter control

The formin Bni1 is actively involved in F-actin cable assembly [Yu et al, 2011; Liu et al, 2010; Xie et al, 2019]. Therefore, we decided to focus next on the role of F-actin in setting the septin ring diameter. To analyze F-actin cables along with the septin ring, we tagged Cdc10 with mNeptune2.5, and Abp140, which has previously been used for actin cytoskeleton visualization [Buttery et al, 2007], with mCitrine. To test a wide range of cell volumes, we again used inducible Whi5. Microfluidics time-lapse experiments revealed that in *bni1Δ* cells, F-actin cables mostly do not polarize toward the bud side, while in WT cells, cable polarization is prominent (Fig. 4A).

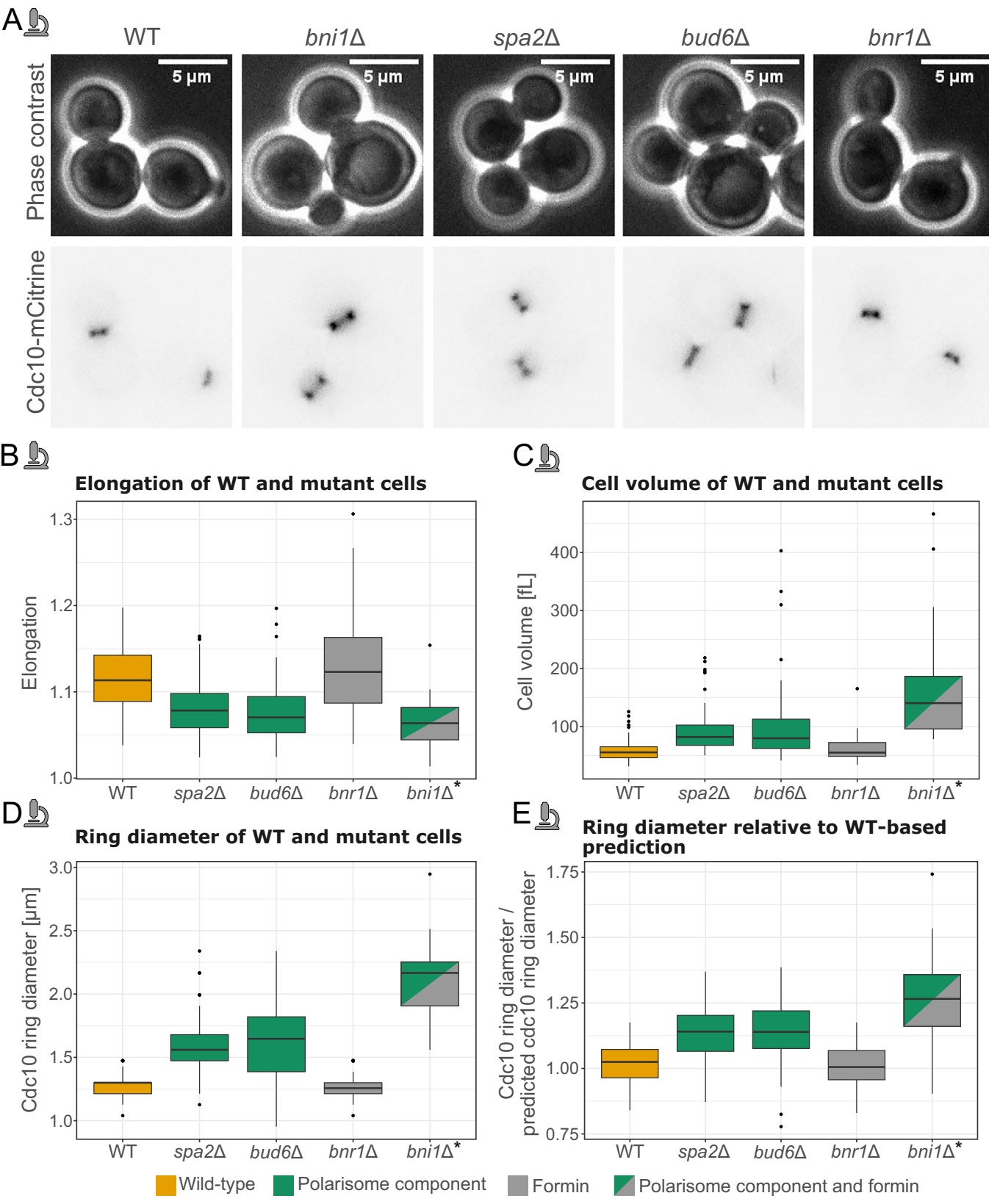

**Figure 3. Experimental analysis reveals the impact of formins and components of the polarisome complex on the septin ring diameter.**

(A) Representative microscopy images of budding yeast cells (phase contrast) and septin ring (Cdc10-mCitrine) for WT, *bni1Δ*, *spa2Δ*, *bud6Δ*, and *bnr1Δ* cells. (B) Elongation (major cell axis/minor cell axis) of WT (n = 81 cells), *spa2Δ* (n = 90 cells), *bud6Δ* (n = 83 cells), *bnr1Δ* (n = 67 cells), and *bni1Δ** (n = 35 cells). (C) Mother cell volume for the same cells as in (B). (D) Median septin ring diameter during its presence for the same cells as in (B). (E) The predicted septin ring diameter for the same cells as in panel b was calculated by dividing the septin ring diameter of a given cell by a model prediction based on the WT data in [Kukhtevich et al, 2020]. For boxplots, the center line indicates the median; box limits show the 25th–75th percentiles (IQR); whiskers extend to the most extreme data points within 1.5× IQR; points represent outliers. Note that the WT dataset shown here is not the same as the data the model prediction is based on. (B–E) * data from [Kukhtevich et al, 2020]. At least two independent replicates were performed for each experiment. Microscope icon—experimental results. Source data are available online for this figure.

Our analysis showed that in WT cells, the Abp140 cluster area scales with cell volume (Fig. 4B). We also observed a clear correlation between the Cdc10 ring diameter and the Abp140 cluster area (Fig. 4C). By contrast, in *bni1Δ* cells, only ~11% of cells show clearly detectable Abp140 clusters. This suggests that F-actin cables play an important role in setting the septin ring diameter. We also obtained similar results using phalloidin staining as an orthogonal approach (Fig. EV3).

We next asked why correct F-actin cable assembly and polarization toward the bud site are important for controlling the septin ring diameter. Okada et al, [Okada et al, 2013] previously suggested that the septin ring is sculpted by polarized exocytosis, which displaces the accumulating septins at the center of the bud site and thereby relieves inhibition of Cdc42 which is mediated via septin-recruited GAPs. Since cargo vesicles are delivered along F-actin cables, actin has a direct impact on exocytosis [Park and Bi, 2007; Liu et al, 2010]. Thus, we decided to investigate if the septin ring diameter depends on exocytosis and how this dependency is affected by deleting *BNI1*. We constructed new strains in which we tagged Exo84, one of the exocyst complex subunits [Jose et al, 2015], with mCitrine, and Cdc10 with mNeptune2.5. Again, we used inducible Whi5 to increase the range of cell volumes and performed time-lapse experiments, where we quantified the Exo84 cluster diameter together with the Cdc10 ring diameter. We found that the Exo84 cluster diameter scales linearly with cell volume on a double logarithmic scale. In addition, in *bni1Δ* cells, the Exo84 cluster diameter is larger compared to WT cells (Fig. 4D,E). Moreover, the Cdc10 ring diameter is larger in *bni1Δ* cells and scales with the Exo84 cluster diameter in both WT and *bni1Δ* cells (Fig. 4F,G). Both strains appear to follow the same scaling relationship, indicating a mechanistic link between exocytosis and septin ring diameter.

These experimental results suggest that in *bni1Δ* cells, perturbed F-actin assembly and polarization lead to diffused, i.e., less focused, exocytosis, which in turn leads to an enlarged septin ring (Fig. 4H).

## Exocytosis-aided recruitment of septins can explain the increase in septin ring size upon diffused exocytosis

Our experimental results suggest that in *bni1Δ* cells, exocytosis at the bud site is more diffused than in WT cells (Fig. 4). Given that exocytosis displaces proteins from the polarity site in *S. pombe* [Gerganova et al, 2021] and that it has been suggested to sculpt the septin ring in *S. cerevisiae* by displacing septins [Okada et al, 2013], we hypothesized that diffused exocytosis causes a larger ring diameter by septin displacement (Fig. 4H). To test this, we turned to computational modeling.

To investigate the role of diffused exocytosis, we developed a three-dimensional mechanistic model of septin ring assembly,

which we refer to as the septin binding and exocytosis (SBE) model (Fig. EV4A,B). In contrast to a model where exocytosis is the sole mechanism driving ring formation, as in [Okada et al, 2013], our SBE model is robust to cell volume changes (see Appendix). Since septin recruitment is likely cell-cycle triggered [Lai et al, 2018], and to reduce simulation time, we decided to temporally separate Cdc42 polarization and septin ring formation in our SBE model. The SBE model, therefore, consists of a Cdc42 polarization module and a septin ring module (Fig. EV4). In the polarization module, we used the positive feedback model. As we demonstrated above, positive feedback effectively replicates the scaling of the Cdc42-GTP cluster area with cell volume (Figs. 1 and EV1). Additionally, positive feedback models have successfully captured other behaviors, such as competition between polarity sites [Goryachev and Pokhilko, 2008]. Moreover, this allows us to reduce simulation runtime for the analysis of septin ring formation (one simulation takes >100 h), as the positive feedback model is faster to simulate than the negative feedback model. For the septin module, to capture that septin is recruited by and interacts with polarity factors such as Gic1/2, Axl2, Cdc24 and potentially Cdc42 itself [Iwase et al, 2006; Sadian et al, 2013; Chollet et al, 2020; Kang et al, 2024], we included the key player Axl2 as a representative for all recruitment factors. Furthermore, as Cdc24 binds Cdc11 in the Cdc42-GTP cluster but not in the septin ring [Chollet et al, 2020], and Cdc42 inhibits septin polymerization in vitro [Sadian et al, 2013], we model that septin binds to polarity factors, represented by Axl2, and that septin cannot polymerize when bound. This drives septin polymerization and subsequently ring formation at the cluster periphery. As a result, the septin ring can form even in the absence of exocytosis (Fig. EV4F), in line with observations that ring formation can start without a clear Sec4 (exocytosis marker) signal [Lai et al, 2018]. Furthermore, exocytosis was modeled to be directed towards the Cdc42 cluster, where, for the meshed model geometry, vesicles could hit a tunable percentage of the mesh nodes, and when occurring, exocytosis displaces proteins. Lastly, following earlier observations [Okada et al, 2013], septin recruits GAP proteins (Fig. EV4A).

To simulate the septin ring module, we initiated the septin module of the SBE model simulations from a Cdc42-GTP cluster obtained with the Cdc42 polarization module (Fig. EV4). Given this approach, the SBE model can form a septin ring, but when we modeled diffused exocytosis by allowing exocytosis to hit a larger number of mesh nodes, we did not observe a larger ring (Fig. EV4E). Similar results also hold for simpler particle models (Appendix Fig. S3).

The SBE model (Fig. EV4) and the particle models (Appendix Fig. S3) thus cannot explain the septin ring enlargement with diffused exocytosis, which we observed experimentally. Interestingly, when exocytosis is delayed via a conditional *BNI1* allele,

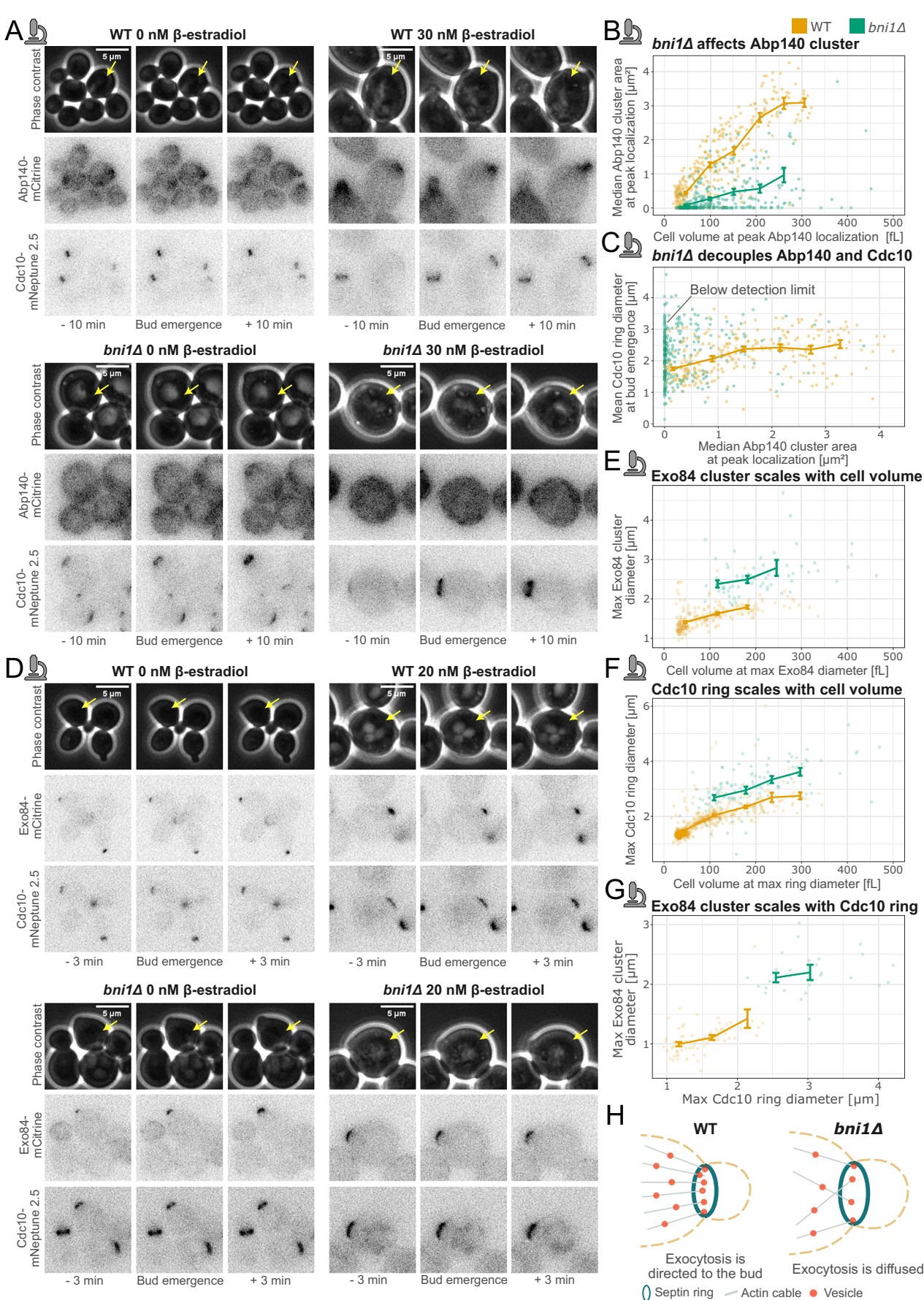

**Figure 4. Experimental analysis shows that exocytosis is diffused in *bni1Δ* cells, likely due to disturbed F-actin cable assembly and polarization toward the bud site.**

(A) Representative microscopy images of budding yeast cells (phase contrast), F-actin (Abp140-mCitrine) and septin ring (Cdc10-mNeptune2.5) for WT and *bni1Δ* cells. Different cell sizes are obtained by different β-estradiol concentrations (0 or 30 nM) to induce *WHI5* expression (see Methods section). (B) Abp140 cluster area measurements from microscopy images plotted against corresponding mother cell volume in a double logarithmic scale for WT ($n = 324$ cells) and *bni1Δ* ($n = 319$ cells). (C) Cdc10 ring diameter at bud emergence plotted against Abp140 cluster area for WT ($n = 290$ cells) and *bni1Δ* ($n = 298$ cells). (D) Representative microscopy images of cells (phase contrast), exocytosis (Exo84-mCitrine) and septin ring (Cdc10-mNeptune2.5) for WT and *bni1Δ* cells. Different cell sizes are obtained by different β-estradiol concentrations (0 or 20 nM) to induce *WHI5* expression. (E) Exo84 cluster diameter measurements from microscopy images plotted against corresponding mother cell volume in a double logarithmic scale for WT ($n = 218$ cells) and *bni1Δ* ($n = 62$ cells). (F) Maximum Cdc10 ring diameter plotted against corresponding mother cell volume in a double logarithmic scale for WT ($n = 556$ cells) and *bni1Δ* ($n = 113$ cells). (G) Exo84 cluster diameter measurements from microscopy images plotted against maximum Cdc10 ring diameter for WT ($n = 68$ cells) and *bni1Δ* ($n = 28$ cells). (A, D) Annotation of bud emergence corresponds to the cells highlighted with yellow arrows. (B, C, E-G) Solid lines show binned means, and error bars show standard error centered at the binned mean. (H) Suggested mechanism for septin ring enlargement in *bni1Δ* cells. At least two independent replicates were performed for each experiment. Microscope icon—experimental results. Source data are available online for this figure.

septin recruitment is reduced [Lai et al, 2018], suggesting that exocytosis aids in septin recruitment. To account for this, we expanded the SBE model to create the *Septin Binding and Exocytosis Recruitment* (SBER) model (Fig. 5A,B). To keep the SBER model simple, and as we do not know which protein might aid in vesicle-supported recruitment, we modeled that exocytosis delivered an unknown species (referred to as *X*), which recruits septin (Fig. 5A).

Our SBER model proposes that exocytosis facilitates septin recruitment, so disrupting secretory vesicle fusion experimentally should affect septin ring assembly. To test this, we measured septin ring and Exo84 cluster diameters in *sec10-2* temperature-sensitive mutant cells [Stalder and Novick, 2015], in which secretory vesicles primarily accumulate rather than fuse at the bud site. Interestingly, after shifting to the non-permissive temperature, the *sec10-2* mutant exhibited a noticeably larger septin ring diameter than wild-type cells, while the Exo84 cluster diameter was only slightly increased (Fig. EV5A–C). Consistent with our experiments, mimicking impaired vesicle fusion in the computational SBER model by decreasing the exocytosis rate also resulted in larger septin rings (Fig. EV5D). Furthermore, consistent with the literature [Okada et al, 2013], when simulating ring formation in the SBER model, the Cdc42-GTP concentration initially decreases upon septin recruitment and then increases after a stable ring has formed (Fig. 5C,D). In earlier computational models [Okada et al, 2013], such behavior was partially achieved by having septin create a diffusion barrier. However, the existence of such a barrier is debated [Sugiyama and Tanaka, 2019]. In the SBER model, a similar effect is obtained without a diffusion barrier, as the septin-recruited GAPs that associate with the septin ring convert any Cdc42-GTP that diffuses into the ring to Cdc42-GDP. This effectively concentrates Cdc42-GTPs inside the ring.

Validation of the SBER model against experimental results enabled us to investigate the effects of diffused exocytosis on septin ring formation. We modeled exocytosis to be targeted towards the Cdc42 cluster, where vesicles could hit between 0.034% and 1.5% of the mesh nodes. Simulating 10 cells for each of eight conditions and using our custom algorithm for ring diameter quantification (Fig. EV4D), we found that the septin ring diameter gradually increases with more diffused exocytosis up to around 1% of nodes being hit (Fig. 5E). This suggests that stronger polarisome perturbations can lead to larger ring diameters. To ensure the robustness of our modeling predictions, we conducted an extensive sensitivity analysis of the SBER septin module by varying each parameter individually by factors of 0.5 and 2.0. For 23 out of 26

parameter sets, the model consistently predicted an enlarged septin ring in response to diffused exocytosis (Appendix Fig. S4).

Since the SBER model accounts for the increase in septin ring diameter experimentally observed in *bni1Δ* cells, we next tested whether it could also capture the increase in septin ring diameter with cell volume. For three cell volumes (33, 65, 133 fL) and three levels of diffused exocytosis, we simulated 10 cells each. In the computational model, the septin ring diameter increased with both cell volume and the level of diffused exocytosis (Fig. 5F). Additionally, cluster area and ring diameter were positively correlated (Fig. 5G), in line with an increase in the Cdc42-cluster area driving the increase in the septin ring diameter.

Taken together, the SBER model provides a three-dimensional mechanistic framework for septin ring assembly that captures key experimental observations. Using this model, we show that the increase in Cdc42 cluster area with cell volume can explain the scaling of septin ring diameter. Further, modeling demonstrates that if exocytosis aids in septin recruitment, diffused exocytosis can decouple the septin ring diameter from the Cdc42 cluster area.

## Cdc24 interaction with septins is important for the timing of septin ring formation, but not its final size

Our SBER model suggests that diffused exocytosis accounts for the increased septin ring diameter observed in *bni1Δ* cells because exocytosis supports septin recruitment. We next asked what would be the effect of weakening the interaction between polarity proteins and septins? To this end, we further investigated with time-lapse imaging a Cdc24 mutant, Cdc24$_{kk}$, which was reported to perturb the interaction between Cdc24 and septins [Chollet et al, 2020]. Both WT and Cdc24$_{kk}$ mutant cells showed similar Cdc11 ring diameters and scaling of the ring diameter with cell volume (Fig. 5H,I). Next, we checked if there were any differences in the dynamics of septin ring formation. Interestingly, we found that in Cdc24$_{kk}$ cells, the Cdc11 ring diameter reaches its maximum faster after bud emergence than in wild-type cells (Fig. 5J).

To further assess the predictive power of the SBER model, we tested its ability to capture perturbations beyond the *bni1Δ* phenotype. Specifically, we examined whether the model could explain the phenotype of the Cdc24$_{kk}$ mutant, which we modeled by reducing the Septin Polarity-factors binding rate (SPR). This adjustment weakens the interaction between septins and polarity proteins in the SBER model, where Axl2 represents septin-interacting polarity factors, including Cdc24. To explore the effects of this perturbation, we simulated ten cells (Fig. 5K,L). Since in the

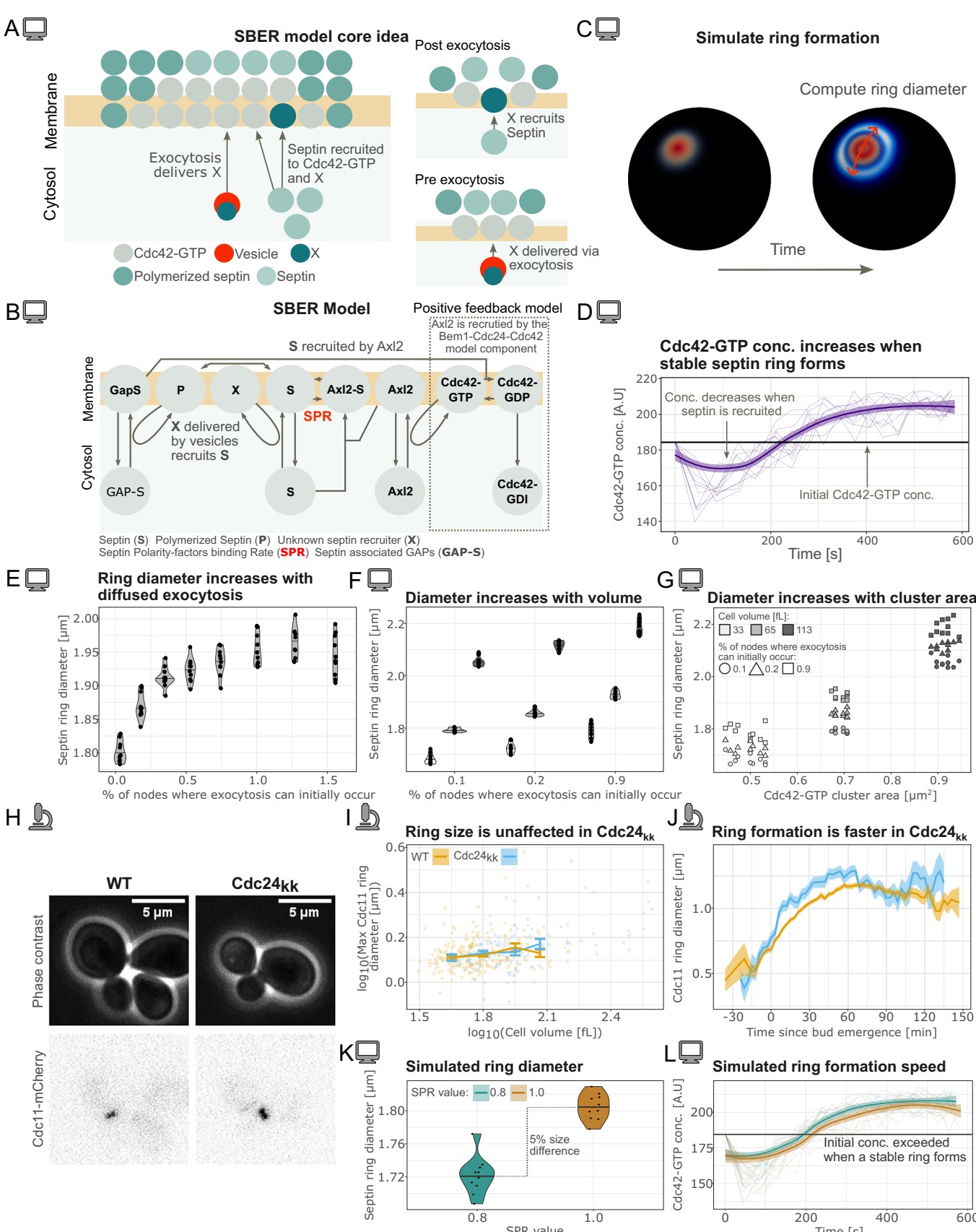

**Figure 5. The septin binding and exocytosis recruitment (SBER) model explains how diffused exocytosis can produce a larger septin ring, and reveals the role of Cdc24 interaction with septins in septin ring formation.**

(A) Core concept of the SBER model. Septin represents the septin monomers (Cdc10, Cdc11, etc.), which polymerize by binding to other septin monomers or to polymerized septins. All model reactions can be found in the Methods section. Briefly, septin is recruited towards the Cdc42-GTP cluster where it binds to polarity factors (see also (B) for more details), which prevents polymerization in the cluster center. This, combined with exocytosis displacing septin, promotes polymerization and ring formation at the cluster periphery. (B) Schematic of the SBER model, where the septin polarity-factors binding rate (SPR) is highlighted. The schematic focuses on the septin module, therefore, the Cdc42 polarity module is simplified on this schematic, for example, the Bem1-Cdc24-Cdc42-GTP complex that recruits Axl2 is omitted. (C) A representative simulation example shows the Cdc42 polarization (red) and consecutive septin ring formation (blue). (D) Cdc42-GTP concentration for the SBER model plotted over time for $n = 10$ simulations each initiated from the same Cdc42-GTP cluster with a loess-smoothing. (E) Septin ring diameter for the SBER model plotted against the % of nodes that can initially be hit by exocytosis. The number of nodes that can be hit are those where the concentration of Cdc42 fulfils: *Cdc42-GTP > ε\*max(Cdc42-GTP)*, where a smaller ε corresponds to more diffused exocytosis. For each condition, $n = 10$ simulations, all starting from the same Cdc42-GTP cluster, were performed. In each case, the model was simulated for a long time to reach a stable ring, and then the septin ring diameter was measured. (F) Septin ring diameter for the SBER model for three cell volumes and three different levels of diffused exocytosis. $n = 10$ for each condition. (G) Septin ring diameter plotted against corresponding Cdc42-GTP cluster area for the simulations in (F). (H) Representative microscopy images of cells (phase contrast) and septin ring (Cdc11-mCherry) for WT and Cdc24$_{kk}$ cells. (I) Experimental Cdc11 ring diameter measurements from microscopy images plotted against mother cell volume in a double logarithmic scale for WT ($n = 271$ cells) and Cdc24$_{kk}$ ($n = 124$ cells). Solid lines show binned means, and error bars show standard error centered at the binned mean. (J) Mean Cdc11 ring diameter measurements from microscopy images plotted over the cell cycle for WT ($n = 167$ cells) and Cdc24$_{kk}$ ($n = 73$ cells). Single-cell traces are aligned at bud emergence. The ribbon shows the standard error. (K) The final ring diameter is computed as in (C) for two different SPR values, at the final time point in (I) (when the model has been simulated to a stable ring). For a larger SPR value, the ring diameter is ~5% larger. For each SPR value, $n = 10$ simulations were performed. (L) Cdc42-GTP concentration over time for the same simulations as in (K). The thin lines correspond to individual simulations, and the thicker line to a loess-smoothing. Two independent replicates were performed for the experiments. Computer icon—modeling results. Microscope icon—experimental results. Source data are available online for this figure.

model, the Cdc42-GTP concentration initially decreases and, after a stable ring is formed, increases (Fig. 5D), we decided to use the time it takes for Cdc42-GTP to exceed its initial concentration as a metric for ring formation speed. Consistent with our experimental observations, we found that the Cdc42-GTP concentration increases faster for lower SPR values (Fig. 5L). This acceleration is likely due to the diminished inhibitory influence of polarity factors, such as Cdc24, on septin polymerization when the SPR value is reduced, thereby facilitating faster ring formation. Moreover, in the model, the septin ring diameter is also slightly smaller for lower SPR values (Fig. 5K). However, given the measurement error, such a small effect would be difficult to detect in experiments.

## Disruption of negative feedback leads to a larger Cdc42-GTP cluster area but not septin ring diameter

Earlier in this study, we showed that in Cdc24[38A] cells (non-phosphorylatable negative feedback mutant), the Cdc42 cluster area is larger compared to WT cells (Fig. 1G; Appendix Fig. S1). We next addressed the role of the negative feedback in the regulation of the septin ring diameter. For this, we constructed new diploid strains in which one allele of Cdc10 was tagged with mCherry in WT and Cdc24[38A] cells. We then performed time-lapse experiments (Fig. 6A). Surprisingly, despite Cdc24[38A] cells having a significantly larger Cdc42-GTP cluster, we did not find any major differences in the septin ring diameter at any point in the cell cycle (Fig. 6B; Appendix Fig. S5). To investigate whether this phenomenon extends to other methods of biasing the polarization network toward Cdc42-GTP production, we measured septin ring diameter in *BEM2* hemizygous deletion mutant cells, which, as we have shown, exhibit substantially larger Cdc42-GTP clusters (Fig. 2E). Strikingly, the septin ring diameter remained largely unchanged compared to wild-type cells (Fig. 6C,D; Appendix Fig. S2A,C), revealing a decoupling between Cdc42-GTP cluster area and septin ring diameter.

To better understand this decoupling, we returned to our computational SBER model. Since limiting GAP availability is one way to explain this effect, we simulated reduced GAP activity within the polarization module of the SBER model. We simulated

ten cells (Fig. 6E,F), and for ease of interpretation, compared the Cdc42-GTP cluster diameter, rather than area, with the septin ring diameter. The increase in ring diameter (fold change of 1.14 for PSR = 0.2, Fig. 6F) was close to that in cluster diameter (1.17). By contrast, our experimental results showed that the septin ring diameter was almost intact (fold change of 1.04 for the maximum ring diameter), even though the maximum Cdc42-GTP cluster diameter (estimated as the square root of cluster area) increased 1.21-fold in Cdc24[38A] averaged across cell volumes. To investigate if this discrepancy between experimental observations and the model could be explained by values chosen for the septin-related model parameters, we altered the activity of septin-recruited GAPs, and septin concentration, but did not find any noticeable effect. However, when we increased the polarity-factors septin recruitment rate (PSR parameter Fig. 6E), the increase in ring diameter compared to the previous simulations was smaller (fold change of 1.10 for PSR = 0.5, Fig. 6F). This suggests that stronger recruitment of septins by polarity proteins can lead to a partial decoupling of Cdc42-GTP cluster diameter and ring diameter.

In summary, the Cdc42-GTP cluster area is increased in Cdc24[38A] and *BEM2* hemizygous deletion mutant cells, while the Cdc10 ring diameter is largely the same. Thus, biasing the polarization network towards Cdc42-GTP production leads to a decoupling of cluster area and Cdc10 ring diameter.

## Discussion

The delicate interplay of positive and negative feedback enables yeast cells to cluster Cdc42-GTP at the presumptive bud site [Chiou et al, 2017], with an area that increases with cell volume [Kukhtevich et al, 2020]. Here, we showed that positive feedback, together with the volume-dependent increase of the amount of polarity proteins, is sufficient to explain this cluster scaling behavior (Figs. 1 and 2A–C). Accordingly, in Cdc24[38A] cells with disrupted negative feedback [Kuo et al, 2014], the Cdc42-GTP cluster area scales with cell volume (Fig. 1G). Moreover, limiting

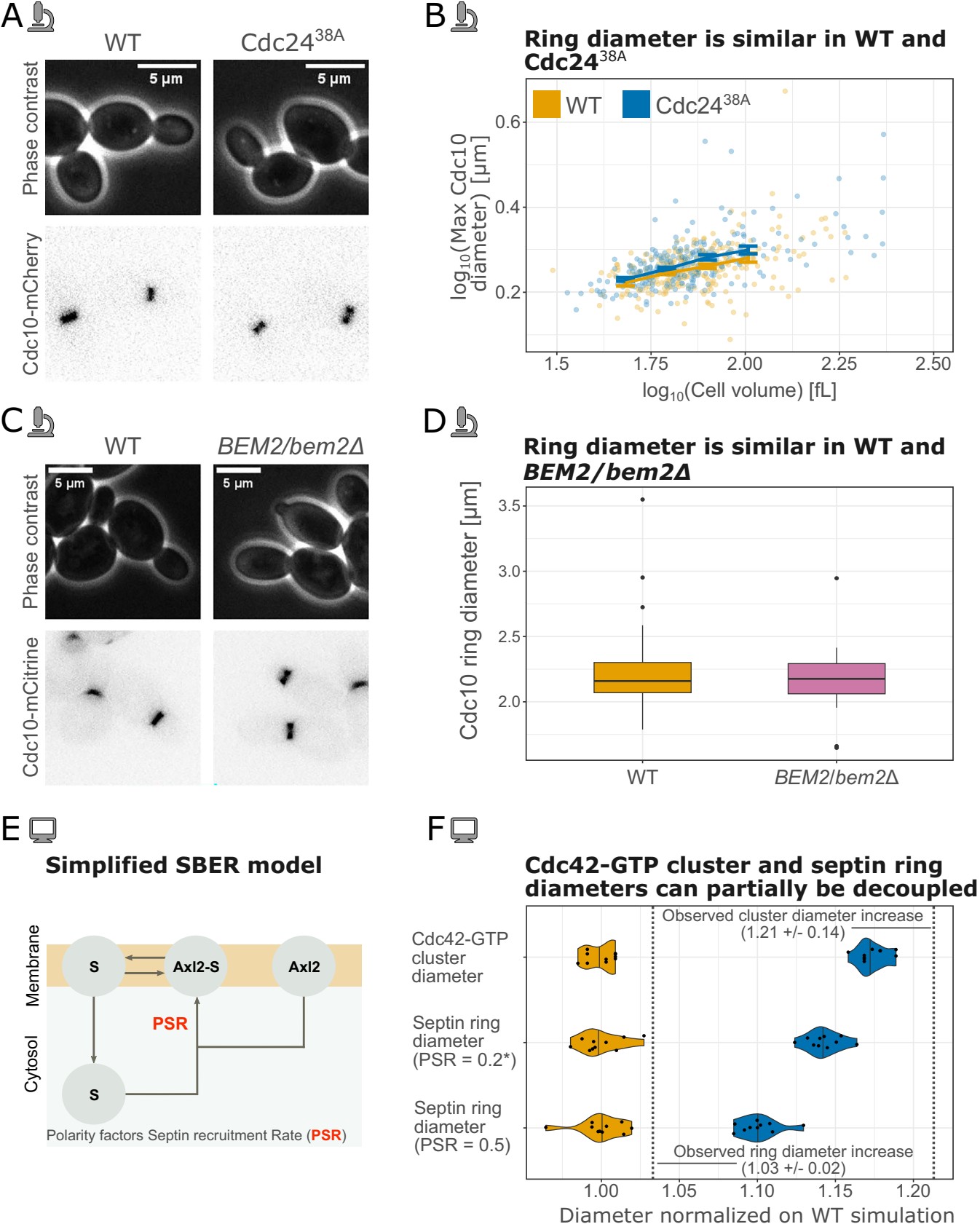

◄

**Figure 6.   Septin ring diameter is decoupled from the Cdc42-GTP cluster area in Cdc24³⁸ᴬ cells.**

(A) Representative microscopy images of cells (phase contrast) and septin ring (Cdc10-mCherry) for WT and Cdc24³⁸ᴬ cells. (B) Maximum Cdc10 ring diameter measurements from microscopy images plotted against corresponding mother cell volume in a double logarithmic scale for WT ($n = 237$ cells) and Cdc24³⁸ᴬ ($n = 240$ cells). Solid lines show binned means, and error bars show standard error centered at the binned mean. (C) Representative microscopy images for WT and *BEM2/bem2Δ* cells. (D) Quantification of septin ring diameter: WT ($n = 136$), *BEM2/bem2Δ* ($n = 43$). The center line indicates the median; box limits show the 25th–75th percentiles (IQR); whiskers extend to the most extreme data points within 1.5× IQR; points represent outliers. (E) Illustration of the PSR rate (Simplified version of Fig. 5B). (F) Cdc42-GTP cluster diameter and septin ring diameter for SBER model simulations of Cdc24³⁸ᴬ and WT, normalized on WT. Star denotes the default value (0.2) for the polarity factors, septin recruitment rate (PSR). The vertical dashed lines correspond to the increase in Cdc42 cluster and septin ring diameters observed experimentally (averaged across cell volume bins). For each condition, $n = 10$ cells were simulated, and the model was simulated for a long time to reach a stable cluster area and septin ring. Two independent replicates were performed for the experiments. Computer icon—modeling results. Microscope icon—experimental results. Source data are available online for this figure.

Gic2 amounts leads to a slightly smaller cluster (Fig. 2G), which is consistent with the prediction from our computational modeling that redundancy among polarity components results in only minor effects on the cluster area from limiting a single polarity protein. We also found that, consistent with previous models [Hubatsch et al, 2019], boosting Cdc42-GTP production, achieved here by disrupting the negative feedback or limiting GAP availability, produces a noticeably larger cluster. Furthermore, using computational modeling, we identified two distinct mechanisms by which the interplay of feedbacks can drive an increase in the size of self-assembled structures with cell volume. In a model with only positive feedback, we found that Cdc42-GTP cluster concentration increases with cell size, leading to cluster growth due to diffusion-driven flux. In the presence of strong negative feedback, however, Cdc42-GTP concentration can plateau while the cluster area continues to increase with cell volume as the positive feedback recruits polarity proteins (Fig. 2D,E,H). To determine which scenario is at play in budding yeast, quantification of the Cdc42-GTP concentration will be required; however, the commonly used approach for visualizing Cdc42-GTP species via tagged Gic2 does not provide sufficient accuracy.

As part of our analysis, we found that yeast cells maintain polarity protein concentrations close to constant [Lanz et al, 2024] (Fig. 2C). This may serve to facilitate the formation of larger Cdc42-GTP clusters necessary for proper cell division, as nuclear size increases with cell volume [Jorgensen et al, 2007]. Alternatively, and perhaps more likely, cluster scaling could be a side effect of maintaining robust Cdc42 polarization. In our model, simulations with constant amounts of Cdc42, the Bem1 complex, and constant GAP concentration, polarization fails (Fig. 2B). Overexpression of both Cdc42 and GEFs is fatal [Howel et al, 2012], and deletion of GAP proteins reduces the replicative lifespan of yeast cells [Meitinger et al, 2014; Kang et al, 2022]. Altogether, this implies that robust polarization requires constant concentrations of polarity proteins. Interestingly, in previous theoretical work, increased polarity protein concentrations also cause larger cells to form multiple stable clusters [Borgqvist et al, 2021; Chiou et al, 2021], aligning with observations that in larger *rsr1Δ* cells (cells with random budding), multiple initial Cdc42 clusters form more frequently [Chiou et al, 2021]. However, with several initial clusters, bud-site selection may become less precise, as observed in older cells [Yang et al, 2022]. This could be detrimental during aging, as *rsr1Δ* cells with imprecise budding have a shorter replicative lifespan.

Following Cdc42-GTP cluster formation, septin ring formation initiates with septin recruitment to the bud site [Park and

Bi, 2007]. Prior work revealed that deleting the formin and polarisome component Bni1 leads to an increased septin ring diameter despite an unchanged maximum Cdc42-GTP cluster area [Kukhtevich et al, 2020]. Given Bni1's role as a formin involved in the assembly of F-actin cables [Park and Bi, 2007], and the suggestion by Okada et al [Okada et al, 2013] that actin-dependent exocytosis regulates septin ring formation, we visualized F-actin and the exocyst subunit Exo84 [Jose et al, 2015] in both WT and *bni1Δ* cells. We found that *bni1Δ* cells show diffused exocytosis, likely due to disturbed F-actin polarization toward the bud site (Fig. 4).

To investigate the effect of diffused exocytosis on the septin ring, we developed the SBER model, a three-dimensional mechanistic description of septin ring assembly (Fig. 5). In this model, exocytosis promotes ring formation. Additionally, consistent with the observation that Cdc42 inhibits septin polymerization in vitro and that ring formation can start without a clear exocytosis signal [Sadian, 2013; Lai et al 2018], polarity proteins bind to septin and inhibit its polymerization. Together, these two mechanisms make ring formation robust with respect to model parameters such as cell volume. Using this model, we showed that diffused exocytosis is sufficient to decouple septin ring diameter from Cdc42 cluster area across different cell volumes if exocytosis aids in septin recruitment, as suggested by experiments [Lai et al, 2018] (Fig. 5A,G). Overall, our SBER model captures key experimental observations and shows that multiple mechanisms acting together make septin ring assembly a robust process, just as multiple mechanisms acting together make Cdc42 polarization a remarkably robust process [Brauns et al 2023].

Lastly, we explored the effect on septin ring diameter when disrupting the Cdc42-GTP to Cdc42-GDP conversion by either disturbing the negative Cdc42-GTP-centered feedback in Cdc24³⁸ᴬ cells, or by limiting GAP availability—both of which lead to increased Cdc42-GTP cluster area. Surprisingly, we observed a decoupling between the Cdc42-GTP cluster area and the septin ring diameter; however, in the opposite direction compared to *bni1Δ* cells [Kukhtevich et al, 2020]. In both cases, the septin ring diameter remains largely unaffected despite the increase in Cdc42-GTP cluster area (Fig. 6). To understand why, we modified the SBER model to produce a larger Cdc42 cluster by reducing GAP activity In contrast to the experimental results, in the model, the septin ring diameter increased almost as strongly as the Cdc42-GTP cluster diameter (Fig. 6F). However, when we increased the recruitment rate of septin via polarity factors in the SBER model, the septin ring diameter increased less than the Cdc42-GTP cluster diameter. Inspecting concentrations in the SBER model, we found

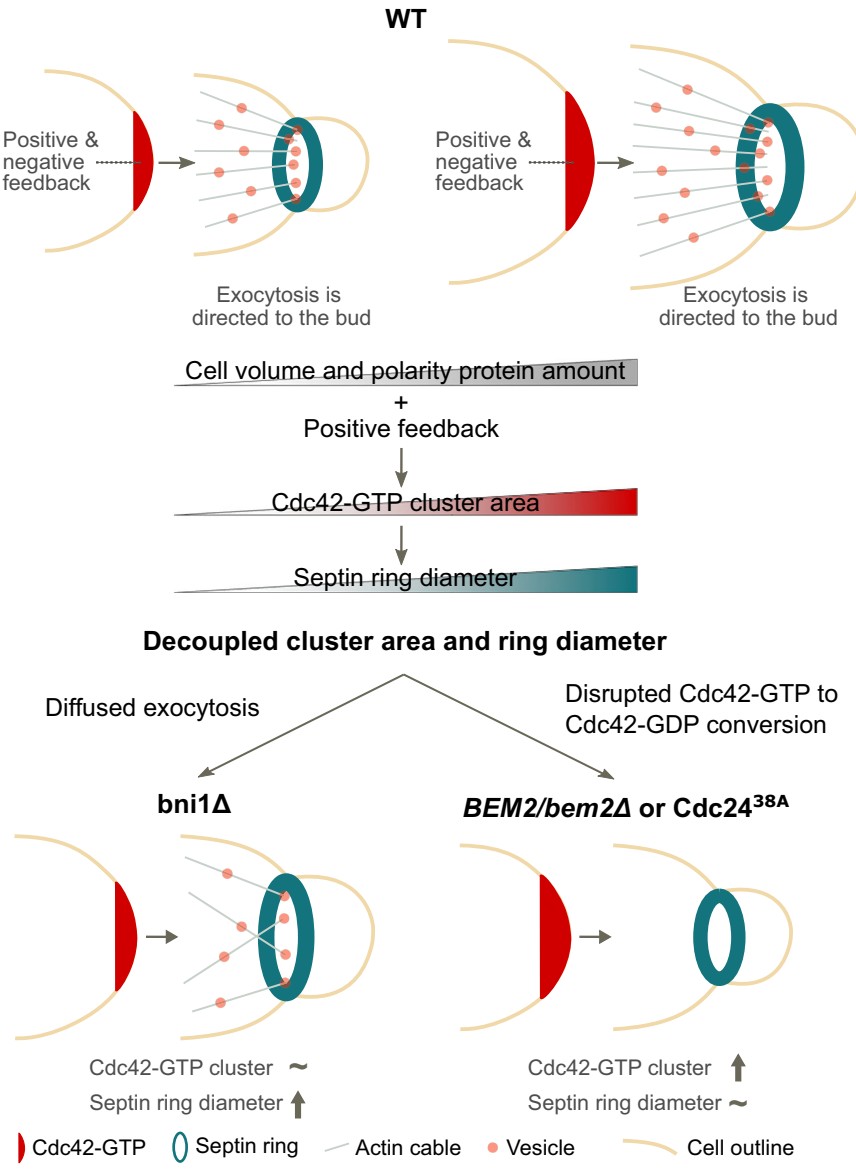

**Figure 7. Illustration summarizing the mechanisms determining Cdc42-GTP cluster area and septin ring diameter.**

Positive feedback, together with polarity proteins increasing in amount as cell size increases, leads to an increase in Cdc42-GTP cluster area with cell volume. As a consequence, the septin ring diameter increases. The coupling of the Cdc42-GTP cluster area and septin ring diameter can be disrupted by more diffuse exocytosis or by a disrupted negative feedback in the Cdc42 polarization pathway.

that stronger septin recruitment depletes the pool of polarity factors that inhibit septin polymerization. Thus, more septin can polymerize close to the cluster center, producing a more concentrated ring that recruits more septin-associated GAPs. This increased GAP activity shrinks the Cdc42-GTP cluster upon ring formation (Fig. 5C), and since Cdc42-GTP–associated proteins recruit septins, this cascade produces a smaller ring than could have been expected given an enlarged Cdc42 cluster, consistent with experimental observations. Still, the model's predicted decoupling is weaker than observed experimentally, suggesting additional mechanisms.

In summary, we identified critical processes governing septin ring formation and size regulation in budding yeast by integrating

a novel experiment-consistent mechanistic model of septin ring assembly with experimental data (Fig. 7). In particular, we found that positive feedback-based Cdc42-GTP cluster formation together with the amount of polarity proteins increasing with cell size can explain the scaling of the septin ring diameter with cell volume. While this highlights the link between the Cdc42-GTP cluster and septin ring size, we also found that tuning specific parameters of the complex self-assembly process can lead to independent changes of both the Cdc42-GTP cluster area and septin ring diameter. Future investigations will be essential to determine whether cells exploit such regulatory mechanisms to dynamically control septin ring formation in response to internal or external signals.

# Methods

## Reagents and tools table

| Reagent/resource | Reference or source | Identifier or catalog number |
| --- | --- | --- |
| Experimental models: organism, strain name, and genotype | | |
| *S. cerevisiae* - DLY16570: Background YEF473; *Mat a/α; rsr1::TRP1/rsr1::TRP1 BEM1-GFP:LEU2/BEM1-GFP:LEU2 GIC2(1-208)-tdTomato:KAN^R/GIC2* | Kuo et al, 2014 Current Biology | DLY16570 |
| *S. cerevisiae* - DLY16571: Background YEF473; *Mat a/α; CDC24^38A/CDC24^38A rsr1::TRP1/rsr1::TRP1 BEM1-GFP:LEU2/BEM1-GFP:LEU2 GIC2(1-208aa)-tdTomato:KAN^R/GIC2* | Kuo et al, 2014, Current Biology | DLY16571 |
| *S. cerevisiae* - KSY195-1: Background W303; *Mat α; ADE2 cdc10::CDC10-mCitrine-ADH1term-HIS3* | Kukhtevich et al, Nature Communications, 2020 | KSY195-1 |
| *S. cerevisiae* - KSY260-2: Background W303; *Mat α; ADE2 cdc10::CDC10-mCitrine-ADH1term-HIS3 bni1Δ::KlacURA3* | Kukhtevich et al, Nature Communications, 2020 | KSY260-2 |
| *S. cerevisiae* - KSY274-1: Background W303; *Mat α; ADE2 cdc10::CDC10-mCitrine-ADH1term-HIS3 bnr1Δ::KlacURA3* | This study | KSY274-1 |
| *S. cerevisiae* - KSY277-2: Background W303; *Mat α; ADE2 cdc10::CDC10-mCitrine-ADH1term-HIS3 spa2Δ::KlacURA3* | This study | KSY277-2 |
| *S. cerevisiae* - KSY278-1/2: Background W303; *Mat α; ADE2 cdc10::CDC10-mCitrine-ADH1term-HIS3 bud6Δ::KlacURA3* | This study | KSY278-1/2 |
| *S. cerevisiae* - KSY306-3: Background W303; *Mat a; his3::LexA-ER-AD-TF-HIS3 whi5::kanMX6-LexApr-WHI5-ADH1term-LEU2 exo84::Exo84-mCitrine-ADH1term-cglaTRP1, cdc10::CDC10-mNeptune2.5-ADH1term-ADE2* | This study | KSY306-3 |
| *S. cerevisiae* - KSY309-7/8: Background W303; *Mat a; his3::LexA-ER-AD-TF-HIS3 whi5::kanMX6-LexApr-WHI5-ADH1term-LEU2 exo84::Exo84-mCitrine-Adh1term-cglaTRP1, cdc10::CDC10-mNeptune2.5-Adh1term-ADE2 bni1Δ::NatMX6* | This study | KSY309-7/8 |
| *S. cerevisiae* - KSY317-9/12: Background W303; *Mat a; his3::LexA-ER-AD-TF-HIS3 whi5::kanMX6-LexApr-WHI5-ADH1term-LEU2 abp140::Abp140-mCitrine-Adh1term-cglaTRP1 cdc10::CDC10-mNeptune2.5-Adh1term-ADE2* | This study | KSY317-9/12 |
| *S. cerevisiae* - KSY318-2: Background W303; *Mat a; his3::LexA-ER-AD-TF-HIS3 whi5::kanMX6-LexApr-WHI5-ADH1term-LEU2 abp140::Abp140-mCitrine-ADH1term-cglaTRP1 cdc10::CDC10-mNeptune2.5-ADH1term-ADE2 bni1Δ::NatMX6* | This study | KSY318-2 |
| *S. cerevisiae* - KSY342-1/2: Background YEF473; *Mat a/α; rsr1::TRP1/rsr1::TRP1 bem1::BEM1-GFP-LEU2/bem1::BEM1-GFP-LEU2 GIC2/gic2::GIC2(1-208)-tdTomato:KAN^R his3/his3::LexA-ER-AD-TF-HIS3 WHI5/whi5::LexApr-WHI5-ADH1term-URA3* | This study, based on Kuo et al, Current Biology, 2014 | KSY342-1/2 |
| *S. cerevisiae* - KSY343-2: Background YEF473; *Mat a/α; CDC24^38A/CDC24^38A rsr1::TRP1/rsr1::TRP1 bem1::BEM1-GFP-LEU2/bem1::BEM1-GFP-LEU2 GIC2/gic2::GIC2(1-208)-tdTomato:KAN^R his3/his3::LexA-ER-AD-TF-HIS3 WHI5/whi5::LexApr-WHI5-ADH1term-URA3* | This study, based on Kuo et al, Current Biology, 2014 | KSY343-2 |
| *S. cerevisiae* - STY892: Background JD47; *Mat a; BEM1::BEM1-GFP TRP1, CDC11::CDC11-mCherry URA3, CDC24::CDC24K525E, K801A-natNT2* | Chollet et al, 2020, Journal of Cell Science | STY892 |
| *S. cerevisiae* - STY893: Background JD47 *Mat a; BEM1::BEM1-GFP TRP1, CDC11::CDC11-mCherry URA3, CDC24::CDC24-natNT2* | Chollet et al, 2020, Journal of Cell Science | STY893 |
| *S. cerevisiae* - Y1818, Y1819: Background YEF473; *Mat a/α; rsr1::TRP1/rsr1::TRP1 BEM1-GFP::LEU2/BEM1-GFP::LEU2 CDC10/cdc10::CDC10-mCherry-ADH1term-KanMX* | This study, based on Kuo et al, Current Biology, 2014 | Y1818, Y1819 |
| *S. cerevisiae* - Y1820, Y1821: Background YEF473; *Mat a/α; CDC24^38A/CDC24^38A rsr1::TRP1/rsr1::TRP1 BEM1-GFP:LEU2/BEM1-GFP:LEU2 CDC10/cdc10::CDC10-mCherry-ADH1term-KanMX* | This study, based on Kuo et al, Current Biology, 2014 | Y1820, Y1821 |
| *S. cerevisiae* - KSY321-19: Background W303; *Mat a/α; ADE2/ADE2 cdc10::CDC10-mCitrine-ADH1term-HIS3/cdc10::CDC10-mCitrine-ADH1term-TRP1 ura3/ura3::YIp211-GIC2PBD(W23A)-1.5tdTomato-v2-URA3* * | This study | KSY321-19 |
| *S. cerevisiae* - KSY322-2: Background W303; *Mat a/α; ADE2/ADE2 cdc10::CDC10-mCitrine-ADH1term-HIS3/cdc10::CDC10-mCitrine-ADH1term-TRP1 ura3/ura3::YIp211-GIC2PBD(W23A)-1.5tdTomato-v2-URA3 CDC24/cdc24::cglaLEU2** | This study | KSY322-2 |
| *S. cerevisiae* - KSY348-1/2: Background W303; *Mat a/α; ADE2/ADE2 cdc10::CDC10-mCitrine-ADH1term-HIS3/cdc10::CDC10-mCitrine-ADH1term-TRP1 ura3/ura3::YIp211-GIC2PBD(W23A)-1.5tdTomato-v2-URA3 CDC42/cdc42::cglaLEU2** | This study | KSY348-1/2 |
| *S. cerevisiae* - KSY349-1/2: Background W303; *Mat a/α; ADE2/ADE2 cdc10::CDC10-mCitrine-ADH1term-HIS3/cdc10::CDC10-mCitrine-ADH1term-TRP1 ura3/ura3::YIp211-GIC2PBD(W23A)-1.5tdTomato-v2-URA3 BEM1/bem1::cglaLEU2** | This study | KSY349-1/2 |
| *S. cerevisiae* - KSY350-1/2: Background W303; *Mat a/α; ADE2/ADE2 cdc10::CDC10-mCitrine-ADH1term-HIS3/cdc10::CDC10-mCitrine-ADH1term-TRP1 ura3/ura3::YIp211-GIC2PBD(W23A)-1.5tdTomato-v2-URA3 BEM2/bem2::cglaLEU2** | This study | KSY350-1/2 |
| *S. cerevisiae* - KSY351-1/2: Background W303; *Mat a/α; ADE2/ADE2 cdc10::CDC10-mCitrine-ADH1term-HIS3/cdc10::CDC10-mCitrine-ADH1term-TRP1 ura3/ura3::YIp211-GIC2PBD(W23A)-1.5tdTomato-v2-URA3 GIC2/gic2::cglaLEU2** | This study | KSY351-1/2 |
| *S. cerevisiae* - Y1939, Y1940: Background NY14; *Mat α; his4-619; exo84::EXO84-mScarlet-I-KanMX; cdc10::CDC10-GFP-NatMX* | This study, based on Stalder and Novick, MBoC, 2015 | Y1939, Y1940: |

| Reagent/resource | Reference or source | Identifier or catalog number |
|---|---|---|
| S. cerevisiae - Y1941, Y1942: Background NY63; *Mat α; his4-619, sec10-2; exo84::EXO84-mScarlet-I-KanMX; cdc10::CDC10-GFP-NatMX* | This study, based on Stalder and Novick, MBoC, 2015 | Y1941, Y1942 |

*We compared the fluorescence intensities of multiple clones of KSY321 and selected KSY321-19, which showed about twice the intensity of other clones, suggesting a double integration of the YIp211-GIC2PBD(W23A)-1.5tdTomato-v2 at the URA3 locus.

| **Chemicals, enzymes and other reagents** | | |
|---|---|---|
| Polydimethylsiloxane kit Sylgard 184 | Dow, USA | Material Number 1673921 |
| Paraformaldehyde, 16% w/v | Thermo Fisher | Cat# 43368 |
| Gibco DPBS | Thermo Fisher | Cat# 14190094 |
| NaN3 (sodium azide) | Sigma-Aldrich | S2002-25G |
| Triton X-100 | Sigma-Aldrich | T9284-500ML |
| Bovine Serum Albumin Microbiological Grade | Sigma-Aldrich | Cat# 810651 |
| Acti-Stain 670 Phalloidin | Cytoskeleton Inc, USA | Cat# PHDN1 |
| ProLong Gold Antifade Mountan | Thermo Fisher | Cat# P36930 |
| **Used for SCGE** | | |
| Synthetic complete mix: | | |
| Adenine | Sigma-Aldrich | A9126-100G |
| L-Arginin | Sigma-Aldrich | A5006-100G |
| L-Aspartic acid sodium salt monohydrate | Sigma-Aldrich | A6683-100G |
| L-Glutamic acid potassium salt monohydrate | Sigma-Aldrich | G1501-100G |
| L-Histidine monohydrochloride monohydrate | Sigma-Aldrich | H8125-100G |
| L-Isoleucin | Sigma-Aldrich | I2752-100G |
| L-Leucin | Sigma-Aldrich | L8000-100G |
| L-Lysine monohydrochloride | Sigma-Aldrich | L5626-100G |
| L-Methionin | Sigma-Aldrich | M9625-100G |
| L-Phenylalanin | Sigma-Aldrich | P2126-100G |
| L-Serin | Sigma-Aldrich | S4500-1KG |
| L-Threonine | Sigma-Aldrich | T8625-500G |
| L-Tryptophan | Sigma-Aldrich | T0254-100G |
| L-Tyrosin | Sigma-Aldrich | T3754-100G |
| Uracil | Sigma-Aldrich | U0750-100G |
| L-Valin | Sigma-Aldrich | V0500-500G |
| Yeast Nitrogen Base w/o, Amino Acids and Ammonium Sulfate | BD Diagnostics | 233520 |
| Ammonium sulfate | Sigma-Aldrich | A4418-500G |
| Glycerol | Sigma-Aldrich | G9012-500ML |
| Ethanol absolute for analysis | Merck Millipore | 1009831000 |
| **Used for SCD** | | |
| Drop-out-mix complete w/o yeast nitrogen base | US Biological | Cat #D9515 |
| Difco Yeast Nitrogen Base w/o Amino Acids | Becton-Dickinson | Ref 291940 |
| D-(+)-Glucose | Sigma-Aldrich | G8270-100G |
| **Software** | | |
| Cell ACDC | Laboratory of Kurt M. Schmoller | https://github.com/SchmollerLab/Cell_ACDC |

| Reagent/resource | Reference or source | Identifier or catalog number |
|---|---|---|
| Python | Python Software Foundation, USA | Python v3.11 |
| Ring quant | Laboratory of Kurt M. Schmoller | https://github.com/ElpadoCan/ringQUANT |
| PhyloCell | Laboratory of Gilles Charvin | https://github.com/gcharvin |
| Autotrack | Laboratory of Gilles Charvin | https://github.com/gcharvin |
| MATLAB | MathWorks | Version R2021a |
| Fiji (ImageJ) | Wayne Rasband National Institutes of Health, USA | Version 1.54 f |
| Nikon NIS Elements | Nikon, Japan | Version 5.02.01 |
| Matlab-based software for segmentation | Doncic et al, PLoS ONE, 2013 | N/A |
| Julia | JuliaLang, USA | Julia 1.10 |
| R | R Foundation, USA | R 4.3.3 |
| FEniCS | Logg et al; FEniCS book, 2012 | Version 2019.1.0 |
| Gmsh | Geuzaine et al, International Journal for Numerical Methods in Engineering, 2009 | Version 4.4.1 |
| Simulation software (septin, Cdc42 and particle simulator models) | Laboratory of Marija Cvijovic | https://github.com/sebapersson/cdc42_and_septin_ring_paper |
| **Other** | | |
| 24 × 50 mm #1.5 glass coverslip | Knittel Glass, Germany | Cat# VD12450Y1A.01 |
| 1 mm biopsy puncher | Integra Miltex, USA | Cat# 33–31AA-P/25 |
| Tygon tubing 0.02 inch ID × 0.060 inch OD | Cole-Parmer, Germany | Cat# GZ-06419-01 |
| 18 gauge 0.5-inch bent 90° blunt needle | Techcon, USA | Cat# TE718050B90PK |
| Microfluidic flow controller | Elveflow, France | Cat# OB1 MK3+ |
| Microfluidic flow sensor | Elveflow, France | Cat# MFS 3 |
| Port selector valve | Idex, USA | Cat# MXX778-605 |
| Objective heater | Okolab, Italy | N/A |
| Heating chip holder | custom-built | N/A |
| Epifluorescence microscope | Nikon, Japan | Cat# Eclipse Ti-E |
| 100× Objective | Nikon, Japan | Cat# plan-apo λ 100×/1.45NA Ph3 oil immersion |
| Light engine | Lumencor, USA | Cat# SPECTRA X |
| EMCCD camera | Andor, UK | Cat# iXon Ultra 888 |

## Strain construction

*S. cerevisiae* strains were constructed using standard lithium acetate transformation. A detailed strain list can be found in the "Reagents and Tools Table". Where specified, we used *CDC10-mCITRINE*, *CDC10-mNEPTUNE2.5*, *CDC10-mCHERRY*, or *CDC11-mCHERRY*

to visualize the septin ring, *GIG2PDB-tdTOMATO* to visualize Cdc42-GTP, *ABP140-mCITRINE* to visualize F-actin, and *EXO84-mCITRINE* to visualize exocytosis. Strains with inducible Whi5 were used to tune cell size while maintaining cycling cell populations with similar doubling times [Kukhtevich et al, 2020; Claude et al, 2021]. Strains are available upon request.

## Growth conditions

Before experiments, cells were cultured in synthetic complete liquid medium at 30 °C at low density to ensure exponential growth, except for the *sec10-2* temperature-sensitive mutant and the corresponding wild-type strain, which were grown at 25 °C. For time-lapse experiments with the *sec10-2* temperature-sensitive mutant, cells were shifted to 37 °C 170 min before the start of image acquisition. As a carbon source for all experiments, we used 2% glycerol and 1% ethanol (SCGE), except for experiments with the *sec10-2* temperature-sensitive mutant, for which 2% glucose (SCD) was used. After pre-culturing, strains carrying β-estradiol-inducible *WHI5* were grown in the presence of the respective β-estradiol concentration for ~24 h before the start of the experiment, to ensure a steady state. Medium without β-estradiol was used during the live-cell microscopy experiments.

## Microfluidic devices

To acquire live single-cell data during time-lapse experiments, we used a previously reported custom-made microfluidic device that allows isolating cells in a dedicated region of interest and limits colony growth to the XY-plane [Kukhtevich et al, 2022]. The device includes eight separate cell culture chambers with a controllable medium exchange that enables parallel imaging of up to eight strains.

The microfluidic device was fabricated by means of standard soft lithography. Briefly, by using photolithography, a master mold for replication of the device design in polydimethylsiloxane (PDMS) was fabricated from SU-8 photoresist (MicroChem, USA) spin-coated on a 3″ Si wafer. The master mold was then filled with a 10:1 mixture of the base to curing agent of PDMS kit Sylgard 184 (Dow Corning, USA) and left at 60 °C for 4 h to crosslink the PDMS. After cross-linking, the PDMS replica was cut and peeled off from the master mold, and necessary inlets and outlets for tubing connections were made using a 1 mm puncher. Next, the replica was sealed with a coverslip after both were treated in O₂ plasma.

## Live-cell microscopy

Live-cell time-lapse experiments (Figs. 1G, 2E, 3, 4A–G, 5H–L, 6A–D; S1, S2; EV5, S5) were performed using the custom-made microfluidic device described above. Different strains were separately loaded in different chambers of the device. Constant medium flow at 20 µL/min was applied, enabling imaging of a colony growing over approximately six generations (10 h). Images were taken every 3 min for all time-lapse experiments, except for experiments imaging Abp140-mCitrine, for which a time interval of 10 min was used instead to allow for the acquisition of five z-slices with 1-µm steps. Temperature control was achieved by setting both a custom-made heatable insertion and an objective heater to 30 °C, except for experiments with the *sec10-2* temperature-sensitive mutant, for which the temperature was set to 37 °C.

A Nikon Eclipse Ti-E with SPECTRA X light engine illumination and an Andor iXon Ultra 888 camera were used for epifluorescence microscopy. A plan-apo λ 100x/1.45NA Ph3 oil immersion objective was used to take phase contrast and fluorescence images. mCitrine fluorescence was imaged by exposure for 400 ms, illuminating with the SPECTRA X light engine at 504 nm and about 12 mW (20%) power for all experiments, except for experiments imaging Abp140-mCitrine, for which the exposure time was set to 300 ms and the power to about 24 mW (40%). tdTomato fluorescence was imaged by exposure for 200 ms, illuminating with the SPECTRA X light engine at 555 nm and about 26 mW (10%) power. mCherry fluorescence was imaged by exposure for 200 ms, illuminating with the SPECTRA X light engine at 555 nm and about 26 mW (10%) power. mNeptune2.5 fluorescence was imaged by exposure for 400 ms, illuminating with the SPECTRA X light engine at 556 nm and about 26 mW (10%) power for all experiments except for experiments imaging Abp140-mCitrine, for which the exposure time was set to 300 ms. GFP fluorescence was imaged by exposure for 300 ms, illuminating with the SPECTRA X light engine at 470 nm and about 39 mW (20%) power. mScarlet-I fluorescence was imaged by exposure for 300 ms, illuminating with the SPECTRA X light engine at 555 nm and about 39 mW (15%) power.

Fluorescence of all fluorescent proteins was detected using suitable emission wavelength filters.

## Phalloidin staining and imaging

The protocol of staining with phalloidin to visualize F-actin in fixed cells was adapted from [Sing et al, 2022]. Briefly, cells were grown in SDC overnight and diluted in the morning to reach mid-log phase. Cells were fixed using paraformaldehyde at a final concentration of 3.7% and incubated for 1 h at 30 °C. Then, cells were washed three times with 1xDPBS, once with 1xDPBS containing 1% bovine serum albumin, 0.1% NaN₃ 10% (w/v), and 0.1% Triton X-100, and resuspended in the same buffer. Acti-Stain 670 Phalloidin (Cytoskeleton Inc, USA) was added and the suspension was incubated at room temperature for 1 h. Next, stained cells were washed three times in DPBS and resuspended in mounting media ProLong Gold (Thermo Fisher Scientific). For imaging, stained cells were loaded into the microfluidic device described above.

For phase contrast and fluorescence imaging, the Nikon Eclipse Ti-E with SPECTRA X light engine illumination, Andor iXon Ultra 888 camera and plan-apo λ 100x/1.45NA Ph3 oil immersion objective was used. mCitrine fluorescence corresponding to Cdc10 was imaged by exposure for 400 ms, illuminating with the SPECTRA X light engine at 504 nm and about 12 mW (20%) power. Phalloidin fluorescence corresponding to F-actin was imaged by exposure for 500 ms, illuminating with the SPECTRA X light engine at 640 and about 231 mW (100%) power. Suitable emission wavelength filters were used. For each ROI, 21 z-slices with 0.2-µm intervals were recorded.

## Quantification and statistical analysis

### Cell segmentation and tracking

For experiments shown in Figs. 1, 2D,E, 4D–G, 5H–J, 6; S1, S2; EV3, EV5; S5 cell segmentation, cell volume calculations, lineage annotations and cell-cycle stage assignments were performed using the Cell-ACDC software available at https://github.com/SchmollerLab/Cell_ACDC [Padovani et al, 2022]. More specifically,

we used the YeaZ [Dietler et al, 2020] neural network option in Cell-ACDC for segmentation and tracking and manually corrected where necessary.

For the experiments shown in Fig. 4A–C, cells were automatically segmented and tracked based on phase-contrast images using the Matlab-based Phylocell software [Fehrmann et al, 2013]. The results of automatic segmentation and tracking were visually inspected and manually corrected if necessary.

For the experiments shown in Fig. 3, cells were segmented and tracked based on phase-contrast images using the Matlab-based software described in [Doncic et al, 2013]. The result was manually checked for each cell included in the analysis, and poorly segmented or wrongly tracked cells were rejected from the analysis.

### Calculation of cell volume and length along the major axis

Cell volume and length along the major axis was calculated based on 2D phase contrast images as described previously [Kukhtevich et al, 2020, Padovani et al, 2022]. Briefly, cell contours were aligned along their major axis, and divided into slices perpendicular to the major axis. To estimate cell volume, we then assumed rotational symmetry of each slice around its middle axis parallel to the cell's major axis.

### Analysis of the septin ring diameter and Exo84 cluster diameter

For the experiments shown in Fig. 3, an analysis of the septin ring diameter was performed using Fiji and Matlab, as previously described [Kukhtevich et al, 2020]. Briefly, Fiji was used to automatically determine the position and orientation of the ring and obtain an intensity line profile along its major axis. This intensity profile was then further analyzed using MATLAB to quantify the ring diameter.

For the experiments shown in Fig. 4A–C, the septin ring diameter based on Cdc10-mNeptune2.5 fluorescence was calculated from the five frames centered around the time of clear bud emergence, using an approach previously described in [Kukhtevich et al, 2020]. To do so, a fluorescence profile along a cell segmentation contour was taken for each frame after applying the $3 \times 3$ mean filter on the fluorescence image. A mean fluorescence profile was then calculated based on the five selected frames. Finally, we used the full width at half maximum as an estimate for the septin ring diameter. For this, the minimum signal in the fluorescence profile was defined as the baseline. All length profiles were visually inspected, and cells were rejected from further analysis when this approach resulted in obvious artifacts.

The septin ring diameters shown in Figs. 4F,G; 5I,J; 6B,D; S2; EV5; S5 and the Exo84 cluster diameter shown in Fig. 4E,G were measured as follows: first, the cells were segmented, tracked over time, and their cell cycle progression was annotated from phase contrast signal using the software Cell-ACDC [Padovani et al, 2022]. Specifically, for segmentation and tracking, we used the model YeaZ (embedded in Cell-ACDC) [Dietler et al, 2020]. Segmentation and tracking were visually inspected, and errors were corrected with Cell-ACDC. The cell cycle progression was determined by annotating bud emergence, mother-bud pairs, and cell division. Note that cell division is visually determined by carefully checking for sudden bud movement (indicating the bud has divided from the mother cell). Next, we used an automatic custom Python routine to calculate the ring/cluster diameter for each complete cell cycle. In the first step, the algorithm applies a

Gaussian filter (with 'sigma = 2.0') to each frame of the video. Next, the routine extracts the intensities from an elliptical region whose longer axis is aligned with the line connecting the centers of mass of the mother and bud, while the center of the ellipse lies on the contact point between the mother and bud. The contact point is determined as the point along the line connecting the centers of mass where the segmentation ID changes. The shorter axis length was set to 10 pixels, while the longer axis of the ellipse was calculated as the distance between the two points intersecting the long axis and the contour of the hull image of the mother-bud object. These two distances were selected to ensure that the brightest part of the ring/cluster is included in the elliptical area. In the second step, the intensities from this area are sorted in descending order, and the 10th value from maximum intensity is taken as a representative data point for each timeframe to build a cell cycle curve over time. Note that other values were tested, including the mean, the max, the median, the 20th value from maximum, etc. and we found the 10th value to be the more robust for the next step. In the third step, a threshold value was determined for each cell cycle as the mean between the maximum and minimum in the curve. In the fourth step, the ring/cluster structure was segmented for each time point and cell by applying a threshold on the Gaussian-filtered intensities in each mother-bud object using the threshold value determined in the previous step. In the fifth step, for each time point, only the cells where one (for Exo84 signal) or a maximum of two (for septin ring signal) objects overlapping by at least 50% with the elliptical region (see above) were kept for the analysis. Note that the overlap was calculated as the intersection-over-area ratio, where intersection is the number of pixels in both the elliptical region and the subcellular objects, while area is the number of pixels in the subcellular objects mask. For the experiments with the *sec10-2* temperature-sensitive mutant and corresponding wild-type strain (Fig. EV5), we analyzed data between 170 min to 410 min after the shift to 37 °C. Since mutant cells tend to have longer S/G2/M phase or even fail to complete cell division, to compare to the reference wild-type strain in a similar S/G2/M phase window, we restricted the search for the maximum ring/Exo84 cluster diameter to G1 plus a maximum of 30 min of the S/G2/M phase.

Finally, the ring diameter/Exo84 cluster diameter is determined as the major axis length of the segmented object using the scikit-image function skimage.measure.regionprops [van der Walt et al, 2014]. The maximum of the septin ring/Exo84 cluster diameter was determined from the evolution of the diameter over time. Cells whose S/G2/M phase was not fully tracked and the tracked duration was less than 45 min, and cells for which the ring/cluster segmentation was not successful for at least three or two frames, respectively, were discarded from the analysis. For the ring diameter over time plots in Fig. 5J and S5a, cells without a fully tracked G1 phase were discarded.

### Analysis of the Cdc42

To determine the Cdc42 cluster areas in Figs. 1G, 2F, S1b, S511b, we developed an automatic custom Python routine that starts from fully annotated cell cycles of single cells and is based on our previously published method [Kukhtevich et al, 2020]. See the previous section for more details about how the cell pedigrees were annotated. In the first step, the algorithm applies a Gaussian filter (with "sigma = 2.0") to each frame of the video. Next, the routine determines a threshold value

**Table 1. Model reactions and kinetic parameter values, positive feedback model.**

| Reaction | Parameter | Value | Reference |
|---|---|---|---|
| $BemGEFc \rightarrow BemGEFm$ | $k_{1a}$ | 10 | [Chiou et al, 2021] |
| $BemGEFm \rightarrow BemGEFc$ | $k_{1b}$ | 10 | [Chiou et al, 2021] |
| $Cdc42D + BemGEFm \rightarrow Cdc42T + BemGEFm$ | $k_{2a}$ | 0.16 | [Chiou et al, 2021] |
| $Cdc42T \rightarrow Cdc42D$ | $k_{2b}$ | 0.35 | [Chiou et al, 2021] |
| $Cdc42D + BemGEF42 \rightarrow Cdc42T + BemGEF42$ | $k_3$ | 0.35 | [Chiou et al, 2021] |
| $BemGEF + Cdc42T \rightarrow BemGEF42$ | $k_{4a}$ | 10 | [Chiou et al, 2021] |
| $BemGEF42 \rightarrow BemGEFm + Cdc42T$ | $k_{4b}$ | 10 | [Chiou et al, 2021] |
| $Cdc42I \rightarrow Cdc42D$ | $k_{5a}$ | 36 | [Chiou et al, 2021] |
| $Cdc42D \rightarrow Cdc42I$ | $k_{5b}$ | 0.65 | [Chiou et al, 2021] |
| $BemGEFc + Cdc42T \rightarrow BemGEF42$ | $k_7$ | 10 | [Chiou et al, 2021] |

for each cell using the following formula:

$$threshold\_val = cell\_median + 2 * cell\_std \qquad (1)$$

where cell_median and cell_std are the median and the standard deviation of the Gaussian-filtered intensities in each cell, respectively. In the second step, the algorithm thresholds the intensities from each cell using the threshold values determined in the previous step to achieve segmentation of the clusters. In the third step, the algorithm identifies individual segmented objects in the cell and keeps only the largest one. Finally, comparing the different time points, the routine extracts the maximum cluster area prior to bud emergence from cells with fully tracked G1 phase. The intensity of the brightest pixel in the cluster was used as a measure for the maximum Cdc42-GTP intensity in Fig. 2F and S1b. For the Cdc42-GTP cluster area over time plot in Fig. S5b, cells without fully tracked G1 were discarded from the analysis.

### Analysis of the Abp140 cluster

All parameters of the Abp140 cluster (Fig. 4B,C) were calculated for the three frames centered around the frame with the peak Abp140 localization, which was determined by visual inspection. Maximum projections of five z-slices with 1 μm intervals were used for the analysis.

To measure the Abp140 cluster area, we applied a thresholding approach similar to that used by Okada et al, to measure Cdc42-GTP cluster area [Okada et al, 2013; Kukhtevich et al, 2020]. For each cell and each time point, a threshold was defined as the median value + 2 standard deviations of the fluorescence pixel intensities within the selected cell. All pixels with a value higher than this threshold were then counted as part of the cluster, and the cluster area was defined as the median number of pixels in the cluster across the three analyzed frames.

### Analysis of the actin cluster based on phalloidin staining

To quantify the actin cluster area around the bud site (Fig. EV3) to verify the result shown in Fig. 4B in the manuscript, we used maximum projections of the phalloidin signal. Since in this case, we could not quantify actin cluster area at peak localization due to the lack of time information, we decided to instead classify cells based on septin localization (visible formed septin ring) and bud size (small bud, i.e., close to bud emergence). Only cells fulfilling both criteria, as determined from visual inspection, were included in the analysis. To calculate the actin cluster area, the images were pre-processed with a Gaussian filter with sigma = 2.0. Then, for each valid cell, we extracted the intensities from an elliptical area with 30 and 20 pixels lengths for the long and short axis, respectively. The ellipse was centered on the mother-bud neck and oriented with the long axis perpendicular to the line connecting the mother and bud centers of mass. Next, from these intensities, we calculated a threshold value as the mean intensity plus two times the standard deviation. We then thresholded the entire image using this threshold value. To keep only the actin cluster of the specific cell, we removed all clusters that did not overlap with the elliptical area by at least 80%. Finally, we calculated the cluster area as the sum of the thresholded pixels.

### Analysis of triple-SILAC proteomics

From the dataset from Lanz et al, we tried to extract protein slopes for the following key Cdc42 polarity proteins: Cdc24, Bud3, Bem2, Bem3, Rga1, Rga2, Cla4, Ste20, Bem1, Gic1, Gic2, Axl2, Rsr1, Cdc42, Cdc11, Cdc12, Cdc3, and Cdc10. Noticeably, data were missing on a subset, including Gic1.

## Computational modeling

### Positive feedback model

The positive feedback model consists of the reactions in Table 1. It was originally published in [Woods et al, 2015], and is based on the model of Goryachev and Pokhilko [Goryachev and Pokhilko, 2008]. Model species are Cdc42-GTP ($Cdc42T$, membrane-bound), Cdc42-GDP ($Cdc42D$, membrane-bound), Cdc42-GDI ($Cdc42c$, cytosolic), Bem-GEF-Cdc42-GTP complex ($BemGEF42$, membrane-bound), and the Bem-GEF/Cdc24 complex ($BemGEFm$ membrane-bound and $BemGEFc$ cytosolic).

Assuming much faster cytosolic than membrane diffusion ($D_c \rightarrow \infty$), the reactions in Table 1 comprise a coupled system of ordinary differential equations (ODEs) and partial differential equations (PDEs):

$$\begin{aligned}
\frac{\partial Cdc42T}{\partial t} = {} & (k_{2a}BemGEFm + k_3BemGEF42)Cdc42D \\
& - (k_{2b} + k_{4a}BemGEFm + k_7BemGEFc)Cdc42T \\
& + k_{4b}BemGEF42 + D_m\nabla^2 Cdc42T
\end{aligned} \qquad (2)$$

$$\begin{aligned}
\frac{\partial Cdc42D}{\partial t} = {} & k_{2b}Cdc42T - (k_{2a}BemGEFm + k_3BemGEF42)Cdc42D \\
& - k_{5b}Cdc42D + k_{5a}Cdc42c + D_m\nabla^2 Cdc42D
\end{aligned}$$

$$\qquad (3)$$

**Table 2. Model reactions and kinetic parameter values negative feedback model.**

| Reaction | Parameter | Value | Reference |
|---|---|---|---|
| $BemGEFc \rightarrow BemGEFm$ | $k_{1a}$ | 10 | [Chiou et al, 2021] |
| $BemGEFm \rightarrow BemGEFc$ | $k_{1b}$ | 10 | [Chiou et al, 2021] |
| $Cdc42D + BemGEF \rightarrow Cdc42T + BemGEFm$ | $k_{2a}$ | 0.16 | [Chiou et al, 2021] |
| $Cdc42T \rightarrow Cdc42D$ | $k_{2b}$ | 0.35 | [Chiou et al, 2021] |
| $Cdc42D + BemGEF42 \rightarrow Cdc42T + BemGEF42$ | $k_3$ | 0.35 | [Chiou et al, 2021] |
| $BemGEF + Cdc42T \rightarrow BemGEF42$ | $k_{4a}$ | 10 | [Chiou et al, 2021] |
| $BemGEF42 \rightarrow BemGEFm + Cdc42T$ | $k_{4b}$ | 10 | [Chiou et al, 2021] |
| $Cdc42I \rightarrow Cdc42D$ | $k_{5a}$ | 36 | [Chiou et al, 2021] |
| $Cdc42D \rightarrow Cdc42I$ | $k_{5b}$ | 0.65 | [Chiou et al, 2021] |
| $BemGEFc + Cdc42T \rightarrow BemGEF42$ | $k_7$ | 10 | [Chiou et al, 2021] |
| $BemGEF^* \rightarrow BemGEFc^*$ | $k_{1b}$ | 10 | [Chiou et al, 2021] |
| $BemGEF^* + Cdc42T \rightarrow BemGEF42^*$ | $k_{4a}$ | 10 | [Chiou et al, 2021] |
| $BemGEF42^* \rightarrow BemGEF^* + Cdc42T$ | $k_{4b}$ | 10 | [Chiou et al, 2021] |
| $BemGEF42 + BemGEF42 \rightarrow BemGEF42 + BemGEF42^*$ | $k_{8\max}, k_{8h}k_{8n}$ | 0.0063, 6, 10 | [Chiou et al, 2021] |
| $BemGEF42 + BemGEF42^* \rightarrow BemGEF42^* + BemGEF42^*$ | $k_{8\max}, k_{8h}k_{8n}$ | 0.0063, 6, 10 | [Chiou et al, 2021] |
| $BemGEFc^* \rightarrow BemGEFc$ | $k_{9\max}, k_{9h}k_{9n}$ | 0.0044, 6, 10.003 | [Chiou et al, 2021] |

$$\frac{\partial BemGEF42}{\partial t} = (k_{4a}BemGEFm + k_7BemGEFc)Cdc42T \\ - k_{4b}BemGEF42 + D_m\nabla^2 BemGEF42 \tag{4}$$

$$\frac{\partial BemGEF_m}{\partial t} = k_{1a}BemGEFc - k_{1b}BemGEFm + k_{4b}BemGEF42 \\ - k_{4a}BemGEF42Cdc42T + D_m\nabla^2 BemGEF_m \tag{5}$$

$$\frac{\partial Cdc42I}{\partial t} = \frac{\eta}{A_m}\int_{A_m}(k_{5a}Cdc42I - k_{5b}Cdc42D) \tag{6}$$

$$\frac{\partial BemGEF_c}{\partial t} = \frac{\eta}{A_m}\int_{A_m}(k_{1a}BemGEF_c - k_{1b}BemGEF_m - k_7BemGEFc * Cdc42T) \tag{7}$$

where $A_m$ is the cell-surface area, and $\eta$ is the membrane-to-cytoplasm volume ratio $V_m/V_c$. We assume a membrane thickness of $R_m \approx 10nm$ [Goryachev and Pokhilko, 2008] and thereby $\eta$ can be computed as:

$$\eta = \left((R + R_m)^3 - R^3\right)/R^3 \tag{8}$$

Thus, the membrane is treated as a compartment, and the concentration for surface species refers to the concentration in the membrane compartment.

### Negative-feedback model

The positive feedback model can be expanded to include negative feedback. The resulting negative feedback model consists of the reactions in Table 2, and was originally published in [Kuo et al, 2014]. The model species are Cdc42-GTP (*Cdc42T*, membrane-bound), Cdc42-GDP (*Cdc42D*, membrane-bound), Cdc42-GDI (*Cdc42c*, cytosolic), Bem-GEF-Cdc42-GTP complex (*BemGEF42*, membrane-bound), and the Bem-GEF/Cdc24 complex (*BemGEFm* membrane-bound and *BemGEFc* cytosolic). Asterisks, e.g., *BemGEF42\**, denote phosphorylated species.

The difference between the negative feedback model and the positive feedback model is the presence of one delayed negative feedback, which, via a Hill activation, is triggered when *BemGEF42* activity is sufficiently high. Briefly, when *BemGEF42* activity gets too high, it autophosphorylates, and in the phosphorylated state, it cannot promote *Cdc42T* production. To become dephosphorylated, *BemGEF\** must be recycled into the cytosol. That dephosphorylation only occurs in the cytosol creates a delayed negative feedback, which, besides reducing Cdc42-GTP activation, facilitates, as observed experimentally [Howell et al, 2012], oscillations in Cdc42-GTP cluster intensity during polarization. Overall, the model equations are:

$$\frac{\partial Cdc42T}{\partial t} = (k_{2a}BemGEFm + k_3BemGEF42)Cdc42D \\ - (k_{2b} + k_{4a}BemGEFm_t + k_7BemGEFc_t)Cdc42T \\ + k_{4b}BemGEF42 + D_m\nabla^2 Cdc42T \tag{9}$$

$$\frac{\partial Cdc42D}{\partial t} = k_{2b}Cdc42T - (k_{2a}BemGEFm + k_3BemGEF42)Cdc42D \\ - k_{5b}Cdc42D + k_{5a}Cdc42c + D_m\nabla^2 Cdc42D \tag{10}$$

$$\frac{\partial BemGEF42}{\partial t} = (k_{4a}BemGEFm + k_7BemGEFc)Cdc42T \\ - k_{4b}BemGEF42 - k_8BemGEF42_t \cdot BemGEF42 \\ + D_m\nabla^2 BemGEF42 \tag{11}$$

$$\frac{\partial BemGEF42^*}{\partial t} = (k_{4a}BemGEFm^* + k_7BemGEFc^*)Cdc42T \\ - k_{4b}BemGEF42^* + k_8BemGEF42_t \\ \cdot BemGEF42 + D_m\nabla^2 BemGEF42^* \tag{12}$$

$$\frac{\partial BemGEF_m}{\partial t} = k_{1a}BemGEFc - k_{1b}BemGEFm + k_{4b}BemGEF42 \\ - k_{4a}BemGEF42Cdc42T + D_m\nabla^2 BemGEF_m \tag{13}$$

**Table 3. Model reactions for the alternative positive feedback model.**

| Reaction | Parameter | Value | Reference |
|---|---|---|---|
| $Cdc42I \rightarrow Cdc42D$ | $k_1, k_{max}$ | 0.28, 2.54 | MS in preparation |
| $Cdc42D \rightarrow Cdc42I$ | $k_{-1}$ | 0.133 | MS in preparation |
| $Cdc42D \rightarrow Cdc42T$ | $k_2$ | 0.001368 | MS in preparation |
| $Cdc42T \rightarrow Cdc42D$ | $k_{-2}$ | 0.028 | MS in preparation |
| $2Cdc42T + Cdc42D \rightarrow 3Cdc42T$ | $k_3$ | 1 | MS in preparation |
| Membrane diffusion $Cdc42T$ | $D_T$ | 0.011 | MS in preparation |
| Membrane diffusion $Cdc42D$ | $D_D$ | 1.21 | MS in preparation |
| Cytosolic diffusion $Cdc42I$ | $D_I$ | 10 | MS in preparation |

$$\frac{\partial BemGEF_m^*}{\partial t} = k_{1a}BemGEFc^* - k_{1b}BemGEFm^* + k_{4b}BemGEF42^*$$
$$- k_{4a}BemGEFm^* Cdc42T + D_m\nabla^2 BemGEF_m^* \tag{14}$$

$$\frac{dCdc42I}{dt} = \frac{\eta}{A}\int_{A_m}(k_{5b}Cdc42D - k_{5a}Cdc42I)dx \tag{15}$$

$$\frac{dBemGEF_c}{dt} = \frac{\eta}{A}\int_A (k_{1b}BemGEFm - k_{1a}BemGEFc$$
$$- k_7BemGEFcCdc42T)dx + k_9BemGEFc^* \tag{16}$$

$$\frac{dBemGEF_c^*}{dt} = \frac{\eta}{A}\int_A (k_{1b}BemGEFm^* - k_{1a}BemGEFc^*$$
$$- k_7BemGEFc^* Cdc42T)dx - k_9BemGEFc^* \tag{17}$$

### Alternative positive feedback model

The alternative Cdc42 positive feedback model (Fig. EV1A) is based on the reactions in Table 3 and was published in [Borgqvist et al, 2021]. The model includes three species: Cdc42-GTP (*Cdc42T*, membrane-bound), Cdc42-GDP (*Cdc42D*, membrane-bound), and Cdc42-GDI (*Cdc42I*, cytosolic). The third-order reaction (reaction 5 in Table 3) is a simplification of the positive feedback mechanisms into a single reaction step. The parameters for this model were derived in a manuscript under preparation.

If the reactions are turned into equations, we obtain:

$$\frac{\partial Cdc42I}{\partial t} = D_I\nabla^2 Cdc42I, x \in \Omega \tag{18}$$

$$-D_I(\nabla Cdc42I \cdot n) = k_1 Cdc42I(k_{max} - (Cdc42T + Cdc42D))$$
$$- k_{-1}Cdc42D, x \in \Gamma \tag{19}$$

$$\frac{\partial Cdc42T}{\partial t} = k_2 Cdc42D - k_{-2}Cdc42T + k_3 Cdc42T^2 Cdc42D$$
$$+ k_3 Cdc42T^2 Cdc42D + D_T\nabla_I^2 Cdc42T, x \in \Gamma \tag{20}$$

$$\frac{\partial Cdc42D}{\partial t} = - k_2 Cdc42D + k_{-2}Cdc42T - k_3 Cdc42T^2 Cdc42D$$
$$+ k_1 Cdc42I(k_{max} - (Cdc42T + Cdc42D))$$
$$- k_{-1}Cdc42D + D_D\nabla_I^2 Cdc42D, x \in \Gamma \tag{21}$$

Here $\Omega$ refers to the cytosol, and $\Gamma$ to the membrane. Note that compared to other models, cytosolic diffusion is here treated as finite, and the cytosol is part of the model geometry (Fig. EV1).

### Septin ring models

The septin ring model consists of the reactions in Table 4. Model species are Cdc42-GTP (*Cdc42T*, membrane-bound), Cdc42-GDP (*Cdc42D*, membrane-bound), Cdc42-GDI (*Cdc42c*, cytosolic), Bem-GEF/Cdc24 complex (*BemGEFm* membrane-bound and *BemGEFc* cytosolic), Bem-GEF-Cdc42-GTP complex (*BemGEF42*, membrane-bound), Axl2 (*Axl2* membrane-bound and *Axl2c* cytosolic), septin-associated Gap (*GapS* membrane-bound and *GapSc* cytosolic), monomeric septin (*S* membrane-bound and *Sc* cytosolic), polymerized septin (*P*, membrane-bound), and septin recruiter (*X*, membrane-bound). For a motivation of the model structure, see Appendix.

Model parameters were tuned with the following rationale:

- Axl2 recruitment parameters ($k_{22}, k_{23}$) were set to allow relatively rapid recycling between membrane and cytosol, to mimic the behavior of other polarity components such as Cdc42.
- Axl2 septin-recruiting parameters ($k_{15}, k_{19}, k_{20}$) were set to enable sufficiently fast septin recruitment. For example, if $k_{20}$ is too small, ring formation does not take off. If $k_{15}$ is too large, septin polymerizes too strongly in the pole center, causing the Cdc42-GTP cluster to collapse due to the septin-recruited GAP proteins.
  - $k_{19}$ is in this work also referred to as the Septin Polarity-factors binding rate (SPR)
  - $k_{20}$ is in this work also referred to as the Polarity-factors septin recruitment Rate (PSR).

- Septin polymerization parameters ($k_{18}, k_{19}$) were selected to allow sufficiently fast polymerization. These, along with Axl2 septin binding parameters ($k_{15}, k_{19}, k_{20}$), were kept small enough to prevent strong polymerization in the Cdc42 cluster center, which causes cluster collapse, but large enough to allow ring formation.
- Septin-associated Gap (GAP-S) parameters ($k_{12a}, k_{12b}, k_{13}$) were chosen to facilitate sufficiently fast recruitment of GAP-S. Parameter $k_{13}$ was set strong enough to allow the septin ring to "capture" Cdc42-GTP, acting as a substitute for a diffusion barrier (see Fig. 5B).
- X-associated parameters ($k_{24}, k_{25}$) were adjusted to ensure a wider pole of X when exocytosis is diffused (see Appendix).
- Parameters $k_{2b}, k_{5a}$ and $k_{5b}$ differ from the model used for assessing pole size (Table 1). This adjustment was made to make the pole more robust to exocytosis, and to obtain a more realistic, smaller cluster (important for forming a septin ring in a realistic

**Table 4.** Model reactions for the SBE and SBER models.

| Reaction | Parameter | Value | Reference |
|---|---|---|---|
| $BemGEFc \rightarrow BemGEFm$ | $k_{1a}$ | 10 | [Chiou et al, 2021] |
| $BemGEFm \rightarrow BemGEFc$ | $k_{1b}$ | 10 | [Chiou et al, 2021] |
| $Cdc42D + BemGEFm \rightarrow Cdc42T + BemGEFm$ | $k_{2a}$ | 0.16 | [Chiou et al, 2021] |
| $Cdc42T \rightarrow Cdc42D$ | $k_{2b}$ | 0.63 | [Ghose et al, 2021] |
| $Cdc42D + BemGEF42 \rightarrow Cdc42T + BemGEF42$ | $k_3$ | 0.35 | [Chiou et al, 2021] |
| $BemGEF + Cdc42T \rightarrow BemGEF42$ | $k_{4a}$ | 10 | [Chiou et al, 2021] |
| $BemGEF42 \rightarrow BemGEFm + Cdc42T$ | $k_{4b}$ | 10 | [Chiou et al, 2021] |
| $Cdc42I \rightarrow Cdc42D$ | $k_{5a}$ | 144 | [Ghose et al, 2021] |
| $Cdc42D \rightarrow Cdc42I$ | $k_{5b}$ | 20.8 | [Ghose et al, 2021] |
| $BemGEFc + Cdc42T \rightarrow BemGEF42$ | $k_7$ | 10 | [Chiou et al, 2021] |
| $GapSc + P \rightarrow GapS + P$ | $k_{12a}$ | 10 | [Chiou et al, 2021] |
| $GapS \rightarrow GapSc$ | $k_{12b}$ | 10 | This work |
| $GapS + Cdc42T \rightarrow GapS + Cdc42D$ | $k_{13}$ | 1.5 | This work |
| $Axl2S \rightarrow Axl2 + S$ | $k_{15}$ | 1.0 | This work |
| $S + S \rightarrow 2P$ | $k_{16}$ | 0.05 | This work |
| $S + P \rightarrow P + P$ | $k_{17}$ | 0.125 | This work |
| $P \rightarrow S$ | $k_{18}$ | 0.1 | This work |
| $Axl2 + S \rightarrow Axl2$ | $k_{19}$ | 4.5 | This work |
| $Axl2 + Sc \rightarrow Axl2S$ | $k_{20}$ | 0.2 | This work |
| $S \rightarrow Sc$ | $k_{21}$ | 0.65 | This work |
| $Axl2 \rightarrow Axl2c$ | $k_{22}$ | 10.5 | This work |
| $BemGEF42 + Axl2c \rightarrow BemGEF42 + Axl2$ | $k_{23}$ | 26 | This work |
| $X \rightarrow \phi$ | $k_{24}$ | 0.1 | This work |
| $X + Sc \rightarrow X + S$ | $k_{25}$ | 5.5 | This work |
| Membrane diffusion | $D_m$ | 0.0025 | [Chiou et al, 2021] |
| Membrane diffusion P and X | $D_p$ | 0.00025 | This work |

The SBE model is identical to the SBER model, except it does not include any of the reactions with species $X$.

context). For assessing pole size, we aimed to maintain similar parameters to the negative feedback model for comparison.

Overall, if the model reactions are translated into equations, we get:

$$\frac{\partial Cdc42T}{\partial t} = (k_{2a}BemGEFm + k_3BemGEF42)Cdc42D - k_{13}Cdc42T$$
$$- (k_{2b} + k_{4a}BemGEFm + k_7BemGEFc)Cdc42T$$
$$+ k_{4b}BemGEF42 + D_m\nabla^2 Cdc42T \tag{22}$$

$$\frac{\partial Cdc42D}{\partial t} = k_{2b}Cdc42T - (k_{2a}BemGEFm + k_3BemGEF42)Cdc42D$$
$$+ k_{13}Cdc42T - k_{5b}Cdc42D + k_{5a}Cdc42c$$
$$+ D_m\nabla^2 Cdc42D$$

$$\frac{\partial BemGEF42}{\partial t} = (k_{4a}BemGEFm + k_7BemGEFc)Cdc42T - k_{4b}BemGEF42$$
$$+ D_m\nabla^2 BemGEF42 \tag{23}$$

$$\frac{\partial BemGEF_m}{\partial t} = k_{1a}BemGEFc - k_{1b}BemGEFm + k_{4b}BemGEF42$$
$$- k_{4a}BemGEF42Cdc42T + D_m\nabla^2 BemGEF_m \tag{24}$$

$$\frac{\partial Axl2}{\partial t} = k_{15}Axl2 * S - k_{19}Axl2 * S - k_{20}Axl2 * Sc - k_{22}Axl2$$
$$+ k_{23}Axl2c * BemGEF42 + Dm\nabla^2 Axl2 \tag{25}$$

$$\frac{\partial Axl2S}{\partial t} = k_{19}Axl2 * S + k_{20}Axl2 * S_c - k_{15}Axl2 * S - k_{22}Axl2$$
$$+ Dm\nabla^2 Axl2S \tag{26}$$

$$\frac{\partial S}{\partial t} = k_{15}Axl2 * S - (2k_{16}S - k_{17}P)S - k_{21}S - k_{19}Axl2 * S$$
$$+ k_{18}P + k_{25}Sc * X + Dm\nabla^2 S \tag{27}$$

**Table 5. Exocytosis model parameters for the SBE and SBER models.**

| Parameter | Value | Interpretation | Reference |
|---|---|---|---|
| $\alpha$ | 0.5 | Mobile membrane fraction | [Gerganova et al, 2021] |
| $\gamma$ | 1.0 | Ratio of lipid to protein velocity | [Gerganova et al, 2021] |
| $r_{exo}$ | 50 nm | Exosome radius | [Layton et al, 2011] |
| $\lambda$ | 0.4/s | Exocytosis rate | This work, based on; [Layton et al, 2011; Okada et al, 2013; Gerganova et al, 2021]. |

$$\frac{\partial P}{\partial t} = (2k_{16}S - k_{17}P)S - k_{18}P + D_p\nabla^2 P \tag{28}$$

$$\frac{\partial X}{\partial t} = -k_{24}X + D_p\nabla^2 X \tag{29}$$

$$\frac{\partial GapS}{\partial t} = k_{12a}GapSc*P - k_{12b}GapS + D_m\nabla^2 GapS \tag{30}$$

$$\frac{dCdc42I}{dt} = \frac{\eta}{A_m}\int_{A_m}(k_{5a}Cdc42I - k_{5b}Cdc42D) \tag{31}$$

$$\frac{dBemGEF_c}{dt} = \frac{\eta}{A_m}\int_{A_m}(k_{1a}BemGEF_c - k_{1b}BemGEF_m \\ - k_7 BemGEFc*Cdc42T) \tag{32}$$

$$\frac{dGapSc}{dt} = \frac{\eta}{A_m}\int_{A_m}(-k_{12a}GapSc*P + k_{12b}GapS) \tag{33}$$

$$\frac{dAxl2c}{dt} = \frac{\eta}{A_m}\int_{A_m}(-k_{23}Axl2c*BemGEF42 + k_{22}Axl2) \tag{34}$$

$$\frac{dSc}{dt} = \frac{\eta}{A_m}\int_{A_m}(-k_{19}Axl2*Sc - k_{25}*X*Sc + k_{21}S) \tag{35}$$

In addition to chemical reactions, the model includes exocytosis. Following experimental observations [Watson et al, 2014; Ghose et al, 2021], vesicles are modeled to deliver Cdc42-GDP at a lower concentration than the current Cdc42 total concentration in the pole (100 µM compared to around 180 µM). Thus, they effectively dilute the pole. $X$ is delivered at a concentration of 20 µM. The nodes that can be hit by exocytosis are those where the concentration of Cdc42 fulfills: $Cdc42\text{-}GTP > \varepsilon*max(Cdc42\text{-}GTP)$, where a smaller $\varepsilon$ corresponds to more diffuse exocytosis, and for each exocytosis event, one of these nodes is randomly selected.

### Exocytosis modeling

Exocytosis is modeled using the approach described in [Gerganova et al, 2021]. Here, exocytosis displaces proteins radially away from the center of exocytosis. The displacement for a molecule at an arc length (geodesic distance) $s$ from the center of exocytosis is given by:

$$\Delta r_{exo}(s) = \frac{1}{\gamma}\left(R'\arccos\arccos\left(1 - \frac{R^2}{R'^2}\left(1 - \cos\cos\left(\frac{s}{R}\right)\right) - \frac{A_{exo}}{2\pi R'^2\alpha}\right) - s\right) \tag{36}$$

$$R' = \sqrt{R^2 + \frac{A_{exo}}{4\pi}} \tag{37}$$

where $R'$ is the radius of the sphere following exocytosis, $R$ is the radius of the cell, $A_{exo}$ is the vesicle surface area, $\gamma$ and $\alpha$ are hydrodynamic parameters. Specifically, $\alpha$ represents the fraction of the fluid component in the membrane and $\gamma \leq 1$ is the ratio of lipid to protein velocity. Like in previous studies on *S. pombe*, we set $\gamma = 1$ and $\alpha = 0.5$ [Gerganova et al, 2021]. Additionally, following earlier work in *S. cerevisiae*, we use $r_{exo} = 50$ nm [Ghose et al, 2021]. A summary of parameter values can be found in Table 5.

Exocytosis recruitment is modeled as a stochastic event with a rate $\lambda$. Previous modeling on septin ring formation used a rate of 0.2/s [Okada et al, 2013]. Measurements under no growth in *S. pombe* yielded a rate of 0.5/s, and for *S. cerevisiae* during yeast polarization the rate was measured to be around 0.41/s [Layton et al, 2011], while when modeling polarity patch movement a rate of 0.83/s has been used [Ghose et al, 2021]. Considering that most estimates are larger than the value in [Okada et al, 2013], we opted for a compromise and set the rate to 0.4/s.

Applying the same principle, we can extend the model to incorporate endocytosis. However, we exclude endocytosis for three reasons. Firstly, due to the smaller size of endosomes (~1/4 the area of exosomes, resulting in a comparatively smaller impact from a single endocytosis event). Secondly, due to their wider occurrence area (thus exerting less influence on ring formation and the pole) [Layton et al, 2011]. Thirdly, due to the uncertainty regarding whether they recycle any noticeable polarity proteins (which could affect polarization). Overall, as in previous septin ring modeling [Okada et al, 2013], we made the choice to exclude endocytosis, which further reduces simulation time.

### Simple particle model

In the simple particle model (Appendix Fig. S3A), we initialize the simulation with 1000 particles randomly distributed within a pole that occupies a fraction $\Omega$ of the cell-surface area. Subsequently, we run the simulation for 20 iterations, applying exocytosis following the approach described above. Different values for the tuning parameters $\alpha$ and $\gamma$ (Table 5) were explored, and the tested parameter values are provided in the first three rows of Table 6.

Exocytosis is permitted within a cluster, denoted as $\Sigma$, which occupies a fraction $\Omega$ of the cell area (the same pole where particles start). To simulate varying degrees of diffused exocytosis, we initially generate 10,000 random points within $\Sigma$. The exocytosis site is then randomly sampled from these points, with each point weighted by:

$$\frac{1}{(d_i + 1e-8)^\beta}, \tag{38}$$

**Table 6.** Tested values for particle simulator.

| | Values | Interpretation |
|---|---|---|
| $\beta$ | 0.1–3.0 | Parameter deciding how concentrated exocytosis is |
| $R$ | 2.5 μm | Sphere radius |
| $r_{exo}$ | 50 nm | Exosome radius, see Table 5 |
| $\Omega$ | 0.01, 0.03, 0.05 | Fraction of pole area particles either start in or are recruited into, and exocytosis occurs in |
| $\alpha$ | 0.2, 0.5, 0.8 | Mobile membrane fraction, see Table 5 |
| $k_{rec}$ | 60/s | Particle recruitment rate |
| $k_{off}$ | 1.0, 0.5, 0.05 /s | Particle dissociation rate |
| $D_m$ | 0.045, 0.0045, 0.00045 $\mu m^2/s$ | Particle diffusion rate |

The values column corresponds to the tested values in simulations. Note for $\Omega$ only results for 0.03 are presented, as results were consistent for different values.

where $d_i$ represents the distance of point $i$ to the center of $\Sigma$. For small values of β, exocytosis is widespread, while larger values make it more concentrated (Appendix Fig. S3B). Alternatively, the exocytosis hit site could be modeled using a Gaussian distribution, as done on the plane in [Ghose et al, 2021]. However, deriving the Gaussian distribution on the surface of a sphere (as in our simulations) is challenging; therefore, we opted for the approach above.

### More complex particle model

In the more complex particle model (Appendix Fig. S3D), particles are randomly recruited within a cluster denoted as $\Sigma$, which occupies a fraction $\Omega$ of the cell area. Particles are recruited at the rate $k_{rec}$, they diffuse on the surface with the rate $D_m$, and disassociate with a rate $k_{off}$. Exocytosis is modeled using the same approach as in the simple particle simulator.

The simulation algorithm is custom-made. Specifically, recruitment is simulated using τ-leaping [Gillespie, 2007] with a step length of $dt = 0.01$ s. Recruited particles are randomly distributed within $\Sigma$. Diffusion is modeled with the same time step; $dt$. Once a particle is recruited to the membrane, its membrane lifetime $t_i$ is determined using the SSA (Gillespie) algorithm [Gillespie, 2007], as the particles are modeled as independent (i.e., they do not interact). Once the simulation time $t$ exceeds $t_i$, the particle is removed from the sphere surface.

Despite having fewer species and reactions than the SBER model (Fig. 5A,B), the complex particle model captures key properties of septin ring formation. Particles are recruited within a defined area (denoted as $\Sigma$, mimicking the Cdc42-GTP cluster). They undergo diffusion, dissociate from the membrane, and experience displacement via exocytosis. For the parameters used, see Table 6.

### Simulation details

To simulate the Cdc42 and septin ring models, we use a finite element method (FEM) solver in space and a finite differences solver in time using the FEniCSx software [Logg et al, 2012]. Following the logic in [Borgqvist et al, 2021], a mixed implicit-explicit Euler scheme is used in time. The nonlinear reaction terms are treated as explicit, while all linear terms and gradients are treated as implicit. This approach incorporates nonlinear terms into the load vector, allowing the FEM weights $\xi$ at time $t$ to be obtained by solving a large linear system. Since this linear system

must be solved at each time step, the choice of linear solver is crucial for simulation performance. Given that our linear system is non-symmetric, we use the GMRES (generalized minimal residual method) solver with incomplete LU factorization (ILU) as a preconditioner.

All models comprise coupled partial differential equations (PDEs) and ordinary differential equations (ODEs). We adopt a mixed-stepping approach, where: (i) the PDEs are updated using the schema above and (ii) the ODEs are then updated using a third-order Runge–Kutta method (explicit midpoint method). To ensure solution correctness and reduce runtime, we employ an adaptive time step algorithm. Additionally, to guarantee accurate solutions, we use a finite-element mesh generated in the Gmsh software [Geuzaine and Remacle, 2009] with a high node density of 9451 nodes.

Cluster area in the model is computed as in [Borgqvist et al, 2021]; by computing the fraction of nodes that fulfill $|Cdc42_{max} - Cdc42| < 0.2$ ($Cdc42_{max} - Cdc42_{min}$). Septin ring diameter is computed using the algorithm in Fig. EV4D, where the points constituting the septin ring are selected by filtering out all nodes where $|P_{max} - P| > 0.2$ $P_{max}$ (results are robust to different threshold values than 0.2). Cluster area and ring diameter are computed at steady-state (after the model has been simulated for a long time), when there is only a single Cdc42-GTP cluster. This avoids measuring the area for multiple clusters and ensures that transient model dynamics are not measured for a fair comparison.

### Computing septin ring diameter

The simulation produces a set of coordinates (points) that correspond to the septin ring, see Fig. EV4D. For each inner point (dmin—blue Fig. EV4D) the inner diameter is computed, and then the outer diameter is computed (dmax— green Fig. EV4D) for each outer point. Then the mean of all dmax and dmin is computed (right part Fig. EV4D).

### Statistics and reproducibility

All measurements are based on at least two independent biological replicates. Results from individual replicates were compared, and no major differences were noted. Samples were not blinded before analysis.

## Data availability

Yeast strains are available upon reasonable request. Source data are provided together with this paper, raw microscopy files are available here: https://www.ebi.ac.uk/biostudies/bioimages/studies/S-BIAD2257. Additional information on image analysis approaches described in "Materials and Methods" and previous publications is also available upon reasonable request. The code for the Cdc42 cluster, Exo84 cluster and septin ring quantification can be found here: https://github.com/ElpadoCan/ringQUANT. The code for model simulations and generating the plots in the figures can be found here: https://github.com/sebapersson/cdc42_and_septin_ring_paper. This code is also available as a Zenodo archive at https://doi.org/10.5281/zenodo.16779394.

The source data of this paper are collected in the following database record: biostudies:S-SCDT-10_1038-S44318-025-00571-5.

## Peer review information

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

## Acknowledgements

We thank the Nils Johnsson, Daniel Lew, Peter Novick, and Rong Li labs for sharing yeast strains, Michael Lanz and Jan Skotheim for sharing proteomics data, and Nils Johnsson and Daniel Lew for feedback on the manuscript. This work was supported by the Swedish Research Council (VR2017-05117 and VR2023-04319), the Swedish Foundation for Strategic Research (FFL15-0238) to MC, the Human Frontier Science Program (career development award to KMS), the Deutsche Forschungsgemeinschaft (DFG, German Research Foundation) through SFB 1064 (Project-ID 213249687) and SFB 1309 (Project-ID 325871075) to RS, and the Helmholtz Gesellschaft (KMS and RS). The funders had no role in study design, data collection and analysis, decision to publish, or preparation of the manuscript.

## Author contributions

**Igor V Kukhtevich**: Conceptualization; Formal analysis; Investigation; Visualization; Methodology; Writing—original draft. **Sebastian Persson**: Conceptualization; Software; Formal analysis; Investigation; Visualization; Methodology; Writing—original draft. **Francesco Padovani**: Software; Formal analysis; Investigation; Methodology; Writing—review and editing. **Robert Schneider**: Supervision; Funding acquisition; Investigation; Writing—review and editing. **Marija Cvijovic**: Conceptualization; Formal analysis; Supervision; Funding acquisition; Investigation; Project administration; Writing—review and editing. **Kurt M Schmoller**: Conceptualization; Formal analysis; Supervision; Funding acquisition; Investigation; Project administration; Writing—review and editing.

Source data underlying figure panels in this paper may have individual authorship assigned. Where available, figure panel/source data authorship is listed in the following database record: biostudies:S-SCDT-10_1038-S44318-025-00571-5.

## Funding

## Disclosure and competing interests statement

The authors declare no competing interests.

# Expanded View Figures

**Figure EV1.** **Alternative computational model of Cdc42 polarization with positive feedback and increased GAP activity and protein dilution in the positive feedback model.** ▶

(**A**) Schematic drawing of the alternative model with positive feedback from [Borgqvist et al, 2021] (left), and result from a representative simulation (right). (**B**) Cdc42-GTP cluster area measured at steady-state (after long simulation time) for the model plotted against cell volume in a double logarithmic scale. In each case, $n = 11$ cells with volumes ranging from 115 to 345 fL were simulated starting from random initial conditions. Note that the volume interval differs from Fig. 1 as the model polarizes in a different parameter regime compared to the models in Fig. 1. (**C**) Cdc42-GTP maximum concentration against cell volume for the same simulated cells as in (**B**), measured at the same time point as the cluster area. Solid lines in (**B**, **C**) show linear regression fits. (**D**, **E**) Cdc42-GTP cluster area (**D**) and Cdc42-GTP concentrations (**E**) for the positive feedback model measured at steady-state (after long simulation time) plotted against cell volume in a double logarithmic scale for normal (1) and stronger (1.8) GAP activity. The increased GAP activity mimics negative feedback by reducing Cdc42 activation. In each case, $n = 60$ (three replicates per volume) cells with volumes ranging from 65 to 270 fL were simulated starting from random initial conditions. Solid lines show loess smoothings. (**F**) Protein slope as in Fig. 2, but obtained from cell size mutants (see [Lanz et al, 2024] for more details). Bars show the mean value of $n = 3$ biological replicates. Computer icon—modeling results. Microscope icon—experimental results.

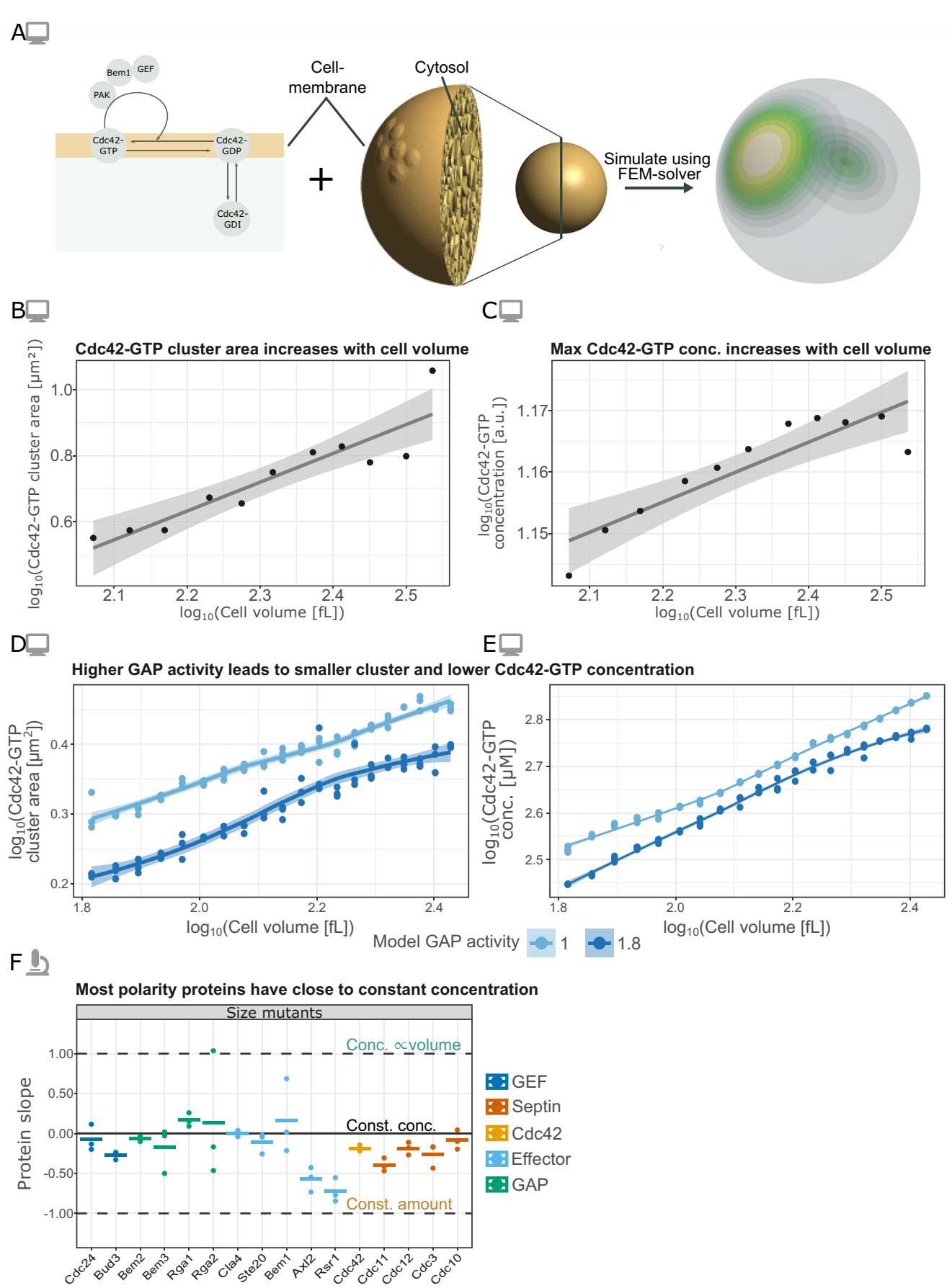

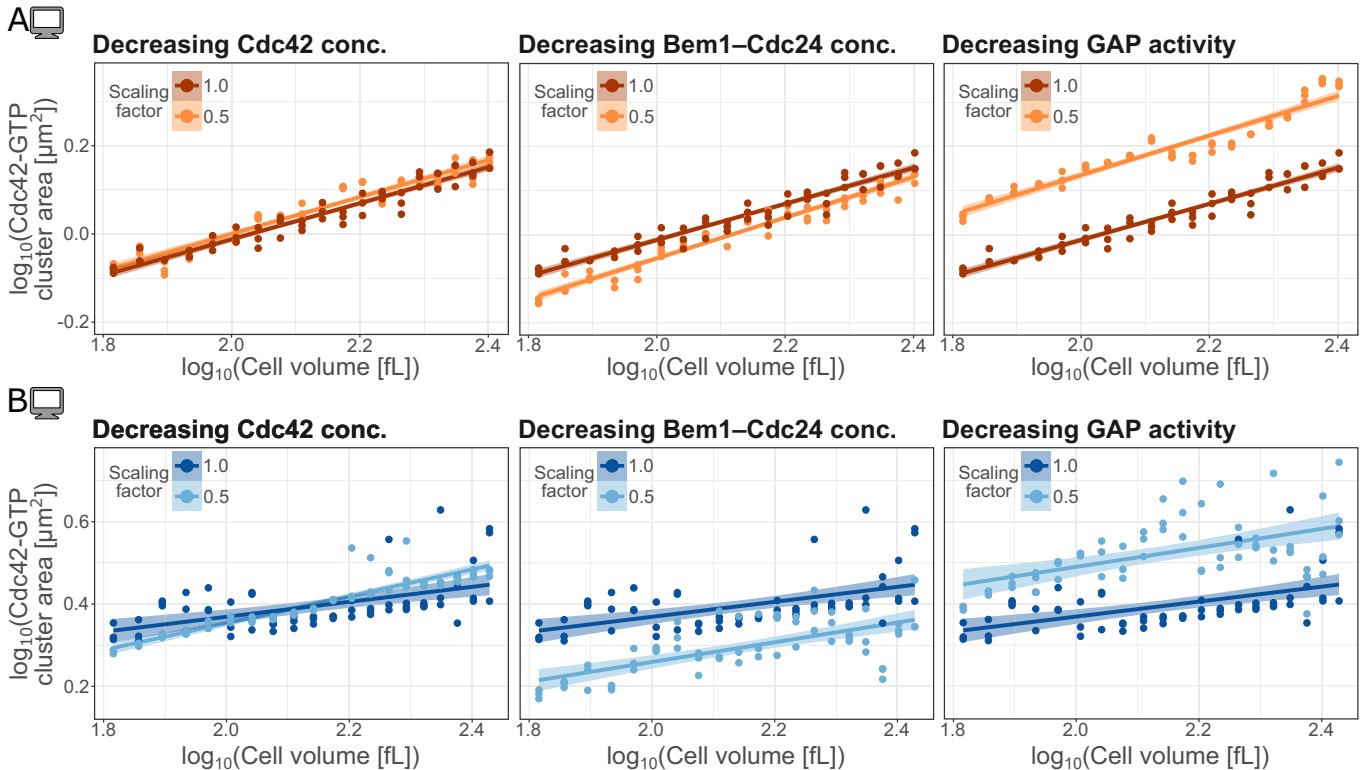

**Figure EV2. Computational modeling of Cdc42 cluster area from various perturbations.**

Reducing Cdc42 concentration (left), Bem1-Cdc24 concentration (middle) and GAP activity (right) for the negative feedback model (**A**) and the positive feedback model (**B**) with a factor of 0.5. In each case, for $n = 60$ (three replicates per volume), cells with volumes in the range of 65 to 270 fL were simulated starting from random initial conditions. Computer icon—modeling results.

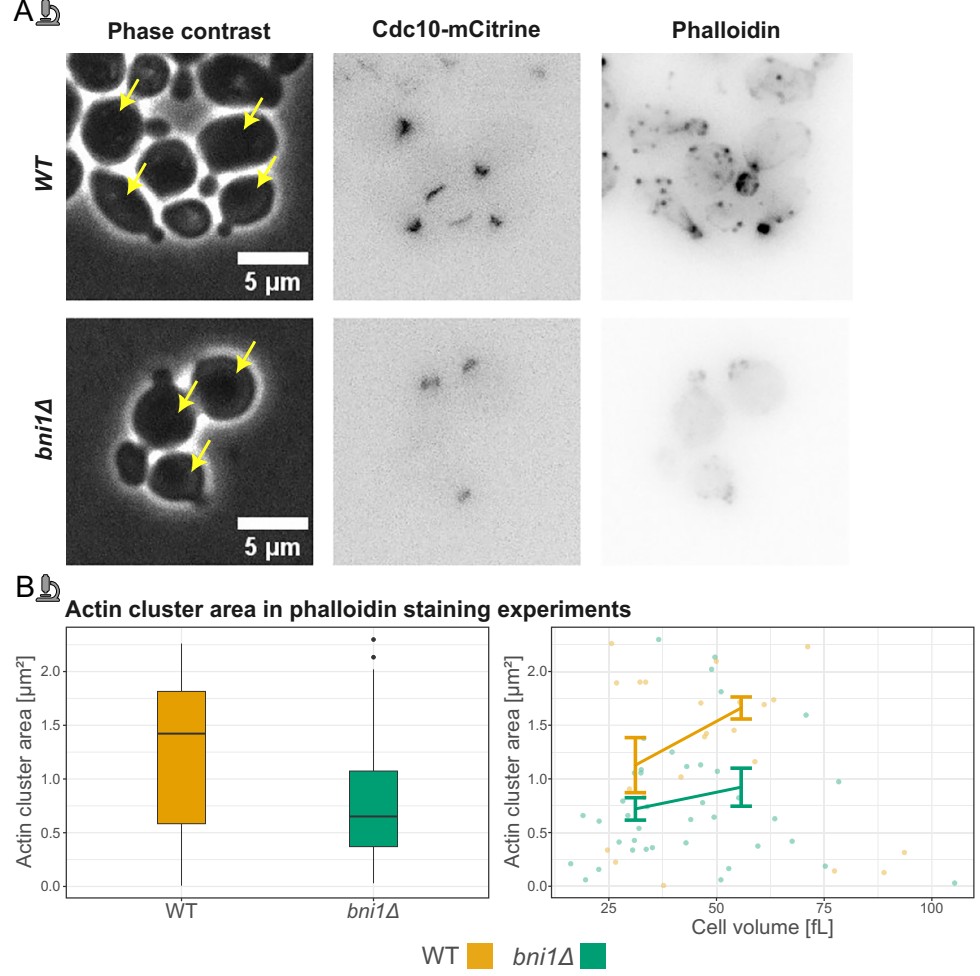

**Figure EV3.    Phalloidin-based measurements of F-actin show that the actin cluster area at the bud site is decreased in *bni1Δ* cells.**

(**A**) Representative microscopy images of budding yeast cells (phase contrast), septin ring (Cdc10-mCitrine), and F-actin (Phalloidin) for wild-type and *bni1Δ* cells. Arrows point to cells selected for analysis. (**B**) Quantification of actin cluster area at the bud site based on phalloidin staining for wild-type ($n = 23$) and *bni1Δ* ($n = 40$) cells. Left plot: the center line indicates the median; box limits show the 25th–75th percentiles (IQR); whiskers extend to the most extreme data points within 1.5× IQR; points represent outliers. Right plot: solid lines show binned means, and error bars show standard error centered at the binned mean. Two independent replicates were performed for the experiments. Microscope icon—experimental results.

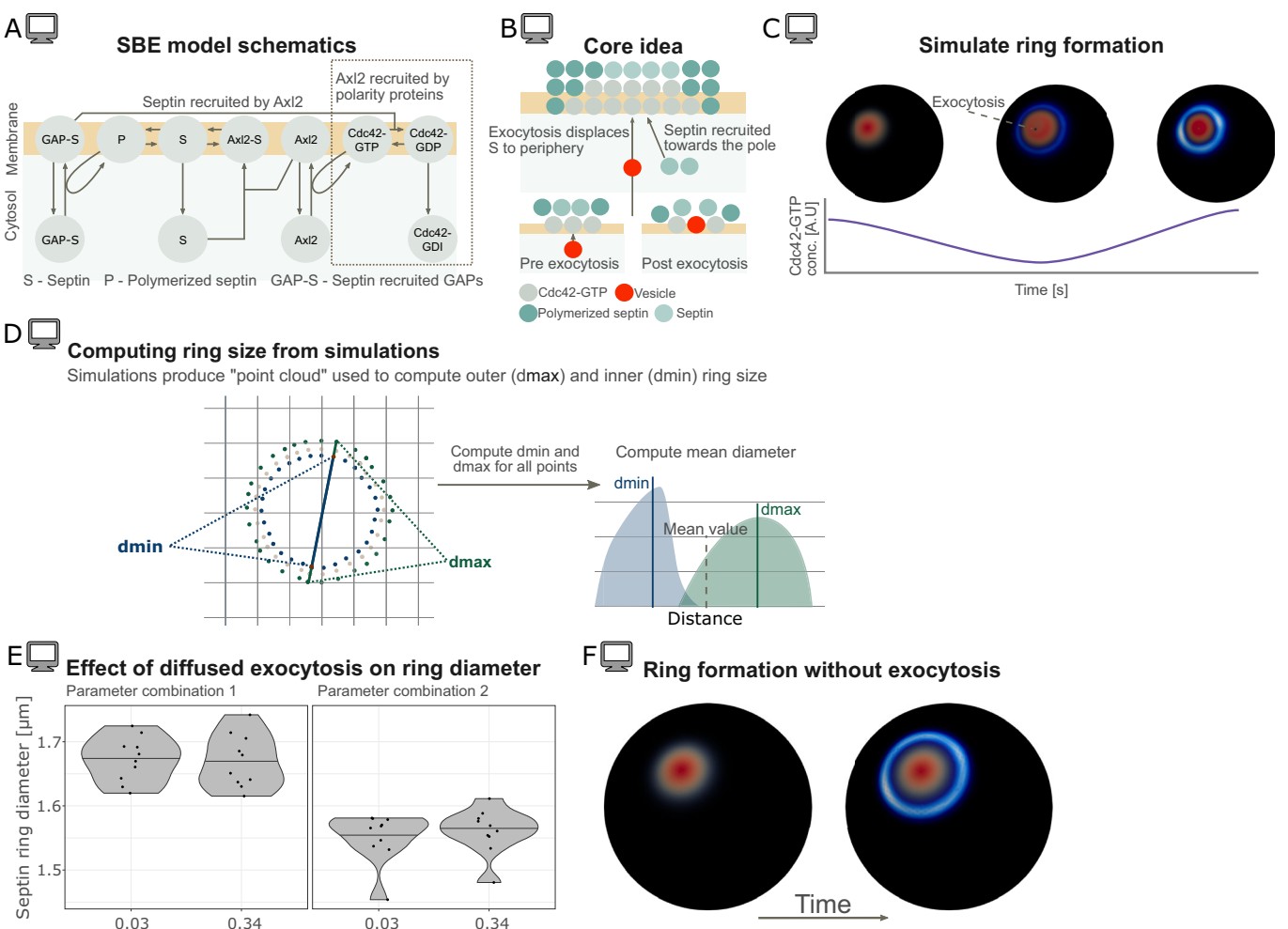

**A** SBE model schematics

**B** Core idea

**C** Simulate ring formation

**D** Computing ring size from simulations
Simulations produce "point cloud" used to compute outer (dmax) and inner (dmin) ring size

**E** Effect of diffused exocytosis on ring diameter

**F** Ring formation without exocytosis

**Figure EV4. The septin binding and exocytosis (SBE) model does not explain why septin ring size increases with diffused exocytosis.**

Model reaction schematics (**A**) and core idea (**B**) of the SBE model. Briefly, septin is recruited by Axl2, and on the membrane, septin binds to Axl2. This binding prevents polymerization in the cluster center, which promotes polymerization and, subsequently, ring formation at the cluster periphery. Additionally, exocytosis is directed towards the cluster, which further pushes septin to the periphery. (**C**) Representative example showing Cdc42 polarization (red) and consecutive septin ring formation (blue). Cdc42-GTP concentration first decreases when septin is recruited, and then increases when a stable septin ring starts to form. (**D**) Schematic explanation of how septin ring diameter is computed from model simulations. (**E**) Septin ring diameter (*dmin+dmax)/2* (see Fig. 5) for the SBE model plotted for two parameter combinations and two levels of diffused exocytosis. For each condition, $n = 10$ simulations, all starting from the same Cdc42-GTP cluster, were performed. In each case, the model was simulated for a long time to reach a stable ring, and then the septin ring diameter was measured. The number of nodes that can be hit corresponds to nodes where the concentration of Cdc42 fulfills: *Cdc42-GTP > ε\*max(Cdc42-GTP)*, where a smaller ε corresponds to more diffused exocytosis. (**F**) Example of septin ring formation for the SBER model without exocytosis. Note that without exocytosis, the SBER model is equivalent to the SBE model. Computer icon—modeling results.

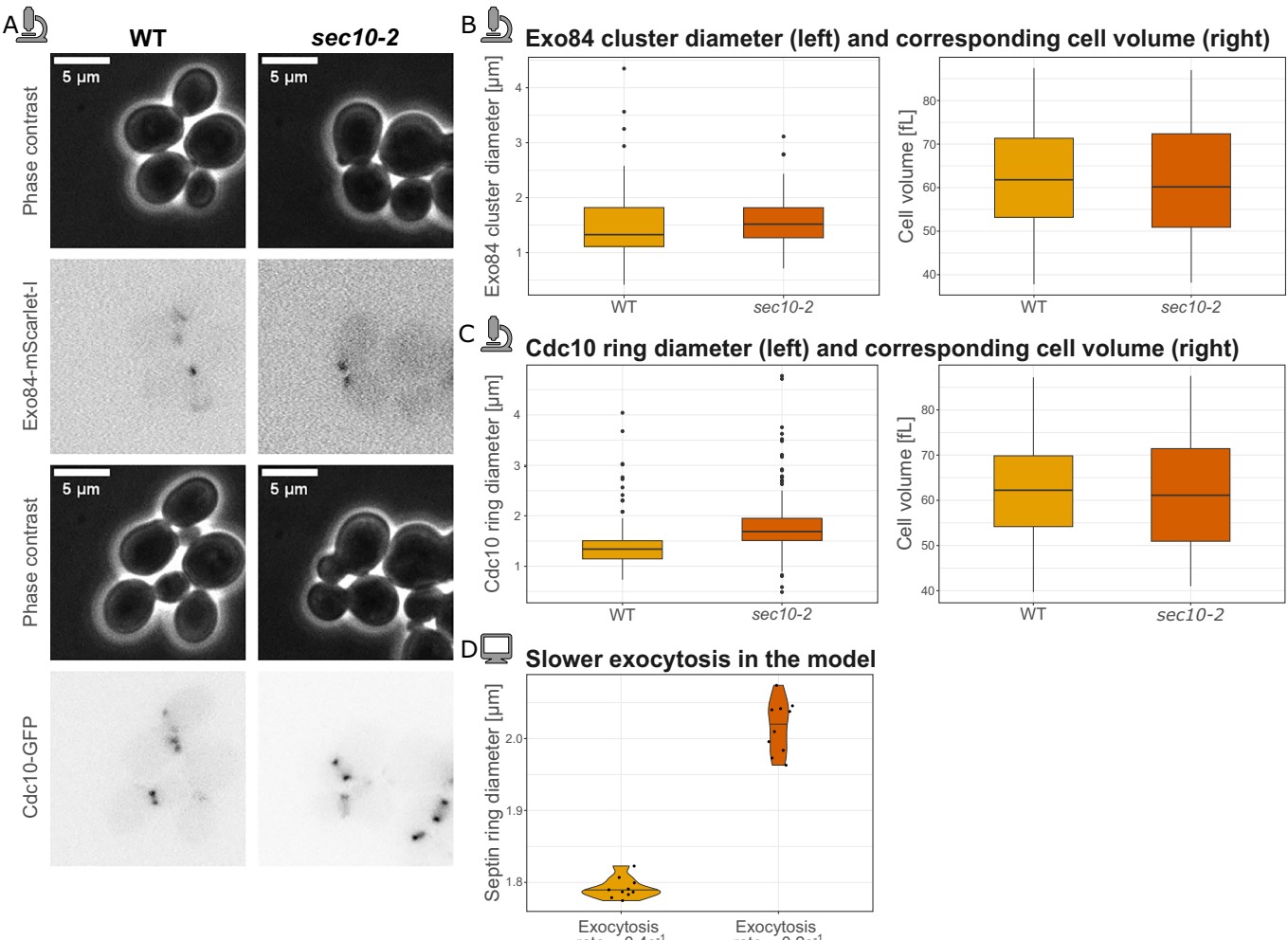

**Figure EV5. Experiments with *sec10-2* temperature-sensitive mutant and reduction of exocytosis rate in the SBER model show enlarged septin rings.**

Cells were analyzed between 170 and 410 min after shifting to the non-permissive temperature. (A) Representative microscopy images for wild-type and *sec10-2* cells. (B) Quantification of Exo84 cluster diameter and corresponding cell volume: WT ($n = 121$), *sec10-2* ($n = 80$). (C) Quantification of septin ring diameter and corresponding cell volume: WT ($n = 380$), *sec10-2* ($n = 175$). For boxplots, the center line indicates the median; box limits show the 25th–75th percentiles (IQR); whiskers extend to the most extreme data points within 1.5× IQR; points represent outliers. (D) Modeling results for normal (0.4 s$^{-1}$, $n = 10$) and reduced (0.2 s$^{-1}$, $n = 10$) exocytosis rate. Two independent replicates were performed for the experiments. Computer icon—modeling results. Microscope icon—experimental results.

