## [Peer Review File · The EMBO Journal]

The origin of septin ring size control in budding yeast

Igor Kukhtevich, Sebastian Persson, Francesco Padovani, Robert Schneider, Marija Cvijovic, and Kurt Schmoller

Corresponding authors: Kurt Schmoller (kurt.schmoller@helmholtz-munich.de) , Marija Cvijovic (marija.cvijovic@chalmers.se)

Review Timeline:

Submission Date:	15th Oct 24
Editorial Decision:	19th Dec 24
Revision Received:	16th Jun 25
Editorial Decision:	23rd Jul 25
Revision Received:	11th Aug 25
Accepted:	5th Sep 25

Editor: Ieva Gailite

Transaction Report:

Dear Kurt,

Thank you for submitting your manuscript for consideration by the EMBO Journal. We have now received comments from two reviewers, which are included below for your information.

Based on the overall interest expressed in the reports of both reviewers and your willingness to engage in a major revision as expressed in the preliminary revision plan provided during the pre-decision consultation, I would like to invite you to revise the manuscript as proposed in your point-by-point response. From the editorial side, I think that attempting to perform phalloidin stainings in response to point 2 by reviewer #1 would be beneficial, although I appreciate that the outcome might not be conclusive. I should add that it is The EMBO Journal policy to allow only a single major round of revision and that it is therefore important to resolve the main concerns at this stage.

We generally allow three months as standard revision time. Should you foresee a problem in meeting this deadline, please let me know in advance to discuss an extension. As a matter of policy, competing manuscripts published during this period will not negatively impact on our assessment of the conceptual advance presented by your study. However, please contact me as soon as possible upon publication of any related work to discuss the appropriate course of action.

When preparing your letter of response to the referees' comments, please bear in mind that this will form part of the Review Process File and will therefore be available online to the community. For more details on our Transparent Editorial Process, please visit our website: <https://www.embopress.org/page/journal/14602075/authorguide#transparentprocess>. Please also see the attached instructions for further guidelines on preparation of the revised manuscript.

Please feel free to contact me if you have any further questions regarding the revision. Thank you for the opportunity to consider your work for publication. I look forward to receiving your revised manuscript.

With best wishes,

Ieva

We realize that it is difficult to revise to a specific deadline. In the interest of protecting the conceptual advance provided by the work, we recommend a revision within 3 months (19th Mar 2025). Please discuss the revision progress ahead of this time with the editor if you require more time to complete the revisions.

Referee #1:

Kuktevich et al, explore mechanisms by which Septin rings that form at the onset of cytokinesis in budding yeast scale their size (e.g diameter) to cell volume. The MS first explores the scaling of active-Cdc42 zone areas with cell volume, and makes use of PDE-based simulations to dissect general principles of polarity patch size scaling (as has been done in many past classical work on cell polarization in budding yeast). The authors find through models and analysis of previously published data, that as polarity protein concentration remains mostly stable in changing cell volumes, polarity zones can become larger in larger cells. This scaling may change slightly if negative feedbacks are removed from models, which is confirmed by using a Cdc24 (a Cdc42 GEF) allele mutated for a key phospho-site (38A). The MS then goes on to test mechanisms of septin ring scaling. First it is shown that the formin bni1, and other components of the polarisome (bud6 and spa2), are important to scale septin ring size with cell volume; then that a proper actin cable network organization is also important for this scaling, as it promotes localized exocytosis to the bud neck. Then another model (SBER) is developed to predict septin ring size; and makes correct prediction that septin rings scale with GTP-CDC42 zones, thereby explaining septin ring scaling with cell volumes. Finally the authors show that in the Cdc24-38A, defective in negative feedback, the coupling between cdc42 patch area and septin ring size is lost, which the authors account for by changing a parameter related to polarity-factors septin recruitment. Overall, this work contains a large body of modelling development and analysis, and some good level of quantitative imaging data and analyses. The paper is often difficult to follow as it hops back and forth between different levels of modelling, analysis of previously published data, and novel experiments. In addition, the main conclusion of the paper, are mostly supported by the models, and only tested with few experiments and mutants. Hence, although I find the general question asked quite interesting and the data quite solid, I have reservation on how much the paper really provides a novel conceptual or molecular framework for how septin ring size may be controlled in budding yeast. My main comments are listed below:

1- Predictive power of models. The MS features 2 different models (SBE end SBER); which are based on solving multiple PDEs that couple the concentration of important polarity or septin regulating components. Both models contain a very high number of adjustable parameters (up to 25 constants!). Many of these parameters are adjusted in this work, with no experimental measurements. This raises an important concern on how much the model can really help to narrow down the key mechanisms. I think it would be important to explore parameter sensitivity analysis of the model in a full supplementary figure, to better explain how parameter values were chosen, and how much they may impact the key prediction of the models.

2- Imaging. Figure 4 provides a set of representative images of actin and septins. But the resolution provided prevents from really assessing the state and arrangement of actin cables around the septin ring. The authors could use high-resolution images of fixed cells marked with phalloidin, to better delineate actin cables, and support their conclusion as presented in Figure 4h.

3- Curvature vs Volume. Previous works, that are not cited, have investigated how polarity zones and septin assembly may depend on cell geometry. Among these, PMID: 31579399 is an important MS to discuss as it already explores how reaction-diffusion models can scale polarity zones to changing cell sizes. Also, there has been previous work in yeast spores (PMID: 26441355) that propose that active-Cdc42 zones adapt their size to local curvature of the cell surface, this hypothesis should be discussed, as cells considered in the current MS also have particular morphologies and distribution of local curvatures. Along this line, another important work has shown how local curvature impact the recruitment and size of septin domains in fungi (PMID: 27044896); this should be discussed as well.

Referee #2:

In this study entitled "The origin of septin ring size control in budding yeast", Kukhtevich and colleagues investigate the mechanism by which cells increase the size of their mitotic septin ring to scale with cell size, following up on work from their lab in 2020. The authors use a combination of mathematical modeling and experimental yeast cell biology work to consider the impact of scaling cell size on two major features: 1) the size of the Cdc42 polarity patch (Cdc42 and its associated machinery) and 2) the size of the mitotic septin ring. They conclude from their modeling that the Cdc42 positive feedback is sufficient to drive increased polarity patch size in a model, as long as protein concentrations stay the same as cell size increases. This necessitates that protein expression increase with cell size for at least a subset of the Cdc42 polarity machinery. They find that negative feedback to Cdc42 polarization limits polarity patch size in the model, while mutations to negative feedback of Cdc24 similarly increase the size of the polar patch in cells. They then turn their attentions to control of the size of the septin ring formed during mitosis. This ring structure is regulated by Cdc42 and scales with cell size, much like the polarity patch. Interestingly, experimental disruption of the polarisome caused the septin ring to get larger, contrary to the predictions one would make based on the model by the Goryachev and Bi labs (Okada et al. 2013). The authors find that focused exocytosis is required to create a normal sized septin ring, and that defocused exocytosis leads to broader septin structures. The authors then used modeling to explore what mechanisms could explain this behavior and find that coupling septin recruitment to exocytosis leads to the connection between defocused exocytosis and increased ring size. They then find that decreasing the negative feedback to the Cdc42 polarity machinery leads to increased polarity patch size without increasing the septin ring size. Modeling suggests that this decoupling depends upon septins having a high rate of recruitment to the site of polarity. There are many things to like about this study. The modeling is excellent, and the cell biology is sophisticated and well done. The questions being asked are interesting, and I think the answers will be important to our understanding of septin regulation and cell size control. I particularly like the idea that exocytosis recruits septins, and that they have modeling that suggests this could be true.

I have reservations, however. They can be summarized by considering that the title would be stronger if it were "The origin of septin size control in budding yeast is Cdc42 positive feedback, protein concentration scaling, and exocytic delivery of septins". Unfortunately, the work at hand can't fully support that claim, so the authors, rightfully, didn't say it. Many of the connections between the modeling and the biology are missing. I can see why this is the case, as choosing the correct experiments to validate the model could be difficult. The writing and storytelling could be a bit clearer as well. This paper was more difficult to read in its current form than it should have been.

Ultimately, if some of these gaps between what the models predict and what we know happens in the cell were filled through wet lab experimentation, and the manuscript was rewritten with regards to clarity, especially emphasizing what the modeling tells us versus what we know happens in the biological system, this work would be suitable for publication in EMBOJ.

Major Issues

1) The authors note on page 4 that "...positive feedback is sufficient to establish increased Cdc42-GTP cluster area with cell volume." However they later point out that this only works if the protein concentrations stay the same in the cytosol. That would seem to mean that positive feedback is necessary, but not sufficient, for the increase. The authors also do not really address the mechanism underlying this. It would seem, from my reading of it, that this has to be due to the difference in the polarity patch being on a 2D surface that scales in a squared relationship ($4\pi r^2$) with cell size while the cytosol scales with a cubed relationship to cell size ($4/3 \pi r^3$). If the protein concentrations stay the same, then recruitment diminishes the cytosolic concentration less as the cell increases in size. Ultimately, something must be limiting, and increasing the size diminishes the impact of that limitation. The authors state "This suggests that the Cdc42-GTP cluster area increases with cell volume because the diffusion-driven flux gets stronger due to higher concentration in the cluster center." But this sentence doesn't make much sense, how does flux get bigger if the concentration stays the same? How is flux impacted by cell size? The authors should really address the underlying logic here. When the cell gets larger and produces more of a protein to maintain concentration, then there is more of that protein left in the cytosol when it is used on the membrane. As we move on from the model, the experimental work to back this up doesn't really address the issue of protein levels. Disruption of positive feedback is problematic for measuring the polarity patch, so I can understand not doing that, but if they are correct and something is limiting for polar patch growth, then increasing that factor should increase polar patch size. Disrupting negative feedback gets us part of the way there, but the conclusion the authors come to is that it is about protein levels, and that is not directly tested. Their analysis of the SILAC data from Lanz et al. is interesting, but should be used to choose candidates for over expression. If the authors could show that a specific protein that stays the same or increases is sufficient to drive larger Cdc42 polarity patch, their conclusions would be much stronger. Again, I understand that a negative result in this sort of experiment would not mean much, as it is possible that there are multiple proteins that will need to be over expressed together, but to truly plant a flag in the sand about protein levels being sufficient, I feel like there needs to be additional biological proof. Bem1 may be the limiting factor for polar patch formation, so it would be an obvious one to try (Johnson et al. 2011). Gic2 goes up significantly, I would think that would also be a good candidate.

2) An experimental verification that septins are somehow recruited by exocytosis would strengthen those conclusions a great deal. I strongly suspect that the authors are correct about this hypothesis. I can see how testing this could be difficult. I wonder if a temperature sensitive exocyst component might allow you to accumulate septins at sites of exocytosis? Defects in exocytosis clearly make the septin ring larger, and I am intrigued by the effects of mutating the Cdc24-Cdc11 interaction on rate. But the

take-home message from the authors is that the mechanism of this is through exocytic recruitment of septins; we need some sort of experimental proof that is true.

3) Conclusions are not always supported by the results. The authors should pay special attention to talking about conclusions supported by modeling versus conclusions supported by the biology. For example: "Exocytosis-aided recruitment of septins explains the increase of septin ring size upon diffused exocytosis" is found on page 13. What the authors show is that diffuse exocytosis changes ring size, and mathematical modeling suggests that exocytosis aided recruitment could explain that. They do not show that exocytosis-aided recruitment happens in cells (see major concern 2). There needs to be much more careful wording around these conclusions. Discussions of the role of protein levels suffers from the same issue.

4) The English and grammar and individual parts of the paper are well written, but overall the clarity is lacking. The narrative can be hard to follow and the switching between models and experiments can be hard to keep track of. This is just an issue of text changes though, and I am sure the authors can make this better.

Minor Issues

1) Authors include cell numbers for experiments, but do not indicate how many independent experiments these are from.

2) In the introduction, the following is stated with no reference "However, the septin ring diameter can be decoupled from the cluster area by deleting the formin BNI1, which leads to a larger septin ring, even though the Cdc42 cluster area is mostly unchanged." It must be referring to Kuhktevich et al. 2020, but as this is a major undertone of the current work, it would be useful to emphasize that you have previously found this to be true and are now going to investigate it further, as this finding is also a component of figure 3. I think this data being in figure 3 is fine, especially with the additional formin associated proteins being tested as well, it should just be more explicitly stated.

3) Figure 1 looks pixelated, hopefully this is just an artifact of the PDF conversion for the submission system.

4) For figure 1F, how were thresholds chosen? Is it in an unbiased way? A particular algorithm?

5) Is "sublinearly" the correct description here? It appears that you mean a lower slope, but the data still appears to be linear?

6) Why is Gic2 in the SCGE data set and not the SCD data set? Given the robust change in Gic2 in SCGE, should you also check its paralogue Gic1?

7) In the septin model in figure 5, what causes septins to convert to polymerized?

8) I don't follow why increased PSR decouples septin size from Cdc42 patch size when the Cdc42 patch is what is putting those Polarity factors in place. Is there a significant different in turn over rate? Are the polarity factors limiting, such that more cdc42 machinery doesn't lead to more polarity factors being present?

9) Figure 7 does not leave me feeling like I fully understand what you have shown. It feels more like a summary of results than of conclusions.

Referee #1 (Report for Author)

Kuktevich et al, explore mechanisms by which Septin rings that form at the onset of cytokinesis in budding yeast scale their size (e.g diameter) to cell volume. The MS first explores the scaling of active-Cdc42 zone areas with cell volume, and makes use of PDE-based simulations to dissect general principles of polarity patch size scaling (as has been done in many past classical work on cell polarization in budding yeast). The authors find through models and analysis of previously published data, that as polarity protein concentration remains mostly stable in changing cell volumes, polarity zones can become larger in larger cells. This scaling may change slightly if negative feedbacks are removed from models, which is confirmed by using a Cdc24 (a Cdc42 GEF) allele mutated for a key phospho-site (38A). The MS then goes on to test mechanisms of septin ring scaling. First it is shown that the formin bni1, and other components of the polarisome (bud6 and spa2), are important to scale septin ring size with cell volume; then that a proper actin cable network organization is also important for this scaling, as it promotes localized exocytosis to the bud neck. Then another model (SBER) is developed to predict septin ring size; and makes correct prediction that septin rings scale with GTP-CDC42 zones, thereby explaining septin ring scaling with cell volumes. Finally the authors show that in the Cdc24-38A, defective in negative feedback, the coupling between cdc42 patch area and septin ring size is lost, which the authors account for by changing a parameter related to polarity-factors septin recruitment.

Overall, this work contains a large body of modelling development and analysis, and some good level of quantitative imaging data and analyses. The paper is often difficult to follow as it hops back and forth between different levels of modelling, analysis of previously published data, and novel experiments. In addition, the main conclusion of the paper, are mostly supported by the models, and only tested with few experiments and mutants. Hence, although I find the general question asked quite interesting and the data quite solid, I have reservation on how much the paper really provides a novel conceptual or molecular framework for how septin ring size may be controlled in budding yeast. My main comments are listed below:

1- Predictive power of models. The MS features 2 different models (SBE end SBER); which are based on solving multiple PDEs that couple the concentration of important polarity or septin regulating components. Both models contain a very high number of adjustable parameters (up to 25 constants!). Many of these parameters are adjusted in this work, with no experimental measurements. This raises an important concern on how much the model can really help to narrow down the key mechanisms. I think it would be important to explore parameter sensitivity analysis of the model in a full supplementary figure, to better explain how parameter values were chosen, and how much they may impact the key prediction of the models.

We thank the reviewer and agree that, even though we discuss parameter selection in the Methods, a sensitivity analysis is necessary to test the robustness of our results. We performed this analysis only for the SBER model because (i) the SBE model is a subset of the SBER model (every reaction and parameter in SBE also appears in SBER), (ii) our main conclusions rely on the SBER model, and (iii) each simulation takes more than 150 hours

per computer core. Further, the robustness of the SBE model is also supported by the particle models in the Supplementary Material, where extensive parameter scans were possible thanks to these models being faster to simulate.

In the SBER model, the 11 parameters of the Cdc42 module were directly taken from the literature (based on experiments). Multiple studies have shown that the Cdc42 module is robust to parameter variation (Goryachev & Pokhilko 2008; Howell et al. 2011; Chiou et al. 2021; Ghose et al. 2021). We therefore centred our sensitivity analysis on the 13 parameters of the septin module (Table 4). For each septin parameter, we tested robustness by scaling its value by 0.5, 0.75, 1.5, and 2.0, and then measured the model septin ring diameter.

The figure Fig. R1 below summarizes the two most extreme perturbations (each parameter was scaled by 0.5 or 2.0). A septin ring formed in 25 of 26 runs; the sole failure occurred when k_{15} was doubled, likely because the higher dissociation rate between polarity proteins and septin allowed excess septin to polymerize at the cluster center, which led to GAP recruitment and subsequent collapse of the Cdc42 cluster. In the 25 successful simulations, diffused exocytosis produced a larger septin ring in 23 cases. The impact of diffused exocytosis on ring diameter varies with the perturbed parameter, likely reflecting competition between the model's two ring-promoting mechanisms. For example, when polarity factors bind and inhibit septin strongly (low k_{22}), exocytosis (the second ring-forming mechanism) has little influence, whereas with weaker binding (high k_{22}), exocytosis has a larger effect. Regardless, this shows the robustness of our model predictions, and the figure below has been added to the Supplementary Information as Fig. S10.

a

Each parameter decreased with factor 0.5

b

Each parameter increased with factor 2.0

Figure R1. Sensitivity analysis for the SBER model. Normalized model ring diameter for the concentrated exocytosis case when perturbing each septin-module parameter by a factor of 0.5 (a) or 2.0 (b). A septin ring failed to form in panel (a) for k15. The reaction each parameter is associated with can be found in Tab. 4. k12b sets the dissociation rate of septin-recruited GAPs from the membrane; k13 the hydrolysis strength of septin-recruited GAPs; k15 the dissociation strength of septin monomers from polarity proteins; k16 and k18 the reaction strengths of septin polymerization; k19 and k20 the binding strengths of septin monomers to polarity proteins (with k20 corresponding to cytosolic septin binding); k21 and k22 the dissociation strengths from the membrane of septin and septin-bound polarity factors; k23 the recruitment strength of polarity proteins that recruit septin; k24 the dissociation strength from the membrane of the vesicle-delivered septin-recruiting protein; and k25 the recruitment strength of septin via this vesicle-delivered protein. Computer icon - modeling results.

2- Imaging. Figure 4 provides a set of representative images of actin and septins. But the resolution provided prevents from really assessing the state and arrangement of actin cables around the septin ring. The authors could use high-resolution images of fixed cells marked with phalloidin, to better delineate actin cables, and support their conclusion as presented in Figure 4h.

We agree with the reviewer that, in principle, it would be valuable to visualize and quantify individual actin cables around the (presumptive) bud site during the time of septin ring formation. In fact, we already performed phalloidin stains in cells with fluorescently tagged Cdc10 before submission of the initial manuscript. While we were able to visualize actin cables, we realized that a meaningful quantification of actin cable number and structure around the bud site is not feasible. Moreover, phalloidin stains in fixed cells come with the inherent disadvantage that we do not have time information, and we can therefore not directly identify cells at the moment of septin ring formation. For these reasons, we then opted for the Abp140-based live-cell strategy shown in Fig. 4.

Despite the limitations mentioned above, we performed new experiments with phalloidin for wild-type and *bni1Δ* cells as the reviewer requested. Our latest results were similar to the earlier acquired results (Fig. R2a) and did not allow a meaningful quantification of actin cable number. However, we were able to quantify the actin cluster area around the bud site to verify the result shown in Fig. 4b in the manuscript. Because this needs to be done in fixed cells, rather than quantifying actin cluster area at peak localization, we decided to classify cells based on septin localization (visible formed septin ring) and bud size (small bud, i.e. close to bud emergence) and only include these cells in the analysis. Our phalloidin-based observations on the actin cluster size at the bud site (Fig. R2b) are in line with the results we acquired using Abp140-mCitrine for F-actin visualization in live cells (Fig. 4b), supporting our conclusion that F-actin is less localized at the bud site in *Bni1del* cells. These new results are now added to the SI as Figure S6.

Figure R2. Phalloidin-based measurements of F-actin show that the actin cluster area at the bud site is decreased in *bni1Δ* cells. (a) Representative microscopy images of budding yeast cells (phase contrast), septin ring (Cdc10-mCitrine) and F-actin (Phalloidin) for wild-type and *bni1Δ* cells. Arrows point to cells taken for analysis. (b) Quantification of actin cluster area at the bud site based on phalloidin staining for wild-type ($n = 23$) and *bni1Δ* ($n = 40$) cells. Two independent replicates were performed for the experiments. Microscope icon - experimental results.

3- Curvature vs Volume. Previous works, that are not cited, have investigated how polarity zones and septin assembly may depend on cell geometry. Among these, PMID: 31579399 is an important MS to discuss as it already explores how reaction-diffusion models can scale polarity zones to changing cell sizes. Also, there has been previous work in yeast spores (PMID: 26441355) that propose that active-Cdc42 zones adapt their size to local curvature of the cell surface, this hypothesis should be discussed, as cells considered in the current MS also have particular morphologies and distribution of local curvatures. Along this line,

another important work has shown how local curvature impact the recruitment and size of septin domains in fungi (PMID: 27044896); this should be discussed as well.

We thank the reviewer for highlighting PMID: 31579399. This study investigates why smaller cells fail to establish a stable cluster. Among other models, using a minimal one-species model of Cdc42 polarization, they show that below a critical cell size, diffusion dominates and produces a single cluster spanning the entire membrane (hence, small cells cannot polarize). In contrast, our models operate well outside this diffusion-dominated regime, as our parameters are tuned to generate small, localized Cdc42 clusters, in line with experimental results. More relevant to our work, they report that an increase in all reaction rate constants produces larger clusters (Fig. 1h in PMID: 31579399). In our work, we show more specifically with both modelling and experiments that enhancing positive feedback, either by removing a negative feedback loop or by reducing GAP activity to boost Cdc42-GTP production, drives cluster enlargement (and in their simple model, increasing all parameters has the same effect, stronger positive feedback strength). Therefore, in the manuscript, we now discuss that our finding that positive-feedback enhancement increases cluster size is consistent with these earlier observations. Further, we have added this reference in the Introduction to highlight the importance of proper cluster-size regulation in different species.

The reviewer is correct, *a priori*, local cell curvature could also play an important role in Cdc42 and septin organization in budding yeast. To investigate this, we have already extensively tested this possibility in our previous work (Kukhtevich 2020, Fig. 8 and Fig. 9, “Two-variable linear regressions” in the Methods section), through combining single-cell analysis and cell geometry mutants. We found that once accounting for the effect of cell volume, cell curvature does not seem to play an important role in budding yeast, in contrast to what was found previously for fission yeast. Now we discuss this important point in the revised manuscript in the discussion.

Figure 8 (Kukhtevich et al., Nature Communication, 2020):

Figure 9 (Kukhtevich et al., Nature Communication, 2020):

Referee #2 (Report for Author)

In this study entitled "The origin of septin ring size control in budding yeast", Kukhtevich and colleagues investigate the mechanism by which cells increase the size of their mitotic septin ring to scale with cell size, following up on work from their lab in 2020. The authors use a combination of mathematical modeling and experimental yeast cell biology work to consider the impact of scaling cell size on two major features: 1) the size of the Cdc42 polarity patch (Cdc42 and its associated machinery) and 2) the size of the mitotic septin ring. They conclude from their modeling that the Cdc42 positive feedback is sufficient to drive increased polarity patch size in a model, as long as protein concentrations stay the same as cell size increases. This necessitates that protein expression increase with cell size for at least a subset of the Cdc42 polarity machinery. They find that negative feedback to Cdc42

polarization limits polarity patch size in the model, while mutations to negative feedback of Cdc24 similarly increase the size of the polar patch in cells. They then turn their attentions to control of the size of the septin ring formed during mitosis. This ring structure is regulated by Cdc42 and scales with cell size, much like the polarity patch. Interestingly, experimental disruption of the polarisome caused the septin ring to get larger, contrary to the predictions one would make based on the model by the Goryachev and Bi labs (Okada et al. 2013). The authors find that focused exocytosis is required to create a normal sized septin ring, and that defocused exocytosis leads to broader septin structures. The authors then used modeling to explore what mechanisms could explain this behavior and find that coupling septin recruitment to exocytosis leads to the connection between defocused exocytosis and increased ring size. They then find that decreasing the negative feedback to the Cdc42 polarity machinery leads to increased polarity patch size without increasing the septin ring size. Modeling suggests that this decoupling depends upon septins having a high rate of recruitment to the site of polarity. There are many things to like about this study. The modeling is excellent, and the cell biology is sophisticated and well done. The questions being asked are interesting, and I think the answers will be important to our understanding of septin regulation and cell size control. I particularly like the idea that exocytosis recruits septins, and that they have modeling that suggests this could be true.

I have reservations, however. They can be summarized by considering that the title would be stronger if it were "The origin of septin size control in budding yeast is Cdc42 positive feedback, protein concentration scaling, and exocytic delivery of septins". Unfortunately, the work at hand can't fully support that claim, so the authors, rightfully, didn't say it. Many of the connections between the modeling and the biology are missing. I can see why this is the case, as choosing the correct experiments to validate the model could be difficult. The writing and storytelling could be a bit clearer as well. This paper was more difficult to read in its current form than it should have been.

Ultimately, if some of these gaps between what the models predict and what we know happens in the cell were filled through wet lab experimentation, and the manuscript was rewritten with regards to clarity, especially emphasizing what the modeling tells us versus what we know happens in the biological system, this work would be suitable for publication in EMBOJ.

Major Issues

1) The authors note on page 4 that "...positive feedback is sufficient to establish increased Cdc42-GTP cluster area with cell volume." However they later point out that this only works if the protein concentrations stay the same in the cytosol. That would seem to mean that positive feedback is necessary, but not sufficient, for the increase. The authors also do not really address the mechanism underlying this. It would seem, from my reading of it, that this has to be due to the difference in the polarity patch being on a 2D surface that scales in a squared relationship ($4\pi r^2$) with cell size while the cytosol scales with a cubed relationship to cell size ($\frac{4}{3}\pi r^3$). If the protein concentrations stay the same, then recruitment diminishes the cytosolic concentration less as the cell increases in size. Ultimately, something must be limiting, and increasing the size diminishes the impact of that limitation.

The authors state "This suggests that the Cdc42-GTP cluster area increases with cell volume because the diffusion-driven flux gets stronger due to higher concentration in the cluster center." But this sentence doesn't make much sense, how does flux get bigger if the concentration stays the same? How is flux impacted by cell size? The authors should really address the underlying logic here. When the cell gets larger and produces more of a protein to maintain concentration, then there is more of that protein left in the cytosol when it is used on the membrane. As we move on from the model, the experimental work to back this up doesn't really address the issue of protein levels. Disruption of positive feedback is problematic for measuring the polarity patch, so I can understand not doing that, but if they are correct and something is limiting for polar patch growth, then increasing that factor should increase polar patch size. Disrupting negative feedback gets us part of the way there, but the conclusion the authors come to is that it is about protein levels, and that is not directly tested. Their analysis of the SILAC data from Lanz et al. is interesting, but should be used to choose candidates for over expression. If the authors could show that a specific protein that stays the same or increases is sufficient to drive larger Cdc42 polarity patch, their conclusions would be much stronger. Again, I understand that a negative result in this sort of experiment would not mean much, as it is possible that there are multiple proteins that will need to be over expressed together, but to truly plant a flag in the sand about protein levels being sufficient, I feel like there needs to be additional biological proof. Bem1 may be the limiting factor for polar patch formation, so it would be an obvious one to try (Johnson et al. 2011). Gic2 goes up significantly, I would think that would also be a good candidate.

We thank the reviewer for pointing out that our explanation of potential mechanisms behind cluster area scaling was unclear. The reviewer is correct that a key factor driving cluster scaling is cell size. Membrane surface area scales quadratically while cytosolic volume scales cubically. Thus, in larger cells, more polarity proteins can be recruited to the membrane without depleting the cytosolic concentration of polarity proteins, given constant protein concentration. Because Cdc42 recruits its own effectors through positive feedback, larger cells therefore concentrate more polarity Cdc42-GTP at the cluster center. This should create a stronger membrane diffusion flux of Cdc42-GTP, since diffusion is driven by concentration gradients. This line of reasoning predicts, and our modeling confirms, that perturbing the Cdc42-GTP to Cdc42-GDP conversion by reducing GAP activity or removing the negative feedback should produce a larger cluster with increased Cdc42-GTP concentration. We have substantially revised the text to communicate this better.

We also thank the reviewer for the suggestion to more closely investigate the effect of changes in the concentration of polarity proteins on Cdc42 cluster area and septin ring diameter. While increasing the concentrations of individual, potentially limiting, proteins may increase the cluster size, other proteins may also become limiting and thereby prevent a pronounced effect. By contrast, reducing the concentration of the most limiting protein should make it even more limiting. Thus, rather than overexpressing key polarity proteins, we investigated a series of hemizygous deletion mutants. Specifically, we determined the effect of deleting one allele of *CDC42*, *CDC24*, *BEM1*, *BEM2* and *GIC2* (one at a time), on Cdc42 cluster area and septin ring diameter in diploid cells (Fig. R3). We found that *GIC2* hemizygotes had slightly smaller clusters, indicating that protein abundance indeed regulates cluster area. Importantly, in *BEM2* hemizygotes with reduced GAP concentration, we observed a noticeably larger Cdc42-GTP cluster, in line with our modelling.

To complement our experiments, we also performed additional modelling (Fig. R4). As we already showed before, simultaneously reducing both Cdc42 and Bem1–Cdc24 concentrations resulted in smaller clusters (Fig. 2b). However, when reducing the concentrations individually, limiting Cdc42 concentration had little impact, while reducing Bem1–Cdc24 alone decreased cluster area, although for the negative feedback model less drastically than GAP disruption. This supports the idea that the abundance of positive-feedback proteins regulates cluster size, but experimentally reducing individual proteins may yield minor effects due to redundancy.

All this can be found in a significantly revised main text of the manuscript and Figures 2g, 6c,d, S4 and S5.

Figure R3. Experimental results for *CDC42*, *CDC24*, *BEM1*, *GIC2*, *BEM2* hemizygote mutants and WT. (a) Representative microscopy images for all stains tested. (Top) Quantification of Cdc42-GTP cluster area and corresponding cell volumes for all strains: WT (n = 198), *CDC42/cdc42 Δ* (n = 52), *CDC24/cdc24 Δ* (n = 131), *BEM1/bem1 Δ* (n = 147), *GIC2/gic2 Δ* (n = 100), *BEM2/bem2 Δ* (n = 54) (Bottom) Quantification of septin ring diameter and corresponding cell volume for all tested strains: WT (n = 136), *CDC42/cdc42 Δ* (n = 46),

CDC24/cdc24Δ (n = 122), *BEM1/bem1Δ* (n = 108), *GIC2/gic2Δ* (n = 89), *BEM2/bem2Δ* (n = 43). At least two independent replicates were performed for the experiments. Microscope icon - experimental results.

Figure R4. Computational modelling Cdc42 cluster area from various perturbations. Reducing Cdc42 concentration (left), Bem1-Cdc24 concentration (middle) and GAP activity (right) for the negative feedback model (a) and the positive feedback model (b) with a factor 0.5. In each case, for n=60 (three replicates per volume) cells with volumes in the range of 65 to 270 fL were simulated starting from random initial conditions.

2) An experimental verification that septins are somehow recruited by exocytosis would strengthen those conclusions a great deal. I strongly suspect that the authors are correct about this hypothesis. I can see how testing this could be difficult. I wonder if a temperature sensitive exocyst component might allow you to accumulate septins at sites of exocytosis? Defects in exocytosis clearly make the septin ring larger, and I am intrigued by the effects of mutating the Cdc24-Cdc11 interaction on rate. But the take-home message from the authors is that the mechanism of this is through exocytic recruitment of septins; we need some sort of experimental proof that is true.

As suggested by the reviewer, we tested the *sec10-2* temperature sensitive mutant (PMID: 15772160) and investigated its effect on fluorescently tagged Exo84 and Cdc10. We found that the *sec10-2* mutant exhibited a noticeably larger septin ring diameter, while Exo84 cluster diameter is only slightly increased (Fig. R5). Importantly, consistent with our experiments, mimicking impaired vesicle fusion in the computational SBER model by decreasing the exocytosis rate also resulted in larger septin rings (Fig. R5). In the revised manuscript, we included these new results (Fig. S9), which further validate our SBER model and support our original conclusion.

Figure R5. Experiments with *sec10-2* temperature-sensitive mutant and reduction of exocytosis rate in the SBER model show enlarged septin rings. Cells were analyzed between 170 and 410 min after shifting to the non-permissive temperature. (a) Representative microscopy images for wild-type and *sec10-2* cells. (b) Quantification of Exo84 cluster diameter and corresponding cell volume: WT (n = 121), *sec10-2* (n = 80). (c) Quantification of septin ring diameter and corresponding cell volume: WT (n = 380), *sec10-2* (n = 175). (d) Modeling results for normal (0.4 s^{-1} , n = 10) and reduced (0.2 s^{-1} , n = 10) exocytosis rate. Two independent replicates were performed for the experiments. Computer icon - modeling results. Microscope icon - experimental results.

3) Conclusions are not always supported by the results. The authors should pay special attention to talking about conclusions supported by modeling versus conclusions supported by the biology. For example: "Exocytosis-aided recruitment of septins explains the increase of septin ring size upon diffused exocytosis" is found on page 13. What the authors show is that diffuse exocytosis changes ring size, and mathematical modeling suggests that exocytosis aided recruitment could explain that. They do not show that exocytosis-aided recruitment happens in cells (see major concern 2). There needs to be much more careful wording around these conclusions. Discussions of the role of protein levels suffers from the same issue.

As the reviewer suggested, we extensively worked on the clarification of which results were acquired via experiments and which were acquired via computational modeling. We added additional clarification to the main text. Also, we added icons corresponding to experimental results (microscope) and modeling results (computer) to each figure.

4) The English and grammar and individual parts of the paper are well written, but overall the clarity is lacking. The narrative can be hard to follow and the switching between models and experiments can be hard to keep track of. This is just an issue of text changes though, and I am sure the authors can make this better.

We are grateful for the feedback. We extensively revised the manuscript to improve its logical structure and clarity. Changes are highlighted in the text with yellow markings.

Minor Issues

1) Authors include cell numbers for experiments, but do not indicate how many independent experiments these are from.

We have now clarified in each figure that "At least two independent replicates were performed for the experiments".

2) In the introduction, the following is stated with no reference "However, the septin ring diameter can be decoupled from the cluster area by deleting the formin BNI1, which leads to a larger septin ring, even though the Cdc42 cluster area is mostly unchanged." It must be referring to Kuhktevich et al. 2020, but as this is a major undertone of the current work, it would be useful to emphasize that you have previously found this to be true and are now going to investigate it further, as this finding is also a component of figure 3. I think this data being in figure 3 is fine, especially with the additional formin associated proteins being tested as well, it should just be more explicitly stated.

We added the suggested clarification in the revised manuscript, more specifically, to the introduction.

3) Figure 1 looks pixelated, hopefully this is just an artifact of the PDF conversion for the submission system.

We agree with the reviewer that this is most likely an artifact of the PDF conversion. We double-checked the revised version of our manuscript before submitting.

4) For figure 1F, how were thresholds chosen? Is it in an unbiased way? A particular algorithm?

Fig. 1f shows a schematic of how the analysis is performed for the data shown in Fig. 1g. The approach is described in detail in the methods section of our manuscript, specifically in the "Analysis of the Cdc42" subsection of "Quantification and statistical analysis". We ensured that we communicated this more clearly in the revised version of the manuscript by adding "for more details on analysis, see Methods" to the figure caption.

5) Is "sublinearly" the correct description here? It appears that you mean a lower slope, but the data still appears to be linear?

The 'sublinear' scaling refers to the dependence of protein amount on cell volume, not the double logarithmic plot shown in Fig. 2a&b. In the manuscript, we defined a 'protein slope' as the slope of the logarithm of concentration as a function of the logarithm of cell volume. Protein amounts increasing in proportion to cell volume (i.e. 'linear scaling' of amounts with

cell volume) correspond to a 'protein slope' of 0. A negative 'protein slope' would translate to sublinear scaling of the protein amount with cell volume, with the specific case of a slope of -1 leading to constant amounts, that is a scaling of concentration with the inverse of cell volume. We thank the reviewer for pointing out that our manuscript was not clear here, and we have clarified the meaning of a negative protein slope in the manuscript.

6) Why is Gic2 in the SCGE data set and not the SCD data set? Given the robust change in Gic2 in SCGE, should you also check its paralogue Gic1?

From the dataset from Lanz et al., we filtered for the following key Cdc42 polarity proteins: Cdc24, Bud3, Bem2, Bem3, Rga1, Rga2, Cla4, Ste20, Bem1, Gic1, Gic2, Axl2, Rsr1, Cdc42, Cdc11, Cdc12, Cdc3, Cdc10. Unfortunately, Lanz et al. could not reliably gather data on all proteins, and most noticeably, Gic1 is not present in the dataset, and for Gic2, only data for the SCGE media was included. We have clarified in the caption of Fig. 2 that data is missing for a subset of polarity proteins, and in the methods, we now provide the list of proteins we filtered for.

7) In the septin model in figure 5, what causes septins to convert to polymerized?

Following the work of Okada et al., and as described in the methods, septin is polymerized when septin reacts with itself; $S + S \rightarrow 2P$ and $S + P \rightarrow P + P$. To make this clearer in the main text, we have also added this information to the caption of Fig. 5.

8) I don't follow why increased PSR decouples septin size from Cdc42 patch size when the Cdc42 patch is what is putting those Polarity factors in place. Is there a significant different in turn over rate? Are the polarity factors limiting, such that more cdc42 machinery doesn't lead to more polarity factors being present?

We thank the reviewer for pointing out that we were unclear on the mechanism here. As the SBER model is a highly non-linear spatial model, deducing exactly why is not trivial. Inspection of concentration dynamics in the SBER model revealed that stronger septin recruitment depletes the pool of polarity factors that normally inhibit septin polymerization. As a result, more septin can polymerize close to the cluster center, resulting in a more concentrated ring that can recruit more septin-associated GAPs. This increased GAP activity shrinks the Cdc42-GTP cluster upon ring formation, and because Cdc42-GTP-associated proteins recruit septins, this cascade produces a smaller septin ring in comparison with the projection for WT with a similar Cdc42-GTP cluster. These dynamics are in line with what we also see in Fig. S11, where the Cdc42 GTP cluster in Cdc24^{38A} reduces drastically in size upon ring formation. Thus, the reviewer is correct that polarity factors, in a sense, become limiting. We have added this clarification to the discussion. We also highlight that the model's decoupling is weaker than observed experimentally. Thus, there are likely other mechanisms, outside the scope of this study, also contributing here.

9) Figure 7 does not leave me feeling like I fully understand what you have shown. It feels more like a summary of results than of conclusions.

We improved the clarity of Fig. 7 in the revised manuscript, considering also our new experimental and modeling results (see Fig. R6 below and Fig. 7 in the manuscript).

Figure R6. Illustration summarizing the mechanisms determining Cdc42-GTP cluster area and septin ring diameter. Positive feedback, together with polarity proteins increasing in amount as cell size increases, leads to an increase of Cdc42-GTP cluster area with cell volume. As a consequence, the septin ring diameter increases. The coupling of Cdc42-GTP cluster area and septin ring diameter can be disrupted by more diffused exocytosis or by a disrupted negative feedback in the Cdc42 polarization pathway.

Dear Kurt,

Thank you for submitting a revised version of your manuscript. I sincerely apologise for the protracted assessment process for your manuscript due to the delay in reviewer report submission, as well as the pre-publication quality control checks.

We have now received input from both original reviewers, who are generally satisfied with the added revisions. Therefore, there now remain only a few editorial points that need to be addressed before I can extend official acceptance of the manuscript:

1. Please reduce the number of keywords to maximally five.
2. Please check that the funding information is correct and identical both in the manuscript and our online system. Currently, the Swedish Foundation for Strategic Research (FFL15-0238) is missing in our system.
3. Please complete the columns D and E of the author checklist as appropriate. The completed author checklist will also be part of the Review Process File.
4. Please make sure that the order of the sections in the manuscript is as follows: abstract, introduction, results, discussion, materials & methods, data availability section, acknowledgments, disclosure statement and competing interests, references, main figure legends, tables, expanded figure legends.
5. Please remove figures from the manuscript text file.
6. Figure panels 3A-E are not mentioned in the manuscript text. Please add the corresponding callouts.
7. Please include Code Availability in the Data Availability section. More information about the format of this section can be found here: <https://www.embopress.org/page/journal/14602075/authorguide#dataavailability>.
8. We can accommodate up to five Expanded View (EV) figures, which should be uploaded as individual, production-quality files, with their legends compiled at the end of the manuscript text file. Please assemble the rest of the EV figures together with their legends into a single PDF file labelled "Appendix", and rename them into Appendix Figure S1, etc. The Appendix will also need a brief table of contents. The pages should be numbered.
9. All Materials and Methods need to be described in the main text using our 'Structured Methods' format. According to this format, the Methods section includes a Reagents and Tools Table (listing key reagents, experimental models, software and relevant equipment and including their sources and relevant identifiers) followed by a Methods and Protocols section describing the methods, ideally using a step-by-step protocol format. The aim is to facilitate adoption of the methodologies across labs. Please download and fill our Reagents and Tools Table template (.docx), which you can find in our author guidelines: <https://www.embopress.org/page/journal/14602075/authorguide#structuredmethods>
The list of the used strains currently included in the Table EV1 should be added in an appropriate section of the Reagents and Tools table template as indicated.
When submitting your revised manuscript, please do not include the Reagents and Tools Table in the Methods section of the manuscript but upload it as a separate file choosing the file type "Reagent Table".
An example of a Method paper with Structured Methods can be found here: <https://www.embopress.org/doi/10.15252/msb.20178071>.
10. In our standard image integrity check, we have noticed image reuse in figure panels 2G, 6C and EV5A. If this is intentional, please indicate the reuse in the figure legends.
11. Please submit the completed source data checklist and the source data files as requested by our source data coordinator. I have attached the source data checklist below.
12. Our data editors have flagged the following issues in figure legends that need correcting:
 - Please provide the exact p values in the legends of figures 1D, G.
 - Please indicate the statistical test used for data analysis in the legends of figures 1D, G.
 - Please note that the box plots need to be defined in terms of minima, maxima, centre, bounds of box and whiskers, and percentile in the legends of figures 2G, 3B-E; 6D, EV3 C, EV5 B, C; EV6 B, EV9 B, C.
 - Please define the error bars in the legend of figure EV6B.
 - Please define the measure of center for the error bars in the legends of figures 4B, C, E, F, G; 5I, 6B.
13. Papers published in The EMBO Journal are accompanied online by a 'Synopsis' to enhance discoverability of the manuscript. It consists of A) a short (1-2 sentences) summary of the findings and their significance, B) 3-4 bullet points highlighting key results and C) a synopsis image that is 550x300-600 pixels large (width x height, jpeg or png format). You can either show a model or key data in the synopsis image. Please note that the image size is rather small and that text needs to be readable at the final size.

With best wishes,

leva

leva Gailite, PhD
Senior Scientific Editor
The EMBO Journal
Meyerhofstrasse 1
D-69117 Heidelberg
Tel: +4962218891309
i.gailite@embojournal.org

We realize that it is difficult to revise to a specific deadline. In the interest of protecting the conceptual advance provided by the work, we recommend a revision within 3 months (21st Oct 2025). Please discuss the revision progress ahead of this time with the editor if you require more time to complete the revisions.

Referee #1:

The authors have addressed my initial concerns by providing a parameter sensitivity analysis , performing better imaging, and considering previous relevant work in their discussion.

Referee #2:

The changes to the manuscript and the new experiments have adequately addressed my concerns, and I recommend this study for publication. My only suggestion at this point is that the authors should add statistical analysis for the new data related to the hemizygous deletions in Figure EV5 prior to publication.

The authors addressed the remaining editorial issues.

Dear Kurt,

Thank you for addressing the final editorial requests. I am now pleased to inform you that your manuscript has been accepted for publication - congratulations!

Before we forward your manuscript to our publishers, we would like to propose minor edits in the manuscript title, abstract and synopsis, which you can find in the attached text file. I have also written a short blurb that will accompany the title of your manuscript in our online table of contents. Please take a look and let me know if any corrections or adjustments are needed.

If you have any questions, please do not hesitate to contact the Editorial Office. Thank you for this contribution to The EMBO Journal and congratulations on a nice study!

With best wishes,

Ieva
